# Adapting Stepsizes by Momentumized Gradients Improves Optimization and Generalization

## Abstract

Adaptive gradient methods, such as ADAM, have achieved tremendous success in machine learning. Scaling gradients by square roots of the running averages of squared past gradients, such methods are able to attain rapid training of modern deep neural networks. Nevertheless, they are observed to generalize worse than stochastic gradient descent (SGD) and tend to be trapped in local minima at an early stage during training. Intriguingly, we discover that substituting the gradient in the second moment estimation term with the momentumized version in ADAM can well solve the issues. The intuition is that gradient with momentum contains more accurate directional information and therefore its second moment estimation is a better choice for scaling than that of the raw gradient. Thereby we propose ADAMOMENTUM as a new optimizer reaching the goal of training fast while generalizing better. We further develop a theory to back up the improvement in optimization and generalization and provide convergence guarantees under both convex and nonconvex settings. Extensive experiments on a wide range of tasks and models demonstrate that ADAMOMENTUM exhibits state-of-the-art performance consistently. The source code is available at `https://anonymous.4open.science/r/AdaMomentum_experiments-6D9B`.

## 1 Introduction

Prevailing first-order optimization algorithms in modern machine learning can be classified into two categories. One is stochastic gradient descent (SGD) (Robbins & Monro, 1951), which is widely adopted due to its low memory cost and outstanding performance. SGDM (Sutskever et al., 2013) which incorporates the notion of momentum into SGD, has become the best choice for optimizer in computer vision. The drawback of SGD(M) is that it scales the gradient uniformly in all directions, making the training slow especially at the begining and fail to optimize complicated models well beyond Convolutional Neural Networks (CNN). The other type is adaptive gradient methods. Unlike SGD, adaptive gradient optimizers adapt the stepsize (a.k.a. learning rate) elementwise according to the gradient values. Specifically, they scale the gradient by the square roots of some form of the running average of the squared values of the past gradients. Popular examples include AdaGrad (Duchi et al., 2011), RMSprop (Tijmen Tieleman, 2012) and Adam (Kingma & Ba, 2015) etc. Adam, in particular, has become the default choice for many machine learning application areas owing to its rapid optimizing speed and outstanding ability to handle sophisticated loss curvatures.

Despite their fast speed in the early training phase, adaptive gradient methods are found by studies (Wilson et al., 2017; Zhou et al., 2020) to be more likely to exhibit poorer generalization ability than SGD. This is discouraging because the ultimate goal of training in many machine learning tasks is to exhibit high performance during testing phase. In recent years researchers have put many efforts to mitigate the deficiencies of adaptive gradient algorithms. AMSGrad (Reddi et al., 2019) corrects the errors in the convergence analysis of Adam and proposes a faster version. Yogi (Reddi et al., 2018) takes the effect of batch size into consideration. M-SVAG (Balles & Hennig, 2018) transfers the variance adaptation mechanism from Adam to SGD. AdamW (Loshchilov & Hutter, 2017b) first-time decouples weight decay from gradient descent for Adam-alike algorithms. SWATS (Keskar & Socher, 2017) switches from Adam to SGD throughout the training process via a hard schedule and AdaBound (Luo et al., 2019) switches with a smooth transation by imposing dynamic bounds

on stepsizes. RAdam (Liu et al., 2019) rectifies the variance of the adaptive learning rate through investigating the theory behind warmup heuristic (Vaswani et al., 2017; Popel & Bojar, 2018). AdaBelief (Zhuang et al., 2020) adapts stepsizes by the belief in the observed gradients. Nevertheless, most of the above variants can only surpass (as they claim) Adam or SGD in limited tasks or under specifically and carefully defined scenarios. Till today, SGD and Adam are still the top options in machine learning, especially deep learning (Schmidt et al., 2021). Conventional rules for choosing optimizers are: from task perspective, choose SGDM for vision, and Adam (or AdamW) for language and speech; from model perspective, choose SGDM for Fully Connected Networks and CNNs, and Adam (or AdamW) for Recurrent Neural Networks (RNN) (Cho et al., 2014; Hochreiter & Schmidhuber, 1997b), Transformers (Vaswani et al., 2017) and Generative Adversarual Networks (GAN) (Goodfellow et al., 2014). Based on the above observations, a natural question is:

*Is there a computationally efficient adaptive gradient algorithm that can converge fast and meanwhile generalize well?*

In this work, we are delighted to discover that simply replacing the gradient term in the second moment estimation term of Adam with its momentumized version can achieve this goal. Our idea comes from the origin of Adam optimizer, which is a combination of RMSprop and SGDM. RMSprop scales the current gradient by the square root of the exponential moving average (EMA) of the squared past gradients, and Adam replaces the raw gradient in the numerator of the update term of RMSprop with its EMA form, i.e., with momentum. Since the momentumized gradient is a more accurate estimation of the appropriate direction to descent, we consider putting it in the second moment estimation term as well. We find such operation makes the optimizer more suitable for the general loss curvature and can theoretically converge to minima that generalize better. Extensive experiments on a broad range of tasks and models indicate that: without bells and whistles, our proposed optimizer can be as good as SGDM on vision problems and outperforms all the competitors in other tasks, meanwhile maintaining fast convergence speed. Our algorithm is efficient with no additional memory cost, and applicable to a wide range of scenarios in machine learning, especially deep learning. More importantly, AdaMomentum requires little effort in hyperparameter tuning and the default parameter setting for adaptive gradient method works well consistently in our algorithm.

**Notation** We use $t, T$ to symbolize the current and total iteration number in the optimization process. $\theta \in \mathbb{R}^d$ denotes the model parameter and $f(\theta) \in \mathbb{R}$ denotes the loss function. We further use $\theta_t$ to denote the parameter at step $t$ and $f_t$ to denote the noisy realization of $f$ at time $t$ because of the mini-batch stochastic gradient mechanism. $g_t$ denotes the $t$-th time gradient and $\alpha$ denotes stepsize. $m_t, v_t$ represent the EMA of the gradient and the second moment estimation term at time $t$ of adaptive gradient methods respectively. $\epsilon$ is a small constant number added in adaptive gradient methods to refrain the denominator from being too close to zero. $\beta_1, \beta_2$ are the decaying parameter in the EMA formulation of $m_t$ and $v_t$ correspondingly. For any vectors $a, b \in \mathbb{R}^d$, we employ $\sqrt{a}, a^2, |a|, a/b, a \geq b, a \leq b$ for elementwise square root, square, absolute value, division, greater or equal to, less than or equal to respectively. For any $1 \leq i \leq d$, $\theta_{t,i}$ denotes the $i$-th element of $\theta_t$. Given a vector $x \in \mathbb{R}^d$, we use $\|x\|_2$ to denote its $l_2$-norm and $\|x\|_\infty$ to denote its $l_\infty$-norm.

## 2 ALGORITHM

**Preliminaries & Motivation** Omitting the debiasing operation and the damping term $\epsilon$, the adaptive gradient methods can be generally written in the following form:

$$\theta_{t+1} = \theta_t - \alpha \frac{m_t}{\sqrt{v_t}}. \tag{1}$$

Here $m_t, v_t$ are called the first and second moment estimation terms. When $m_t = g_t$ and

Table 1: Comparison of AdaMomentum and classic adaptive gradient methods in $m_t$ and $v_t$ in (1).

| Optimizer | $m_t$ | $v_t$ |
|---|---|---|
| SGD | $g_t$ | $1$ |
| Rprop | $g_t$ | $g_t^2$ |
| RMSprop | $g_t$ | $(1-\beta_2)\sum_{i=1}^t \beta_2^{t-i} g_i^2$ |
| Adam | $(1-\beta_1)\sum_{i=1}^t \beta_1^{t-i} g_i$ | $(1-\beta_2)\sum_{i=1}^t \beta_2^{t-i} g_i^2$ |
| **Ours** | $(1-\beta_1)\sum_{i=1}^t \beta_1^{t-i} g_i$ | $(1-\beta_2)\sum_{i=1}^t \beta_2^{t-i} \boldsymbol{m}_i^2$ |

$v_t = 1$, (1) degenerates to the vanilla SGD. Rprop (Duchi et al., 2011) is the pioneering work using the notion of adaptive learning rate, in which $m_t = g_t$ and $v_t = g_t^2$. Actually it is equivalent to only using the sign of gradients for different weight parameters. RMSprop (Tijmen Tieleman, 2012) forces the number divided to be similar for adjacent mini-batches by incorporating momentum acceleration into $v_t$. Adam (Kingma & Ba, 2015) is built upon RMSprop in which it turns

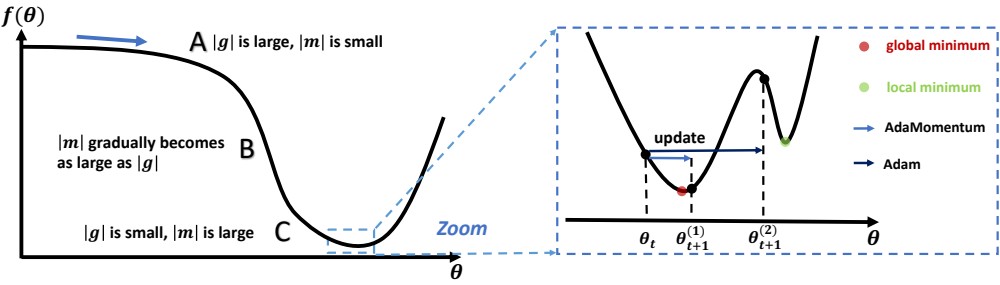

Figure 1: Illustration of the optimization process of Adam and AdaMomentum. A general loss curve can be composed to three areas: **A)** transition from a plateau to a downgrade; **B)** a steep downgrade; **C)** from downgrade to entering the basin containing the optimum. An ideal optimizer ought to sustain large stepsize before reaching the optimum and reduce its stepsize near the optimum. Compared to Adam, AdaMomentum can adapt the true stepsize more appropriately along the loss curve and maintain smaller stepsize near convergence. Refer to Section 3.1 for more detailed analysis.

$g_t$ into momentumized version. Both RMSprop and Adam boost their performance thanks to the smoothing property of EMA using momentum. Due to the fact that momentumized gradient is a more accurate estimation than raw gradient, we deem that there is no reason to use $g_t$ in lieu of $m_t$ in second moment estimation term $v_t$. Therefore we propose to replace the $g_i$s in $v_t$ of Adam with their momentumized versions $m_i$s, which further smooths the exponential moving average.

---

**Algorithm 1** AdaMomentum (ours). All mathematical operations are element-wise.

---
1: **Initialization** : Parameter initialization $\theta_0$, step size $\alpha$, damping term $\epsilon$, $m_0 \leftarrow 0, v_0 \leftarrow 0, t \leftarrow 0$
2: **while** $\theta_t$ not converged **do**
3:      $t \leftarrow t + 1$      $\triangleright$ Updating time step
4:      $g_t \leftarrow \nabla_\theta f_t(\theta_{t-1})$      $\triangleright$ Acquiring stochastic gradient at time $t$
5:      $m_t \leftarrow \beta_1 m_{t-1} + (1 - \beta_1) g_t$      $\triangleright$ EMA of gradients
6:      $v_t \leftarrow \beta_2 v_{t-1} + (1 - \beta_2) m_t{}^2 + \epsilon$      $\triangleright$ EMA of squared momentumized gradients
7:      $\widehat{m_t} \leftarrow m_t / (1 - \beta_1^t)$      $\triangleright$ Bias correction of first moment estimation
8:      $\widehat{v_t} \leftarrow v_t / (1 - \beta_2^t)$      $\triangleright$ Bias correction of second moment estimation
9:      $\theta_t \leftarrow \theta_{t-1} - \alpha \cdot \widehat{m_t} / \sqrt{\widehat{v_t}}$      $\triangleright$ Updating parameters
10: **end while**

---

**Detailed Algorithm** The detailed procedure of our proposed optimizer is displayed in Algorithm 1. There are two major modifications based on Adam, which are marked in red and blue respectively. One is that we replace the $g_t$ in $v_t$ of Adam with $m_t$, which is the momentumized gradient. Hence we name our proposed optimzier as *AdaMomentum*. The other is the location of $\epsilon$ (in Adam $\epsilon$ is added after $\sqrt{\cdot}$ in line 10 of Alg.1). We discover that moving the adding $\epsilon$ from outside the radical symbol to inside can consistently enhance performance. To the best of our knowledge, our method is the first attempt to put momentumized gradient in the second moment estimation term of adaptive gradient methods. Note that although the modifications seem simple to some degree, they can lead to siginificant changes in the performance of an adaptive gradient optimizer due to the iterative nature of optimization methods, which will also be elaborated in the following section.

## 3 WHY ADAMOMENTUM OVER ADAM?

### 3.1 ADAMOMENTUM IS MORE SUITABLE FOR GENERAL LOSS CURVATURE

In this section, we show that AdaMomentum can converge to (global) minima faster than Adam does via a 1-D example. The left part of Figure 1 is the process of optimization from a plateau to a basin area, where global optimum is assumed to exist. The right part is the zoomed-in version of the situation near the minimum, where we have some peaks and valleys. This phenomenon frequently takes place in optimization since there is only one global minimum with probably a great number of local minima surrounding (Hochreiter & Schmidhuber, 1997a; Keskar et al., 2017).

**Benefits of substituting $g_t$ with $m_t$.** We first explain how substituting $m_t$ for $g_t$ in the preconditioner $v_t$ can improve training via decomposing the trajectory of parameter point along the loss curve. **1)** In area A, the parameter point starts to slide down the curve and $|g_t|$ begins to enlarge abruptly. So the actual stepsize $\alpha/\sqrt{v_t}$ is small for Adam. However the absolute value of the momentumized gradient $m_t$ is small since it is the EMA of the past gradients, making $\alpha/\sqrt{v_t}$ still large for AdaMomentum. Hence AdaMomentum can maintain higher training speed than Adam in this changing corner of the loss curve, which is what an optimal optimizer should do. **2)** In area B, since the exponential moving average decays the impact of past gradients exponentially w.r.t. $t$, the magnitude of the elements of $m_t$ will gradually becomes as large as $g_t$. **3)** In area C, when the parameter approaches the basin, the magnitude of $g_t$ decreases, making the stepsizes of Adam increase immediately. In contrast, the stepsize of AdaMomentum is still comparatively small as $|m_t|$ is still much larger than $|g_t|$, which is desired for an ideal optimizer. Small stepsize near optimum has benefits for convergence and stability. A more concrete illustration is given in the right part of Figure 1. If the stepsize is too large (e.g. in Adam), the weight parameter $\theta_t$ may rush to $\theta_{t+1}^{(2)}$ and miss the global optimum. In contrast, small stepsize can guarantee the parameter to be close to the global minimum (see $\theta_{t+1}^{(1)}$) even if there may be tiny oscillations within the basin before the final convergence.

**Benefits of changing the location of $\epsilon$.** Next we elaborate why putting $\epsilon$ under the $\sqrt{\cdot}$ is beneficial. We denote the debiased second moment estimation in AdaMomentum as $\widehat{v}_t$ and the second moment estimation term without $\epsilon$ as $\widehat{v}_t'$. By simple calculation, we have

$$\widehat{v}_t = \left((1-\beta_2)/(1-\beta_2^t)\right) \cdot \sum_{i=1}^{t} \beta_2^{t-i} m_i^2 + \frac{\epsilon}{1-\beta_2}, \quad \widehat{v}_t' = \left((1-\beta_2)/(1-\beta_2^t)\right) \cdot \sum_{i=1}^{t} \beta_2^{t-i} m_i^2.$$

Hence we have $\widehat{v}_t = \widehat{v}_t' + \epsilon/(1-\beta_2)$. Then the actual stepsizes are $\alpha/(\sqrt{\widehat{v}_t' + \epsilon/(1-\beta_2)})$ and $\alpha/(\sqrt{\widehat{v}_t'} + \epsilon)$ respectively. In the final stage of optimization, $\widehat{v}_t'$ is very close to 0 (because the values of gradients are near 0) and far less than $\epsilon$ hence the actual stepsizes can be approximately written as $\sqrt{1-\beta_2}\alpha/\sqrt{\epsilon}$ and $\alpha/\epsilon$. As $\epsilon$ usually takes very tiny values ranging from $10^{-8}$ to $10^{-16}$ and $\beta_2$ usually take values that are extremely close to 1 (usually 0.999), we have $\sqrt{1-\beta_2}\alpha/\sqrt{\epsilon} \ll \alpha/\epsilon$. Therefore we may reasonably come to the conclusion that after moving $\epsilon$ term into the radical symbol, AdaMomentum further reduces the stepsizes when the training is near minima, which contributes to enhancing convergence and stability as we have discussed above.

## 3.2 ADAMOMENTUM CONVERGES TO MINIMA THAT GENERALIZE BETTER

The outline of Adam and our proposed AdaMomentum can be written in the following unified form:

$$m_t = \beta_1 m_{t-1} + (1-\beta_1)g_t, \quad v_t = \beta_2 v_{t-1} + (1-\beta_2)k_t^2,$$

$$\theta_{t+1} = \theta_t - \alpha\, m_t \Big/ \left((1-\beta_1^t)\sqrt{v_t/(1-\beta_2^t)}\right). \tag{2}$$

where $k_t = g_t$ in Adam and $k_t = m_t$ in AdaMomentum. Inspired by a line of work (Pavlyukevich, 2011; Simsekli et al., 2019; Zhou et al., 2020), we can consider (2) as a discretization of a continuous-time process and reformulate it as its corresponding Lévy-driven stochastic differential equation (SDE). Assuming that the gradient noise $\zeta_t = g_t - \nabla f(\theta_t)$ is independent and centered symmetric $\widetilde{\alpha}$-stable ($\mathcal{S}\widetilde{\alpha}\mathcal{S}$) (Lévy & Lévy, 1954) distributed with covariance matrix $\Sigma_t$ possessing a heavy-tailed signature ($\widetilde{\alpha} \in (0,2]$), we are able to derive the Lévy-driven SDE of (2) as:

$$d\theta_t = -q_t R_t^{-1} m_t dt + \upsilon R_t^{-1} \Sigma_t dL_t, \quad dm_t = \beta_1(\nabla f(\theta_t) - m_t), \quad dv_t = \beta_2(k_t^2 - v_t), \tag{3}$$

where $R_t = \mathrm{diag}(\sqrt{v_t/(1-\beta_2^t)})$, $\upsilon = \alpha^{1-1/\widetilde{\alpha}}$, $q_t = 1/(1-\beta_1^t)$ and $L_t$ is the $\widetilde{\alpha}$-stable Lévy motion with independent components. We are interested in the local stability of the optimizers and therefore we suppose process (3) is initialized in a local basin $\mathbf{\Omega}$ with a minimum $\theta^*$ (w.l.o.g., we assume $\theta^* = \mathbf{0}$). To investigate the escaping behavior of $\theta_t$, we first introduce two technical definitions.

**Definition 1 (Radon Measure (Simon et al., 1983)).** If a measure $m(\cdot)$ defined on the $\sigma$-algebra of Borel sets of a Hausdorff topological space $X$ is 1) inner regular on open sets, 2) outer regular on all Borel sets, and 3) finite on all compact sets, then the measure is called a Radon measure.

**Definition 2 (Escaping Time & Escaping Set).** We define escaping time $\Gamma := \inf\{t \geq 0 : \theta_t \notin \mathbf{\Omega}^{-\upsilon^\gamma}\}$, where $\mathbf{\Omega}^{-\upsilon^\gamma} = \{y \in \mathbf{\Omega} : \mathrm{dis}(\partial\mathbf{\Omega}, y) \geq \upsilon^\gamma\}$. Here $\gamma > 0$ is a constant. We define escaping set $\Upsilon := \{y \in \mathbb{R}^d : R_{\theta^*}^{-1}\Sigma_{\theta^*}y \notin \mathbf{\Omega}^{-\upsilon^\gamma}\}$, where $\Sigma_{\theta^*} = \lim_{\theta_t \to \theta^*}\Sigma_t$, $R_{\theta^*} = \lim_{\theta_t \to \theta^*}R_t$.

We study the relationship between $\Gamma$ and $\Upsilon$ and impose some standard assumptions before proceeding.

**Assumption 1.** $f$ is non-negative with an upper bound, and locally $\mu$-strongly convex in $\mathbf{\Omega}$.

**Assumption 2.** There exists some constant $L > 0$, s.t. $\|\nabla f(x) - \nabla f(y)\|_2 \leq L \|x - y\|_2, \forall x, y$.

**Assumption 3.** We assume that $\int_0^\Gamma \langle \nabla f(\theta_t)/(1 + f(\theta_t)), q_t R_t^{-1} m_t \rangle \, dt \geq 0$ a.e., and $\beta_1 \leq \beta_2 \leq 2\beta_1$. There exist $v_-, v_+ > 0$ s.t. each coordinate of $\sqrt{v_t}$ can be uniformly bounded in $(v_-, v_+)$ and there exist $\tau_m, \tau > 0$ s.t. $\|m_t - \widehat{m}_t\|_2 \leq \tau_m \left\| \int_0^{t^-} (m_x - \widehat{m}_x) \, dx \right\|_2$ and $\|\widehat{m}_t\|_2 \geq \tau \left\| \nabla f(\widehat{\theta}_t) \right\|_2$, where $\widehat{m}_t$ and $\widehat{\theta}_t$ are calculated by solving (3) with $v = 0$.

Assumption 1 and 2 impose some standard assumptions of stochastic optimization Ghadimi & Lan (2013); Johnson & Zhang (2013). Assumption 3 requires momentumized gradient $m_t$ and $\nabla f(\theta_t)$ to have similar directions for most of the time, which have been empirically justified to be true in Adam (Zhou et al., 2020). Based on the above assumptions, we can prove that for algorithm of form (2), the expected escaping time is inversely proportional to the Radon measure of the escaping set:

**Lemma 1.** Under Assumptions 1-3, let $v^{\widetilde{\alpha}+1} = \Theta(\widetilde{\alpha})$ and $\ln\left(2\Delta/(\mu v^{1/3})\right) \leq 2\mu\tau(\beta_1 - \beta_2/4)/(\beta_1 v_+ + \mu\tau)$, where $\Delta = f(\theta_0) - f(\theta^*)$. Then given any $\theta_0 \in \mathbf{\Omega}^{-2v^\gamma}$, for (3) we have

$$\mathbb{E}(\Gamma) = \Theta(v/m(\Upsilon)),$$

where $m(\cdot)$ is a non-zero Radon measure satisfying that $m(\mathcal{U}) < m(\mathcal{V})$ if $\mathcal{U} \subset \mathcal{V}$.

Because larger set has larger volume, i.e., $V(\mathcal{U}) \leq V(\mathcal{V})$ if $\mathcal{U} \subset \mathcal{V}$, from Lemma 1 we have the escaping time is negatively correlated with the volume of the set $\Upsilon$. Therefore, we can come to the conclusion that for both Adam and AdaMomentum, if the basin $\mathbf{\Omega}$ is sharp which is ubiquitous during the early stage of training, $\Upsilon$ has a large Radon measure, which leads to smaller escaping time $\Gamma$. This means both Adam and AdaMomentum prefer relatively flat or asymmetric basin He et al. (2019) through the training process.

On the other hand, upon encountering a comparatively flat basin or asymmetric valley $\mathbf{\Omega}$, we are able to prove that AdaMomentum will stay longer inside. Before we proceed, we need to impose two mild assumptions.

**Assumption 4.** There exists a constant $H > 0$ s.t. $\|\nabla f(\theta_t)\|_2 \leq H, \|g_t\|_2 \leq H, \forall t \in [T]$.

**Assumption 5.** For AdaMomentum, there exists $T_0 \in \mathbb{N}$ s.t., $\mathrm{diag}(\Sigma_t) \leq \beta_1 \mathbb{E}(m_{t-1}^2)/(2 - \beta_1)$ when $t > T_0$.

Here Assumption 4 is a common assumption in stochastic optimization (Ghadimi & Lan, 2013; Johnson & Zhang, 2013). As $\beta_1$ is always set as positive number close to 1, Assumption 5 basically requires that the gradient noise variance to be smaller than the second moment of $m$ when $t$ is very large. This assumption is mild as 1) we can select mini-batch size to be large enough to satisfy it as the noise variance is inversely proportional to batch size (Bubeck, 2014). 2) The magnitudes of the variances of the stochastic gradients are usually much lower than that of the gradients (Faghri et al., 2020). Then we can come to the following result.

**Proposition 1.** Under Assumptions 1-5, upon encountering a comparatively flat basin or asymmetric valley $\mathbf{\Omega}$, we have

$$\mathbb{E}\left(\Gamma^{(\text{ADAMOMENTUM})}\right) \geq \mathbb{E}\left(\Gamma^{(\text{ADAM})}\right).$$

In other words, when falling into a flat/asymmetric basin, AdaMomentum is more stable than Adam and will not easily escape from it. Combining the aforementioned results and the fact that minima at the flat or asymmetric basins tend to exhibit better generalization performance (as observed in Keskar et al. (2017); He et al. (2019); Hochreiter & Schmidhuber (1997a); Izmailov et al. (2018); Li et al. (2018)), we are able to conclude that AdaMomentum is more likely to converge to minima that generalize better, which may buttress the improvement of AdaMomentum in empirical performance. All the proofs in section 3.2 are provided in Appendix B.

## 4 CONVERGENCE ANALYSIS OF ADAMOMENTUM

In this section, we establish the convergence theory for AdaMomentum under both convex and non-convex object function conditions. We omit the two bias correction steps in the Algorithm 1 for simplicity and the following analysis can be easily adapted and applied to the de-biased version as well.

### 4.1 CONVERGENCE ANALYSIS IN CONVEX OPTIMIZATION

We analyze the convergence of AdaMomentum in convex setting utilizing the online learning framework (Zinkevich, 2003). Given a sequence of convex cost functions $f_1(\theta), \cdots, f_T(\theta)$, the regret is defined as $R(T) = \sum_{t=1}^T [f_t(\theta_t) - f_t(\theta^*)]$, where $\theta^* = \arg\min_\theta \sum_{t=1}^T f_t(\theta)$ is the optimal parameter and $f_t$ can be interpreted as the loss function at the $t$-th step. Then we have:

**Theorem 1.** Let $\{\theta_t\}$ and $\{v_t\}$ be the sequences yielded by AdaMomentum. Let $\alpha_t = \alpha/\sqrt{t}, \beta_{1,1} = \beta_1, 0 < \beta_{1,t} \leq \beta_1 < 1, v_t \leq v_{t+1}$ for all $t \in [T]$ and $\gamma = \beta_1/\sqrt{\beta_2} < 1$. Assume that the distance between any $\theta_t$ generated by AdaMomentum is bounded, $\|\theta_m - \theta_n\|_\infty \leq D_\infty$ for any $m, n \in \{1, \cdots, T\}$. Then we have the following bound on the regret:

$$R(T) \leq \frac{D_\infty^2 \sqrt{T}}{2\alpha(1-\beta_1)} \sum_{i=1}^{d} \sqrt{v_{T,i}} + \frac{D_\infty^2}{2(1-\beta_1)} \sum_{t=1}^{T} \sum_{i=1}^{d} \frac{\beta_{1,t}\sqrt{v_{t,i}}}{\alpha_t} + \frac{\alpha\sqrt{1+\log T}}{(1-\beta_1)^3(1-\gamma)\sqrt{1-\beta_2}} \sum_{i=1}^{d} \|g_{1:T,i}\|_2.$$

Theorem 1 implies that the regret of AdaMomentum can be bounded by $\widetilde{O}^1(\sqrt{T})$, especially when the data features are sparse as Section 1.3 in Duchi et al. (2011) and then we have $\sum_{i=1}^{d} \sqrt{v_{T,i}} \ll \sqrt{d}$ and $\sum_{i=1}^{d} \|g_{1:T,i}\|_2 \ll \sqrt{dT}$. When we impose additional assumptions that $\beta_{1,t}$ decays exponentially and that the gradients of $f_t$ are bounded (Kingma & Ba, 2015; Liu et al., 2019), we can obtain the following corollary:

**Corollary 1.** Further Suppose $\beta_{1,t} = \beta_1\lambda^t$ and the function $f_t$ has bounded gradients, $\|\nabla f_t(\theta)\|_\infty \leq G_\infty$ for all $\theta \in \mathbb{R}^d$, AdaMomentum achieves the guarantee $R(T)/T = \widetilde{O}(1/\sqrt{T})$ for all $T \geq 1$:

$$\frac{R(T)}{T} \leq \frac{dG_\infty\alpha\sqrt{1+\log T}}{(1-\beta_1)^3(1-\gamma)\sqrt{(1-\beta_2)T}} + \frac{dD_\infty^2 G_\infty}{2\alpha(1-\beta_1)\sqrt{T}} + \frac{dD_\infty^2 G_\infty\beta_1}{2\alpha(1-\beta_1)(1-\lambda)^2 T}.$$

Clearly observed from Corollary 1, the average regret of AdaMomentum converges to zero as $T$ goes to infinity. The proofs of Theorem 1 and Corollary 1 are provided in Appendix C.1.

## 4.2 CONVERGENCE ANALYSIS IN NON-CONVEX OPTIMIZATION

When $f$ is non-convex and lower-bounded, we derive the non-asymptotic convergence rate of AdaMomentum.

**Theorem 2.** We suppose that Assumptions 2 and 4 hold, and $\beta_{1,t}$ is chosen such that $0 \leq \beta_{1,t+1} \leq \beta_{1,t} < 1, 0 < \beta_{2,t} < 1, \forall t \in [T]$. We further assume $\alpha_t/\sqrt{v_t} \geq \alpha_{t+1}/\sqrt{v_{t+1}}$ for any $t \in [T]$, $\sum_{t=1}^{T} \alpha_t^2 \leq \eta(T) \lesssim T\alpha_T$, $\min_{t \in [T], j \in [d]} v_{t,j} \geq c \geq \epsilon$ and $\min_{t \in [T]} \alpha_t \leq \alpha$, then we have:

$$\min_{t \in [T]} \mathbb{E}\|\nabla f(\theta_t)\|_2^2 \leq \frac{H}{T\alpha_T}\left[\frac{C_1 H^2 \eta(T)}{c} + C_2\frac{d\alpha}{\sqrt{c}} + C_3\frac{d\alpha^2}{c} + C_4\right]$$
$$= \frac{1}{T\alpha_T}(Q_1 + Q_2\eta(T)),$$

for some positive constants $Q_1, Q_2$. Here $C_1, C_2, C_3$ are positive constants independent of $d$ or $T$, while $C_4$ is a positive constant independent of $T$.

Note that the conditions in Theorem 2 can be satisfied in most scenarios (Kingma & Ba, 2015; Chen et al., 2019; Reddi et al., 2019). For instance, we can simply employ the common setting $\alpha_t = \alpha/\sqrt{t}, \beta_{2,t} = 1/t$.

**Corollary 2.** When $\alpha_t$ is further chosen to be $\alpha/\sqrt{t}$, AdaMomentum satisfies:

$$\min_{t \in [T]} \mathbb{E}\|\nabla f(\theta_t)\|_2^2 \leq \frac{H}{\sqrt{T}\alpha}\left[\frac{C_1 H^2 \alpha^2(1+\log(T))}{c} + C_2\frac{d\alpha}{\sqrt{c}} + C_3\frac{d\alpha^2}{c} + C_4\right]$$
$$= \frac{1}{\sqrt{T}}(Q_1^* + Q_2^*\log(T)),$$

for some constants $Q_1^*, Q_2^*$. $C_1, C_2, C_3, C_4$ are similarly defined in Theorem 2.

Corollary 2 manifests the $O(\log(T)/\sqrt{T})$ convergence rate of AdaMomentum in the nonconvex case when we commonly use $\alpha_t = \alpha/\sqrt{t}$. We refer readers to the detailed proof in Appendix C.2.

## 5 EXPERIMENTS

We empirically investigate the performance of AdaMomentum in both optimization and generalization.

### 5.1 FASTER & BETTER OPTIMIZATION

#### 5.1.1 2-D TOY EXAMPLES

We compare the optimization process of AdaMomentum with three prevalent optimziers SGDM, RMSprop, Adam and recently proposed AdaBelief , on four classic and representative 2-variable objective functions in

---

[1] $\widetilde{O}(\cdot)$ denotes $O(\cdot)$ with hidden logarithmic factors.

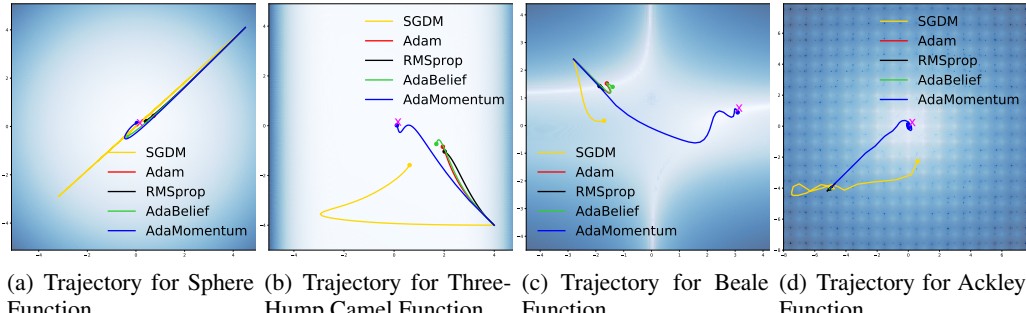

(a) Trajectory for Sphere Function.   (b) Trajectory for Three-Hump Camel Function.   (c) Trajectory for Beale Function.   (d) Trajectory for Ackley Function.

Figure 2: 2D Trajectory visualization of SGDM, Adam, RMSprop, AdaBelief and AdaMomentum on classic functions. AdaMomentum reaches the optimal point (marked as purple cross) the fastest in all the cases and converges stably to the optimum without big oscillations. Best viewed in color.

Table 2: FID score ($\downarrow$) of Spectral Normalized Generative Adversarial Network on CIFAR-10 (Krizhevsky & Hinton, 2009) dataset. $\dagger$ is reported in Zhuang et al. (2020).

| SGDM[†] | Adam(W)[†] | Yogi[†] | AdaBound[†] | RAdam[†] | AdaBelief[†] | **Ours** |
|---|---|---|---|---|---|---|
| $49.70 \pm 0.41$ | $13.05 \pm 0.19$ | $14.25 \pm 0.15$ | $55.65 \pm 2.15$ | $12.70 \pm 0.12$ | $12.52 \pm 0.16$ | $\mathbf{12.06 \pm 0.21}$ |

numerical optimization literature: Sphere Function (bowl-shaped) (Dixon, 1978), Three-Hump Camel Function (valley-shaped)[2], Beale Function (multimodal)[3] and Ackley Function (with numerous local minima) (Adorio & Diliman, 2013). To ensure fair comparison, we use the same hyperparameters in the four adaptive gradient methods and finetune the learning rate of SGDM for the best performance. As illustrated in Figure 2, AdaMomentum achieves the most rapid convergence in all the cases and is stable once reaching the global mininum. Meanwhile, SGDM is highly unstable and inaccurate in the descending directions, and RMSprop, Adam, AdaBelief is much slower. Although these toy examples are simple, they give hints to the behavior of optimizers in complex deep learning tasks as they can be viewed as the local dynamics which occur frequently in deep learning (Zhuang et al., 2020). The details of the loss functions and hyperparameter configurations of the experiment are in Appendix A. The GIFs and the 3D trajectory figures are included in the supplementary.

### 5.1.2 GENERATIVE ADVERSARIAL NETWORK

Training of GANs is extremely unstable. To further study the optimization ability and stability of AdaMomentum, we experiment on GAN equipped with spectal normalization (Miyato et al., 2018). For the generator and the discriminator network, we adopt ResNets for adequate expression ability. We train the model for 100000 iterations on CIFAR-10 with batch size 64, and the two learning rates are set both as 0.0002. For AdaMomentum all the other hyperparameters are set as default values. Experiments are run 5 times independently and we report the mean and standard deviation of Frechet Inception Distance (FID, the lower the better) Heusel et al. (2017) in Table 2. From Table 2 it is reasonable to draw the conclusion that AdaMomentum outperforms all the best tuned baseline optimizers by a large margin, reaching mean FID score as low as 12.06 with its default hyperparameter values. Here Adam equals AdamW because the optimal weight decay parameter value is 0.

### 5.2 SUPERIOR GENERALIZATION

We conduct experiments on various modern network architectures for different tasks covering both vision and language processing area: **1)** image Classification on CIFAR-10 (Krizhevsky & Hinton, 2009) and ImageNet (Russakovsky et al., 2015) with CNN; **2)** language modeling on Penn Treebank (Marcus et al., 1993) dataset using Long Short-Term Memory (LSTM) (Hochreiter & Schmidhuber, 1997b); **3)** neural machine translation on IWSTL'14 DE-EN (Cettolo et al., 2014) dataset employing Transformer. We compare AdaMomentum with seven state-of-the-art optimizers: SGDM (Sutskever et al., 2013), Adam (Kingma & Ba, 2015), AdamW (Loshchilov & Hutter, 2017b), Yogi (Reddi et al., 2018), AdaBound (Luo et al., 2019), RAdam (Liu et al., 2019) and AdaBelief (Zhuang et al., 2020). We perform a careful and extensive hyperparameter tuning for

---

[2]https://en.wikipedia.org/wiki/Test_functions_for_optimization
[3]http://www-optima.amp.i.kyoto-u.ac.jp/member/student/hedar/Hedar_files/TestGO.htm

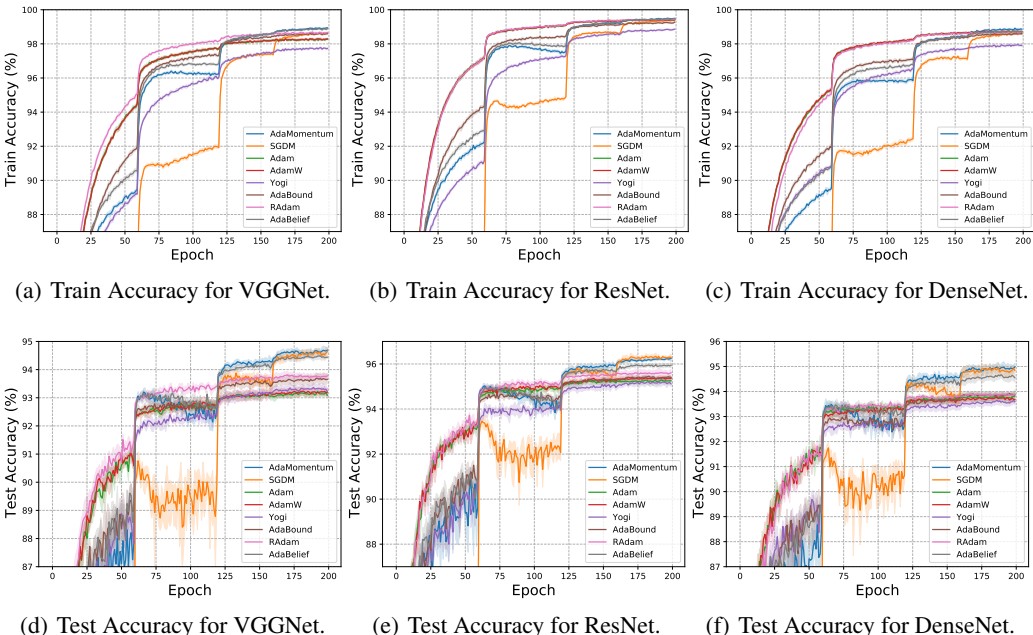

| (a) Train Accuracy for VGGNet. | (b) Train Accuracy for ResNet. | (c) Train Accuracy for DenseNet. |
|:---:|:---:|:---:|
| (d) Test Accuracy for VGGNet. | (e) Test Accuracy for ResNet. | (f) Test Accuracy for DenseNet. |

Figure 3: Train and test accuracy of different optimizers on CIFAR-10 (Krizhevsky & Hinton, 2009).

all the optimizers compared, and the detailed strategy and configuration are given in Appendix D due to space limit. It is worth mentioning that in experiments we discover that setting $\alpha = 0.001, \beta_1 = 0.9, \beta_2 = 0.999$ (the default setting for adaptive gradient methods in applied machine learing) works well in most cases. This elucidates that our optimizer is tuning-friendly, which reduces human labor and time cost and is crucial in practice. The mean results with standard deviations over 5 seeds are reported in all the following experiments.

### 5.2.1 CNN FOR IMAGE CLASSIFICATION

Table 3: Test accuracy (%) of CNNs on CIFAR-10 dataset. The best in Red and second best in blue.

| Architecture | Non-adaptive | Adaptive gradient methods | | | | | | |
|---|---|---|---|---|---|---|---|---|
| | SGDM | Adam | AdamW | Yogi | AdaBound | RAdam | AdaBelief | **Ours** |
| VGGNet-16 | $94.73^{\pm0.12}$ | $93.29^{\pm0.10}$ | $93.33^{\pm0.15}$ | $93.44^{\pm0.16}$ | $93.79^{\pm0.17}$ | $93.90^{\pm0.10}$ | $94.57^{\pm0.09}$ | $94.80^{\pm0.10}$ |
| ResNet-34 | $96.47^{\pm0.09}$ | $95.39^{\pm0.11}$ | $95.48^{\pm0.10}$ | $95.28^{\pm0.19}$ | $95.51^{\pm0.07}$ | $95.67^{\pm0.16}$ | $96.04^{\pm0.07}$ | $96.33^{\pm0.07}$ |
| DenseNet-121 | $95.03^{\pm0.19}$ | $93.92^{\pm0.20}$ | $93.87^{\pm0.14}$ | $93.72^{\pm0.18}$ | $93.99^{\pm0.08}$ | $94.00^{\pm0.07}$ | $94.74^{\pm0.14}$ | $95.08^{\pm0.19}$ |

Table 4: Top-1 test accuracy (%) on ImageNet (Russakovsky et al., 2015) dataset.

| SGDM | Adam | AdamW | Yogi | AdaBound | RAdam | AdaBelief | **Ours** |
|---|---|---|---|---|---|---|---|
| 70.41±0.13 | 65.36±0.25 | 68.77±0.14 | 68.93±0.08 | 69.32±0.19 | 69.24±0.12 | 69.98±0.09 | **70.45±0.06** |

**CIFAR-10** We experimented with three prevailing deep CNN architectures: VGG-16 (Simonyan & Zisserman, 2015), ResNet-34 (He et al., 2016) and DenseNet-121 (Huang et al., 2017). The growth rate of DenseNet-121 is set as 12 to match CIFAR-10 dataset. In each experiment we train the model for 200 epochs with batch size 128 and decay the learning rate by 0.2 at the 60-th, 120-th and 160-th epoch. We employ label smoothing technique (Szegedy et al., 2016) and the smoothing factor is choosen as 0.1. Figure 3 displays the training and testing results of all the compared optimizers . As indicated, both the training accuracy and the testing accuracy using AdaMomentum can be improved as fast as with other adaptive gradient methods, being much faster than SGDM, especially before the third learning rate annealing. In testing phase, AdaMomentum can exhibit performance as good as SGDM and far exceeds other baseline adaptive gradient methods, including the recently proposed AdaBelief (Zhuang et al., 2020) optimizer. This contradicts the result reported in Zhuang et al. (2020), where they claim AdaBelief can be better than SGDM. This largely stems from the fact that Zhuang et al. (2020) did not take an appropriate stepsize annealing strategy or tune the hyperparameters well. Training 200 epochs with ResNet-34 on CIFAR-10, our experiments show that AdaMomentum and SGDM can reach over 96% accuracy, while in Zhuang et al. (2020) the accuracy of SGDM is only around 94% .

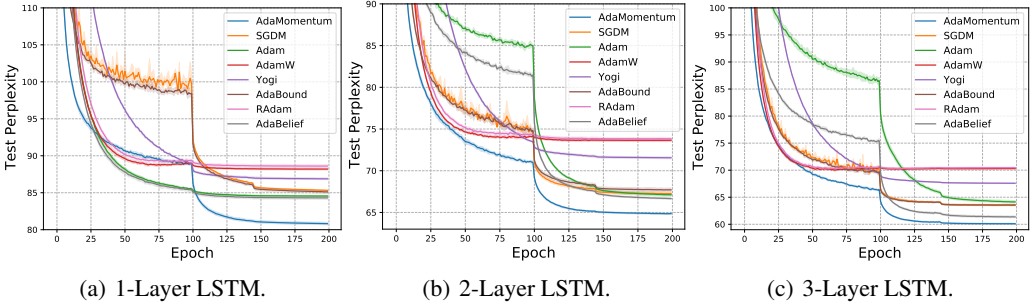

Figure 4: Test perplexity curve on Penn Treebank (Marcus et al., 1993) dataset.

Table 5: Test perplexity (↓) results of LSTMs on Penn Treebank (Marcus et al., 1993) dataset.

| Layer # | SGDM | Adam | AdamW | Yogi | AdaBound | RAdam | AdaBelief | **Ours** |
|---|---|---|---|---|---|---|---|---|
| 1 | $85.31^{\pm0.09}$ | $84.55^{\pm0.10}$ | $88.18^{\pm0.14}$ | $86.87^{\pm0.14}$ | $85.10^{\pm0.22}$ | $88.60^{\pm0.22}$ | $84.30^{\pm0.23}$ | $\mathbf{80.82^{\pm0.19}}$ |
| 2 | $67.25^{\pm0.20}$ | $67.11^{\pm0.20}$ | $73.61^{\pm0.15}$ | $71.54^{\pm0.14}$ | $67.69^{\pm0.24}$ | $73.80^{\pm0.25}$ | $66.66^{\pm0.11}$ | $\mathbf{64.85^{\pm0.09}}$ |
| 3 | $63.52^{\pm0.16}$ | $64.10^{\pm0.25}$ | $69.91^{\pm0.20}$ | $67.58^{\pm0.08}$ | $63.52^{\pm0.11}$ | $70.10^{\pm0.16}$ | $61.33^{\pm0.19}$ | $\mathbf{60.08^{\pm0.11}}$ |

**ImageNet**   To further corroborate the effectiveness of our algorithm on more comprehensive dataset, we perform experiments on ImageNet utilizing ResNet-18 as backbone network. We execute each optimizer for 90 epochs utilizing cosine annealing strategy, which can exhibit better performance results than step-based decay strategy on ImageNet (Loshchilov & Hutter, 2017a; Ma, 2021). As indicated in Table 4, AdaMomentum far exceeds Adam in Top-1 test accuracy and outperforms all the competitors including SGD with momentum.

### 5.2.2   LSTM for Language Modeling

We implement LSTMs with layer number from 1 to 3 on Penn Treebank dataset, where adaptive gradient methods are the main-stream choices (much better than SGD). In each experiment we train the model for 200 epochs with batch size of 20 and decay the learning rate by 0.1 at 100-th and 145-th epoch. Test perplexity (the lower the better) against training epochs is plotted in Figure 4 and the best perplexity value is summarized in Table 5. Clealy observed from Figure 4 and Table 5, AdaMomentum achieves the lowest perplexity in all the settings and consistently outperform other competitors by a considerable margin. The training curve is given in Figure 5 in Appendix D due to space limit. Particularly on 2-layer and 3-layer LSTM, AdaMomentum maintains both the fastest convergence and the best performance, which substantiates its superiority.

### 5.2.3   Transformer for Neural Machine Translation

Transformers have been the dominating architecture in NLP and adaptive gradient methods are usually employed to train transformers due to their stronger ability to handle attention-models (Zhang et al., 2019). To test AdaMomentum on transformer, we experiment on IWSTL'14 German-to-English with

Table 6:  BLEU score (↑) on IWSTL'14 DE-EN (Cettolo et al., 2014) dataset.

| SGDM | Adam | AdamW | AdaBelief | **Ours** |
|---|---|---|---|---|
| 28.22±0.24 | 30.14±1.56 | 35.62±0.13 | 35.60±0.12 | **35.66±0.11** |

the Transformer *small* model adapting the code from fairseq package.[4] We set the length penalty as 1.0, the beam size as 5, warmup initial stepsize as $10^{-7}$ and the warmup updates iteration number to be 8000. We train the models for 55 epochs and the results are reported according to the average of the last 5 checkpoints. As shown in Table 12, our optimizer achieves the highest average BLEU score with the lowest variance.

## 6   Conclusion

In this work, we proposed AdaMomentum as a new optimizer for machine learning. We theoretically demonstrate why AdaMomentum outperforms Adam in optimization and generalization. We further validates the superiority of AdaMomentum through both toy examples and large-scale experiments on real-world datasets. Our algorithm is simple and effective with four key advantages: **1)** maintaining fast convergence rate; **2)** closing the generalization gap between adaptive gradient methods and SGD; **3)** applicable to various tasks and models; **4)** introducing no additional parameters and easy to tune. Combination of AdaMomentum with other techniques such as Nesterov's accelerated gradient (Dozat, 2016) may be of independent interest in the future.

---

[4]https://github.com/pytorch/fairseq

## 7 ETHICS STATEMENT

Our work follows all ethical standards and laws. All the experiments were conducted on publically available datasets, with no new data concerning human or animal subjects generated.

## 8 REPRODUCIBILITY STATEMENT

We adhere to ICLR reproducibility standards and provide all necessary information to reproduce our experimental and theoretical results. We ensure the reproducibility of our work through several ways, namely

- All the source code and presented figures are available at anonymous link `https://anonymous.4open.science/r/AdaMomentum_experiments-6D9B`.
- The detailed descriptions of the classic loss functions used in Toy exmpales in Section 5.1.1 are given in Appendix A.
- All the technical details and proofs in Section 3.2 are included in Appendix B. All the proofs in Section 4 are provided in Appendix C.
- The detailed hyperparameter tuning rule and configurations of the experiments in Section 5 are given in Appendix D.

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

# A   DETAILS OF TOY EXAMPLES

The detailed description of the four 2-parameter loss functions are given as below:

The Sphere Function is

$$f(x) = x_1^2 + x_2^2,$$

whose global minimum is $f(x^*) = 0$ at $x^* = (0, \cdots, 0)$.

The Three Hump Camel Function is

$$f(x) = 2x_1^2 - 1.05x_1^4 + \frac{x_1^6}{6} + x_1 x_2 + x_2^2,$$

whose global minimum is $f(x^*) = 0$ at $x^* = (0, \cdots, 0)$.

The Beale Function is

$$f(x) = (1.5 - x_1 + x_1 x_2)^2 + (2.2 - x_1 + x_1 x_2^2)^2 + (2.625 - x_1 + x_1 x_2^3)^2,$$

whose global minimum is $f(x^*) = 0$ at $x^* = (3, 0.5)$.

The Ackley Function is

$$f(x) = -20 \exp\left(-0.2\sqrt{\frac{1}{2}\left(x_1^2 + x_2^2\right)}\right) - \exp\left(\frac{1}{2}\left(\cos(2\pi x_1) + \cos(2\pi x_2)\right)\right) + 20 + \exp(1),$$

whose global minimum is $f(x^*) = 0$ at $x^* = (0, \cdots, 0)$.

**Hyperparameter configuration**   For all the four toy experiments, we employ the same stepsize for the three adaptive gradient methods: RMSProp, Adam and AdaMomentum. For SGDM We set the momentum parameter as $0.9$ and finetune the learning rate for each toy experiment in set $\{0.0001, 0.001, 0.01, 0.1, 1, 10\}$ as the optimal learning rate varies for different loss functions. After careful parameter tuning, the configuration is: for Sphere Function, the learning rate for adaptive gradient methods and SGDM are both $0.1$; for Three-Hump Camel Function, the learning rate for adaptive gradient methods is $0.1$ and the learning rate for SGDM is $0.001$; for Beale Function, the leanring rate for adaptive, gradient methods is $0.1$ and the learning rate for SGDM is $0.1$; for Ackley Function, the leanring rate for adaptive, gradient methods is $0.1$ and the learning rate for SGDM is $0.1$. Combining this with Figure 2, we can see that AdaMomentum with the stepsize $0.1$ manifests universally faster and more stable convergence to the optimum than discreetly tuned SGDM and other adaptive gradient approaches.

# B   TECHNICAL DETAILS OF SUBSECTION 3.2

Here we provide more construction details and technical proofs for the Lévy-driven SDE in Adam-alike adaptive gradient algorithm (2). In the beginning we introduce a detailed derivation of the process (3) as well as its corresponding escaping set $\Upsilon$ in definition 2. Then we give some auxiliary theorems and lemmas, and summarize the proof of Lemma 1. Finally we prove the proposition 1 and give a more detailed analysis of the conclusion that the expected escaping time of AdaMomentum is longer than that of Adam in a comparatively flat basin.

## B.1   DERIVATION OF THE LÉVY-DRIVEN SDE (3)

To derive the SDE of Adam-alike algorithms (2), we firstly define $m_t' = \beta_1 m_{t-1}' + (1 - \beta_1)\nabla f(\theta_t)$ with $m_0' = 0$. Then by the definition it holds that

$$m_t' - m_t = (\beta_1 - 1)\sum_{i=0}^{t} \beta_1^{t-i}\zeta_t.$$

Following Simsekli et al. (2019), the gradient noise $\zeta_t$ has heavy tails in reality and hence we assume that $\frac{1}{1-\beta_1}(m_t' - m_t)$ obeys $\mathcal{S}\widetilde{\alpha}\mathcal{S}$ distribution with time-dependent covariance matrix $\Sigma_t$. Since we can formulate (2) as

$$\theta_{t+1} = \theta_t - \alpha\frac{m_t'}{z_t} + \alpha\frac{(m_t' - m_t)}{z_t} \text{ where } z_t = (1 - \beta_1^t)\sqrt{\frac{v_t}{(1 - \beta_2^t)}}, \qquad (4)$$

and we can replace the term $(m_t' - m_t)$ by $\alpha^{-\frac{1}{\alpha}}(1 - \beta_1^t)\Sigma_t S$ where each coordinate of $S$ is independent and identically distributed as $\mathcal{S}\widetilde{\alpha}\mathcal{S}(1)$ based on the property of centered symmetric $\widetilde{\alpha}$-stable distribution. Let

$R_t = \text{diag}(\sqrt{\frac{v_t}{(1-\beta_2^t)}})$, and we further assume that the step size $\alpha$ is small, then the continuous-time version of the process (4) becomes the following SDE:

$$d\theta_t = -R_t^{-1}\frac{m_t' dt}{(1-\beta_1^t)} + \alpha^{1-\frac{1}{\alpha}}R_t^{-1}\Sigma_t dL_t, \ dm_t = \beta_1(\nabla f(\theta_t) - m_t), \ dv_t = \beta_2(k_t^2 - v_t).$$

After replacing $m_t'$ with $m_t$ for brevity, we get the SDE (3) consequently.

## B.2 PROOF OF LEMMA 1

To prove Lemma 1, we first introduce Theorem 3.

**Theorem 3.** Suppose Assumptions 1-3 hold. We define $\kappa_1 = \frac{c_1 L}{v - |\tau_m - 1|}$ and $\kappa_2 = \frac{2\mu\tau}{\beta_1 v_+ + \mu\tau}\left(\beta_1 - \frac{\beta_2}{4}\right)$ with a constant $c_1$. Let $v^{\tilde{\alpha}+1} = \Theta(\tilde{\alpha})$, $\rho_0 = \frac{1}{16(1+c_2)}$ and $\ln\left(\frac{2\Delta}{\mu v^{1/3}}\right) \leq \kappa_2 v^{-1/3}$ where $\Delta = f(\theta_0) - f(\theta^*)$ and a constant $c_2$. Then for any $\theta_0 \in \Omega^{-2v^\gamma}$, $u > -1$, $v \in (0, v_0]$, $\gamma \in (0, \gamma_0]$ and $\rho \in (0, \rho_0]$ satisfying $v^\gamma \leq \rho_0$ and $\lim_{v \to 0}\rho = 0$, the Adam-alike algorithm in (2) obey

$$\frac{1-\rho}{1+u+\rho} \leq \mathbb{E}\left[\exp\left(-um(\Upsilon)\Theta(v^{-1})\Gamma\right)\right] \leq \frac{1+\rho}{1+u-\rho}.$$

From Theorem 3, by setting $v$ small, it holds that for any adaptive gradient algorithm the upper and lower bounds of its expected escaping time $\Gamma$ is at the order of $\left(\frac{v}{m(\Upsilon)}\right)$, which directly implies Lemma 1 conclusively. Therefore, it suffices to validate Theorem 3.

Theorem 3 is adapted from Theorem 1 in Zhou et al. (2020) and the proof is given in Section B.2.3. Before we proceeed, we first provide some prerequisite notations in Section B.2.1 and list some useful theorems and lemmas in Section B.2.2.

### B.2.1 PRELIMINARIES

For analyzing the uniform Lévy-driven SDEs in (3), we first introduce the Lévy process $L_t$ into two components $\xi_t$ and $\varepsilon_i$, namely

$$L_t = \xi_t + \varepsilon_t, \tag{5}$$

whose characteristic functions are respectively defined as

$$\mathbb{E}\left[e^{i\langle\lambda,\xi_t\rangle}\right] = e^{t\int_{\mathbb{R}^d\setminus\{0\}}\varepsilon I\left\{\|y\|_2 \leq \frac{1}{v^\delta}\right\}\nu(dy)},$$

$$\mathbb{E}\left[e^{i\langle\lambda,\varepsilon_t\rangle}\right] = e^{t\int_{\mathbb{R}^d\setminus\{0\}}\varepsilon I\left\{\|y\|_2 \leq \frac{1}{v^\delta}\right\}\nu(dy)},$$

where $\varepsilon = e^{i\langle\lambda,y\rangle} - 1 - i\langle\lambda,y\rangle I\{\|y\|_2 \leq 1\}$ with $v$ defined in (3) and a constant $\delta$ s.t. $v^{-\delta} < 1$. Accordingly, the Lévy measure $\nu$ of the stochastic processes $\xi$ and $\varepsilon$ are

$$\nu_\xi = \nu\left(A \cap \left\{\|y\|_2 \leq \frac{1}{v^\delta}\right\}\right), \quad \nu_\varepsilon = \nu\left(A \cap \left\{\|y\|_2 \geq \frac{1}{v^\delta}\right\}\right), \quad \text{where } A \in \mathcal{B}(\mathbb{R}^d).$$

Besides, for analysis, we should consider affects of the Lévy motion $L_t$ to the Lévy-driven SDE of Adam variants. Here we define the Lévy-free SDE accordingly:

$$\begin{cases} d\widehat{\theta}_t = & -\mu_t \widehat{Q}_t^{-1}\widehat{m}_t, \\ d\widehat{m}_t = & \beta_1(\nabla f(\widehat{\theta}_t) - \widehat{m}_t), \\ d\widehat{v}_t = & \beta_2(\nabla(f\widehat{\theta}_t)^2 - \widehat{v}_t). \end{cases} \tag{6}$$

where $\widehat{Q}_t = \text{diag}(\sqrt{\widehat{v}_t})$.

### B.2.2 AUXILIARY THEOREMS AND LEMMAS

**Theorem 4** (Zhou et al. (2020)). Suppose Assumptions 1-3 hold. Assume the sequence $\{(\widehat{\theta}_t, \widehat{m}_t, \widehat{v}_t)\}$ are produced by (6). Let $\widehat{s}_t = \frac{h_t}{q_t}\left(\sqrt{\omega_t \widehat{v}_t}\right)$ with $h_t = \beta_1$, $q_t = (1-\beta_1^t)^{-1}$ and $\omega_t = (1-\beta_2^t)^{-1}$. We define $\|x\|_y^2 = \sum_i y_i x_i^2$. Then for Lévy-driven Adam SDEs in (6), its Lyapunov function $\mathcal{L}(t) = f(\widehat{\theta}_t) - f(\widehat{\theta}^*) + \frac{1}{2}\|\widehat{m}_t\|_{\widehat{s}_t^{-1}}$ with the optimum solution $\theta^*$ in the current local basin $\Omega$ obeys

$$\mathcal{L}(t) \leq \Delta\exp\left(-\frac{2\mu\tau}{\beta_1 v_+ + \mu\tau}\left(\beta_1 - \frac{\beta_2}{4}\right)t\right),$$

where $\Delta = f(\widehat{\theta}_0) - f(\widehat{\theta}^*)$ due to $\widehat{m}_0 = 0$. The sequence $\{\widehat{\theta}_t\}$ produced by (6) obeys

$$\left\|\widehat{\theta}_t - \theta^*\right\|_2^2 \le \frac{2\Delta}{\mu} \exp\left(-\frac{2\mu\tau}{\beta_1 v_+ + \mu\tau}\left(\beta_1 - \frac{\beta_2}{4}\right)t\right).$$

**Lemma 2** (Zhou et al. (2020)). (1) The process $\xi$ in the Lévy process decomposition can be decomposed into two processes $\widehat{\xi}$ and linear drift, namely,

$$\xi_t = \widehat{\xi}_t + \mu_v t, \tag{7}$$

where $\widehat{\xi}$ is a zero mean Lévymartingale with bounded jumps.
(2) Let $\delta \in (0,1), \mu_v = \mathbb{E}(\xi_1)$ and $T_v = v^{-\theta}$ for some $\theta > 0, \rho_0 = \rho_0(\delta) = \frac{1-\delta}{4} > 0$ and $\theta_0 = \theta_0(\delta) = \frac{1-\delta}{3} > 0$. Suppose $v$ is sufficiently small such that $\Theta(1) \le v^{-\frac{1-\delta}{6}}$ and $v^{-\rho} - 2(C + \Theta(1))v^{\frac{7}{6}(1-\delta)+\frac{\rho}{2}} \ge 1$ with a constant $C = |\int_{0 < u \le 1} u^2 d\Theta(u)| \in (0, +\infty)$. Then for all $\delta \in (0, \delta_0), \theta \in (0, \theta_0)$ there are $p_0 = p_0(\delta) = \frac{\delta}{2}$ and $v_0 = v_0(\delta, \rho)$ such that the estimates

$$\|v\xi_{T_v}\|_2 = v\|\mu_v\|_2 T_v < v^{2\rho} \text{ and } P([v\xi]_{T_v}^d \ge v^\rho) \le \exp(-v^{-p}),$$

hold for all $p \in (0, p_0]$ and $v \in (0, v_0]$

**Lemma 3** (Zhou et al. (2020)). Let $\delta \in (0,1)$ and $g_{t \ge 0}^t$ be a bounded adapted càdlàg stochastic process with values in $\mathbb{R}^d$, $T_v = v^{-\theta}$, $\theta > 0$. Suppose $\sup_{t \ge 0} \|g^t\|$ is well bounded. Assume $\rho_0 = \rho_0(\delta) = \frac{1-\delta}{16} > 0$, $\theta_0 = \theta_0(\delta) = \frac{1-\delta}{3} > 0$, $p_0 = \frac{\rho}{2}$. For $\widehat{\xi}_t$ in (7), there is $\delta_0 = \delta_0(\delta) > 0$ such that for all $\rho \in (0, \rho_0)$ and $\theta \in (0, \theta_0)$, it holds

$$\mathbb{P}\left(\sup_{0 \le t \le T_v} v\left|\sum_{i=1}^d \int_0^t g_{s-}^i d\widehat{\xi}_s^i\right| \ge v^\rho\right) \le 2\exp\left(-v^{-p}\right),$$

for all $p \in (0, p_0]$ and $0 < v \le v_0$ with $v_0 = v(\rho)$, where $\widehat{\xi}_s^i$ represents the i-th entry in $\widehat{\xi}_s$.

**Lemma 4** (Zhou et al. (2020)). Under Assumptions 1-3 hold, assume $\delta \in (0,1), \rho_0 = \rho_0(\delta) = \frac{1-\delta}{16(1+c_1\kappa_1)} > 0, \theta_0 = \theta_0(\delta) = \frac{1-\delta}{3} > 0, p_0 = \min(\frac{\widehat{\rho}(1+c_1\kappa_1)}{2}, p), \frac{1}{c_2}\ln\left(\frac{2\Delta}{\mu v \widehat{\rho}}\right) \le v^{-\theta_0}$ where $\kappa_1 = \frac{c_2 l}{v_- |\tau_m - 1|}$ and $c_2 = \frac{2\mu\tau}{\beta_1 v_+ + \mu\tau}\left(\beta_1 - \frac{\beta_2}{4}\right)$ in Adam-alike adaptive gradient algorithms. For all $\widehat{\rho} \in (0, \rho_0), p \in (0, p_0], 0 < v \le v_0$ with $v_0 = v_0(\widehat{\rho})$, and $\theta_0 = \widehat{\theta}_0$, we have

$$\sup_{\theta_0 \in \mathbf{\Omega}} \mathbb{P}\left(\sup_{0 \le t < \sigma_1} \left\|\theta_t - \widehat{\theta}_t\right\|_2 \ge 2v^{\widehat{\rho}}\right) \le 2\exp(-v^{-\frac{p}{2}}), \tag{8}$$

where the sequences $\theta_t$ and $\widehat{\theta}_t$ are respectively produced by (3) and (6) in adaptive gradient method.

### B.2.3 Proof of Theorem 3

*Proof.* The idea of this proof comes from (8) we showed in Lemma 4 where the sequence $\theta_t$ and $\widehat{\theta}_t$ start from the same initialization. Based on Theorem 4, we know that the sequence $\{\widehat{\theta}_t\}$ from (6) exponentially converges to the minimum $\theta^*$ of the local basin $\mathbf{\Omega}$. To escape the local basin $\mathbf{\Omega}$, we can either take small steps in the process $\zeta$ or large jumps $J_k$ in the process $\varepsilon$. However, (8) suggests that these small jumps might not be helpful for escaping the basin. And for big jumps, the escaping time $\Gamma$ of the sequence $\{\theta_t\}$ most likely occurs at the time $\sigma_1$ if the big jump $vJ_1$ in the process $\varepsilon$ is large.
The verification of our desired results can be divided into two separate parts, namely establishing upper bound and lower bound of $\mathbb{E}\left[\exp\left(-um(\Upsilon)\Theta(v^{-1})\Gamma\right)\right]$ for any $u > -1$. Both of them can be established based on the following facts:

$$\left|\mathbb{P}\left(R_\theta^{-1}\Sigma_\theta vJ_k \notin \mathbf{\Omega}^{\pm v^\gamma}, \|vJ_k\|_2 \le R\right) - \mathbb{P}\left(R_{\theta^*}^{-1}\Sigma_{\theta^*} vJ_k \notin \mathbf{\Omega}^{\pm v^\gamma}, \|vJ_k\|_2 \le R\right)\right| \le \frac{\delta'}{4} \cdot \frac{\Theta(v^{-1})}{\Theta(v^{-\delta})},$$

$$\left|\mathbb{P}\left(R_\theta^{-1}\Sigma_\theta vJ_k \notin \mathbf{\Omega}, \|vJ_k\|_2 \le R\right) - \mathbb{P}\left(R_{\theta^*}^{-1}\Sigma_{\theta^*} vJ_k \notin \mathbf{\Omega}, \|vJ_k\|_2 \le R\right)\right| \le \frac{\delta'}{4} \cdot \frac{\Theta(v^{-1})}{\Theta(v^{-\delta})},$$

$$\mathbb{P}\left(R_{\theta^*}^{-1}\Sigma_{\theta^*} vJ_k \notin \mathbf{\Omega}\right) - \mathbb{P}\left(R_{\theta^*}^{-1}\Sigma_{\theta^*} vJ_k \notin \mathbf{\Omega}, \|vJ_k\|_2 \le R\right) \le \frac{\delta'}{4} \cdot \frac{\Theta(v^{-1})}{\Theta(v^{-\delta})}. \tag{9}$$

Specifically, for the upper bound of $\mathbb{E}\left[\exp\left(-um(\Upsilon)\Theta(v^{-1})\Gamma\right)\right]$, we consider both the big jumps in the process $\varepsilon$ and small jumps in the process $\zeta$ which may escape the local minimum. Instead of estimating the escaping time $\Gamma$ from $\mathbf{\Omega}$, we first estimate the escaping time $\widetilde{\Xi}$ from $\mathbf{\Omega}^{-\bar{\rho}}$. Here we define the inner part of $\mathbf{\Omega}$ as $\mathbf{\Omega}^{-\bar{\rho}} := \{y \in \mathbf{\Omega} : \text{dis}(\partial\mathbf{\Omega}, y) \ge \bar{\rho}\}$. Then by setting $\bar{\rho} \to 0$, we can use $\widetilde{\Xi}$ for a decent estimation of $\Gamma$.

We denote $\bar{\rho} = \upsilon^{\gamma}$ where $\gamma$ is a constant such that the results of Lemma 2-4 hold. So for the upper bound we mainly focus on $\widetilde{\Xi}$ in the beginning and then transfer the results to $\Gamma$. In the beginning, we can show that for any $u > -1$ it holds that,

$$\mathbb{E}\left[\exp\left(-um(\Upsilon)\Theta(\upsilon^{-1})\widetilde{\Xi}\right)\right] \leq \sum_{k=1}^{+\infty} \mathbb{E}\left[e^{-um(\Upsilon)\Theta(\upsilon^{-1})t_k}I\left\{\widetilde{\Xi} = t_k\right\} + Res_k\right],$$

where

$$Res_k \leq \begin{cases} \mathbb{E}\left[e^{-um(\Upsilon)\Theta(\upsilon^{-1})t_k}I\left\{\widetilde{\Xi} \in (t_{k-1}, t_k)\right\}\right], & \text{if } u \in (-1, 0] \\ \mathbb{E}\left[e^{-um(\Upsilon)\Theta(\upsilon^{-1})t_{k-1}}I\left\{\widetilde{\Xi} \in (t_{k-1}, t_k)\right\}\right], & \text{if } u \in (0, +\infty). \end{cases}$$

Then using the strong Markov property we can bound the first term $\mathbb{E}\left[e^{-um(\Upsilon)\Theta(\upsilon^{-1})t_k}I\left\{\widetilde{\Xi} = t_k\right\}\right]$ as

$$\begin{aligned} R_1 = \sum_{k=1}^{+\infty} \mathbb{E}\left[e^{-um(\Upsilon)\Theta(\upsilon^{-1})t_k}I\left\{\Gamma = t_k\right\}\right] &\leq \frac{\alpha_\upsilon(1+\rho/3)}{1+u\alpha_\upsilon} \sum_{k=1}^{+\infty}\left(\frac{1-\alpha_\upsilon(1-\rho)}{1+u\alpha_\upsilon}\right)^{k-1} \\ &\leq \frac{\alpha_\upsilon(1+\rho/3)}{1+u\alpha_\upsilon} \sum_{k=0}^{+\infty}\left(\frac{1-\alpha_\upsilon(1-\rho)}{1+u\alpha_\upsilon}\right)^{k-1} \\ &= \frac{1+\rho/3}{1+u-\rho}. \end{aligned}$$

On the other hand, for the lower bound of $\mathbb{E}\left[\exp\left(-um(\Upsilon)\Theta(\upsilon^{-1})\Gamma\right)\right]$, we only consider the big jumps in the process $\varepsilon$ which could escape from the basin, and ignore the probability that the small jumps in the process $\zeta$ which may also lead to an escape from the local minimum $\theta^*$. Specifically, we can find a lower bound by discretization:

$$\mathbb{E}\left[\exp\left(-um(\Upsilon)\Theta(\upsilon^{-1})\Gamma\right)\right] \geq \sum_{k=1}^{+\infty} \mathbb{E}\left[\exp\left(-um(\Upsilon)\Theta(\upsilon^{-1})t_k\right)I\{\Gamma = t_k\}\right].$$

Then we can lower bound each term by three equations (9) we just listed here, which implies that for any $\theta_0 \in \mathbf{\Omega}^{-\upsilon^{\gamma}}$,

$$\mathbb{E}\left[e^{-um(\Upsilon)\Theta\upsilon^{-1}\Gamma}\right] \geq \frac{\alpha_\upsilon(1-\rho)}{1+u\alpha_\upsilon} \sum_{k=1}^{+\infty}\left(\frac{1-\alpha_\upsilon(1+\rho)}{1+u\alpha_\upsilon}\right)^{k-1} = \frac{1-\rho}{1+u+\rho},$$

where $\rho \to 0$ as $\upsilon \to 0$. The proof is completed. $\qquad\square$

### B.3 PROOF OF PROPOSITION 1

*Proof.* Since we assumed the minimizer $\theta^* = \mathbf{0}$ in the basin $\Omega$ which is usually small, we can employ second-order Taylor expansion to approximate $\Omega$ as a quadratic basin whose center is $\theta^*$. In other words, we can write

$$\Omega = \left\{y \in \mathbb{R}^d \;\middle|\; f(\theta^*) + \frac{1}{2}y^\top H(\theta^*)y \leq h(\theta^*)\right\},$$

where $H(\theta^*)$ is the Hessian matrix at $\theta^*$ of function $f$ and $h(\theta^*)$ is the basin height. Then according to Definition 2, we have

$$\Upsilon = \left\{y \in \mathbb{R}^d \;\middle|\; y^\top \Sigma_{\theta^*} R_{\theta^*}^{-1} H(\theta^*) R_{\theta^*}^{-1} \Sigma_{\theta^*} y \geq h_f^*\right\}.$$

Here $R_{\theta^*} = \lim_{\theta_t \to \theta^*} \operatorname{diag}(\sqrt{v_t/(1-\beta_2^t)})$ is a matrix depending on the algorithm, $h_f^* = 2(h(\theta^*) - f(\theta^*))$ and $\Sigma_{\theta^*}$ is independent of the algorithm, i.e. the same for Adam and AdaMomentum. Firstly, we will prove that $v_t^{(\text{ADAMOMENTUM})} \geq v_t^{(\text{ADAM})}$ when $t \to \infty$. To clarify the notation, we use $\theta_t, m_t, v_t, g_t$ to denote the symbols for Adam and $\widetilde{\theta}_t, \widetilde{m}_t, \widetilde{v}_t, \widetilde{g}_t$ for AdaMomentum, and $\zeta_t$ is the gradient noise. By using Lemma 1 and above results, we have $\theta_t \approx \widetilde{\theta}_t \approx \theta^*$ before escaping when $t$ is large, and thus $v_t = \lim_{\theta_t \to \theta^*}[\nabla f(\theta_t) + \zeta_t]^2$ and $\widetilde{v}_t = \lim_{\theta_t \to \theta^*}[\beta_1 \widetilde{m}_{t-1} + (1-\beta_1)(\nabla f(\widetilde{\theta}_t) + \zeta_t)]^2$. We will firstly show that $\mathbb{E}(\widetilde{v}_t) \geq \mathbb{E}(v_t)$ when $t$ is large.

$$\begin{aligned} \mathbb{E}(v_t) = \mathbb{E}(\lim_{\theta_t \to \theta^*}[\nabla f(\theta_t) + \zeta_t]^2) &\overset{(i)}{=} \lim_{\theta_t \to \theta^*} \mathbb{E}([\nabla f(\theta_t) + \zeta_t]^2) \\ &= \lim_{\theta_t \to \theta^*}\left(\mathbb{E}(\nabla f(\theta_t)^2) + \mathbb{E}(2\nabla f(\theta_t)\zeta_t) + \mathbb{E}(\zeta_t^2)\right) \\ &\overset{(ii)}{=} \mathbb{E}(\lim_{\theta_t \to \theta^*} \nabla f(\theta_t)^2) + \lim_{\theta_t \to \theta^*}\mathbb{E}(2\nabla f(\theta_t)\zeta_t) + \lim_{\theta_t \to \theta^*}\mathbb{E}(\zeta_t^2) \\ &\overset{(iii)}{=} \lim_{\theta_t \to \theta^*}\mathbb{E}(\zeta_t^2), \end{aligned}$$

where (i) and (ii) are due to the dominated convergence theorem (DCT) since we have that we know both $\|\nabla f(\theta_t)\|_2$ and $\|\nabla f(\theta_t) + \zeta_2\|_2$ could be bounded by $H$ in Assumption 4. And (iii) is due to the fact that $\nabla f(\theta^*) = 0$ since function $f$ attains its minimum point at $\theta^*$, and $\zeta_t$ has zero mean, i.e.

$$\lim_{\theta_t \to \theta^*} \mathbb{E}(\nabla f(\theta_t)\zeta_t) = \lim_{\theta_t \to \theta^*} \mathbb{E}(\nabla f(\theta_t))\mathbb{E}(\zeta_t) = 0.$$

And similarly we can prove that,

$$\mathbb{E}(\widetilde{v}_t) = \mathbb{E}\left(\lim_{\theta_t \to \theta^*}[\beta_1\widetilde{m}_{t-1} + (1-\beta_1)(\nabla f(\widetilde{\theta}_t) + \zeta_t)]^2\right)$$

$$= \lim_{\theta_t \to \theta^*}\left(\mathbb{E}(\beta_1^2\widetilde{m}_{t-1}^2) + \mathbb{E}((1-\beta_1)^2(\nabla f(\widetilde{\theta}_t) + \zeta_t)^2) + \mathbb{E}(2\beta_1(1-\beta_1)\widetilde{m}_{t-1}\nabla(f(\widetilde{\theta}_t) + \zeta_t))\right)$$

$$\overset{(i)}{=} \beta_1^2 \lim_{\theta_t \to \theta^*}\mathbb{E}(\widetilde{m}_{t-1}^2) + (1-\beta_1)^2 \lim_{\theta_t \to \theta^*}\mathbb{E}(\zeta_t^2),$$

where we can get the equality (i) simply by the same argument with dominated convergence theorem we just used:

$$\lim_{\widetilde{\theta}_t \to \theta^*}\mathbb{E}(\nabla(f(\widetilde{\theta}_t)^2)) = \mathbb{E}(\lim_{\widetilde{\theta}_t \to \theta^*}\nabla(f(\widetilde{\theta}_t)^2)) \overset{(i)}{=} 0,$$

$$\lim_{\widetilde{\theta}_t \to \theta^*}\mathbb{E}(\nabla(f(\widetilde{\theta}_t)\zeta_t)) = \mathbb{E}(\lim_{\widetilde{\theta}_t \to \theta^*}\nabla(f(\widetilde{\theta}_t)\zeta_t)) \overset{(ii)}{=} 0,$$

$$\lim_{\widetilde{\theta}_t \to \theta^*}\mathbb{E}(\widetilde{m}_{t-1}(\nabla f(\widetilde{\theta}_t) + \zeta_t)) = \mathbb{E}(\lim_{\widetilde{\theta}_t \to \theta^*}\widetilde{m}_{t-1}\nabla f(\widetilde{\theta}_t)) + \lim_{\widetilde{\theta}_t \to \theta^*}\mathbb{E}(\widetilde{m}_{t-1})\mathbb{E}(\zeta_t) \overset{(iii)}{=} 0,$$

where we get the equality (i) and (ii) since the function $f(\widetilde{\theta}_t)^2$ and $f(\widetilde{\theta}_t)\zeta_t$ could be absolutely bounded by $H^2$. And the first term in equality (iii) is 0 since we have $\|\widetilde{m}_{t-1}\|_2 \leq H$ by its definition and $\nabla f(\theta^*) = 0$, and the second term vanishes since the noise $\zeta_t$ has zero mean. Based on the Assumption 5, we have

$$\mathbb{E}(\widetilde{m}_{t-1}^2) \geq \frac{2-\beta_1}{\beta_1}\mathbb{E}(\zeta_t^2),$$

which implies that $\mathbb{E}(\widetilde{v}_t) \geq \mathbb{E}(v_t)$ when $t$ is large. It further indicates that $R_{\theta^*}^{(\text{ADAMOMENTUM})} \geq R_{\theta^*}^{(\text{ADAM})}$. We consider the volume of the complementary set

$$\Upsilon^c = \left\{y \in \mathbb{R}^d \;\middle|\; y^\top\Sigma_{\theta^*}R_{\theta^*}^{-1}H(\theta^*)R_{\theta^*}^{-1}\Sigma_{\theta^*}y < h_f^*\right\},$$

which can be viewed as a $d$-dimensional ellipsoid. We can further decompose the symmetric matrix $M := \Sigma_{\theta^*}R_{\theta^*}^{-1}H(\theta^*)R_{\theta^*}^{-1}\Sigma_{\theta^*}$ by SVD decomposition

$$M = U^\top A U,$$

where $U$ is an orthogonal matrix and $A$ is a diagonal matrix with nonnegative elements. Hence the transformation $y \to Uy$ is an orthogonal transformation which means the volume of $\Upsilon^c$ equals the volume of set

$$\left\{y' \in \mathbb{R}^d \;\middle|\; y'^\top A y' < h_f^*\right\}.$$

Considering the fact that the volume of a $d$-dimensional ellipsoid centered at $\mathbf{0}$ $E_d(r) = \{(x_1, x_2, \cdots, x_n) : \sum_{i=1}^d \frac{x_i^2}{R_i^2} \leq 1\}$ is

$$V(E_d(r)) = \frac{\pi^{\frac{n}{2}}}{\Gamma(\frac{n}{2}+1)}\Pi_{i=1}^n R_i,$$

and the fact we just proved that $R_{\theta^*}^{(\text{ADAMOMENTUM})} \geq R_{\theta^*}^{(\text{ADAM})}$. Therefore we deduce the volume of $\Upsilon^{(\text{ADAMOMENTUM})}$ is smaller than that of $\Upsilon^{(\text{ADAM})}$, which indicates that for Radon measure $m(\cdot)$ we have $m(\Upsilon^{(\text{ADAMOMENTUM})}) \geq m(\Upsilon^{(\text{ADAM})})$. Based on Lemma 1, we consequently have $\mathbb{E}(\Gamma^{(\text{ADAMOMENTUM})}) \geq \mathbb{E}(\Gamma^{(\text{ADAM})})$. $\qquad\square$

## C  PROOFS IN SECTION 4

### C.1  PROOF OF THE CONVERGENCE RESULTS FOR THE CONVEX CASE

### C.1.1 PROOF OF THEOREM 1

*Proof.* Firstly, according to the definition of AdaMomentum in Algorithm 1, by algebraic shrinking we have

$$\sum_{t=1}^{T} \frac{m_{t,i}^2}{\sqrt{tv_{t,i}}} = \sum_{t=1}^{T-1} \frac{m_{t,i}^2}{\sqrt{tv_{t,i}}} + \frac{\left(\sum_{j=1}^{T}(1-\beta_{1,j})\Pi_{k=1}^{T-j}\beta_{1,T-k+1}g_{j,i}\right)^2}{\sqrt{T\sum_{j=1}^{T}(1-\beta_2)\beta_2^{T-j}m_{j,i}^2}}$$

$$\leq \sum_{t=1}^{T-1} \frac{m_{t,i}^2}{\sqrt{tv_{t,i}}} + \frac{(\sum_{j=1}^{T}\Pi_{k=1}^{T-j}\beta_{1,T-k+1})(\sum_{j=1}^{T}\Pi_{k=1}^{T-j}\beta_{1,T-k+1}g_{j,i}^2)}{\sqrt{T\sum_{j=1}^{T}(1-\beta_2)\beta_2^{T-j}m_{j,i}^2}}$$

$$\overset{(i)}{\leq} \sum_{t=1}^{T-1} \frac{m_{t,i}^2}{\sqrt{tv_{t,i}}} + \frac{(\sum_{j=1}^{T}\beta_1^{T-j})(\sum_{j=1}^{T}\beta_1^{T-j}g_{j,i}^2)}{\sqrt{T(1-\beta_2)\sum_{j=1}^{T}\beta_2^{T-j}m_{j,i}^2}}$$

$$\leq \sum_{t=1}^{T-1} \frac{m_{t,i}^2}{\sqrt{tv_{t,i}}} + \frac{1}{1-\beta_1} \frac{\sum_{j=1}^{T}\beta_1^{T-j}g_{j,i}^2}{\sqrt{T(1-\beta_2)\sum_{j=1}^{T}\beta_2^{T-j}m_{j,i}^2}}$$

$$= \sum_{t=1}^{T-1} \frac{m_{t,i}^2}{\sqrt{tv_{t,i}}} + \frac{1}{(1-\beta_1)\sqrt{T(1-\beta_2)}} \sum_{j=1}^{T} \frac{\beta_1^{T-j}g_{j,i}^2}{\sqrt{\sum_{j=1}^{T}\beta_2^{T-j}\left(\sum_{l=1}^{j}(1-\beta_{1,l})\Pi_{k=1}^{j-l}\beta_{1,j-k+1}g_{l,i}\right)^2}}$$

$$\leq \sum_{t=1}^{T-1} \frac{m_{t,i}^2}{\sqrt{tv_{t,i}}} + \frac{1}{(1-\beta_1)\sqrt{T(1-\beta_2)}} \sum_{j=1}^{T} \frac{\beta_1^{T-j}g_{j,i}^2}{\sqrt{\sum_{j=1}^{T}\beta_2^{T-j}\left((1-\beta_{1,j})g_{j,l}\right)^2}}$$

$$\leq \sum_{t=1}^{T-1} \frac{m_{t,i}^2}{\sqrt{tv_{t,i}}} + \frac{1}{(1-\beta_1)\sqrt{T(1-\beta_2)}} \sum_{j=1}^{T} \frac{\beta_1^{T-j}g_{j,i}^2}{\sqrt{\beta_2^{T-j}(1-\beta_{1,j})^2 g_{j,i}^2}}$$

$$\overset{(ii)}{\leq} \sum_{t=1}^{T-1} \frac{m_{t,i}^2}{\sqrt{tv_{t,i}}} + \frac{1}{(1-\beta_1)^2\sqrt{T(1-\beta_2)}} \sum_{j=1}^{T} \gamma^{T-j}g_{j,i},$$

where (i) arises from $\beta_{1,t} \leq \beta_1$, and (ii) comes from the definition that $\gamma = \frac{\beta_1}{\sqrt{\beta_2}}$. Then by induction, we have

$$\sum_{t=1}^{T} \frac{m_{t,i}^2}{\sqrt{tv_{t,i}}} \leq \sum_{t=1}^{T} \frac{1}{(1-\beta_1)^2\sqrt{t(1-\beta_2)}} \sum_{j=1}^{t} \gamma^{t-j}g_{j,i}$$

$$\leq \frac{1}{(1-\beta_1)^2\sqrt{1-\beta_2}} \sum_{t=1}^{T} \frac{1}{\sqrt{t}} \sum_{j=1}^{t} \gamma^{t-j}g_{j,i}$$

$$\overset{(i)}{\leq} \frac{1}{(1-\beta_1)^2\sqrt{1-\beta_2}} \sum_{t=1}^{T} g_{t,i} \sum_{j=t}^{T} \frac{\gamma^{j-t}}{\sqrt{j}}$$

$$\leq \frac{1}{(1-\beta_1)^2\sqrt{1-\beta_2}} \sum_{t=1}^{T} g_{t,i} \sum_{j=t}^{T} \frac{\gamma^{j-t}}{\sqrt{t}}$$

$$\leq \frac{1}{(1-\beta_1)^2\sqrt{1-\beta_2}} \sum_{t=1}^{T} g_{t,i} \cdot \frac{1}{(1-\gamma)\sqrt{t}}$$

$$\leq \frac{1}{(1-\beta_1)^2(1-\gamma)\sqrt{1-\beta_2}} \sum_{t=1}^{T} \frac{g_{t,i}}{\sqrt{t}}$$

$$\overset{(ii)}{\leq} \frac{1}{(1-\beta_1)^2(1-\gamma)\sqrt{1-\beta_2}} \|g_{1:T,i}\|_2 \sqrt{\sum_{t=1}^{T} \frac{1}{t}}$$

$$\overset{(iii)}{\leq} \frac{\sqrt{1+\log T}}{(1-\beta_1)^2(1-\gamma)\sqrt{1-\beta_2}} \|g_{1:T,i}\|_2,$$

where (i) exchangings the indices of summing, (ii) employs Cauchy-Schwarz Inequality and (iii) comes from the following bound on harmonic sum:

$$\sum_{t=1}^{T} \frac{1}{t} \leq 1 + \log T.$$

Due to convexity of $f_t$, we get

$$f_t(\theta_t) - f_t(\theta^*) \le g_t^\top(\theta_t - \theta^*)$$

$$= \sum_{i=1}^d g_{t,i}(\theta_{t,i} - \theta_{,i}^*). \tag{10}$$

According to the updating rule, we have

$$\theta_{t+1} = \theta_t - \alpha_t \frac{m_t}{\sqrt{v_t}}$$

$$= \theta_t - \alpha_t \left( \frac{\beta_{1,t}}{\sqrt{v_t}} m_{t-1} + \frac{1-\beta_{1,t}}{\sqrt{v_t}} g_t \right). \tag{11}$$

Substracting $\theta^*$, squaring both sides and considering only the $i$-th element in vectors, we obtain

$$(\theta_{t+1,i} - \theta_{,i}^*)^2 = (\theta_{t,i} - \theta_{,i}^*)^2 - 2\alpha_t \left( \frac{\beta_{1,t}}{\sqrt{v_{t,i}}} m_{t-1,i} + \frac{1-\beta_{1,t}}{\sqrt{v_{t,i}}} g_{t,i} \right)(\theta_{t,i} - \theta_{,i}^*) + \alpha_t^2 \left( \frac{m_{t,i}}{\sqrt{v_{t,i}}} \right)^2.$$

By rearranging the terms, we have

$$2\alpha_t \frac{1-\beta_{1,t}}{\sqrt{v_{t,i}}} g_{t,i}(\theta_{t,i} - \theta_{,i}^*) = (\theta_{t,i} - \theta_{,i}^*)^2 - (\theta_{t+1,i} - \theta_{,i}^*)^2 - 2\alpha_t \cdot \frac{\beta_{1,t}}{\sqrt{v_{t,i}}} \cdot m_{t-1,i}(\theta_{t,i} - \theta_{,i}^*) + \alpha_t^2 \left( \frac{m_{t,i}}{\sqrt{v_{t,i}}} \right)^2.$$

Further we have

$$g_{t,i}(\theta_{t,i} - \theta_{,i}^*) = \frac{\sqrt{v_{t,i}}}{2\alpha_t(1-\beta_{1,t})} [(\theta_{t,i} - \theta_{,i}^*)^2 - (\theta_{t+1,i} - \theta_{,i}^*)^2] + \frac{\alpha_t\sqrt{v_{t,i}}}{2(1-\beta_{1,t})} \left( \frac{m_{t,i}}{\sqrt{v_{t,i}}} \right)^2$$

$$+ \frac{\beta_{1,t}}{1-\beta_{1,t}}(\theta_{,i}^* - \theta_{t,i})m_{t-1,i}$$

$$= \frac{\sqrt{v_{t,i}}}{2\alpha_t(1-\beta_{1,t})} [(\theta_{t,i} - \theta_{,i}^*)^2 - (\theta_{t+1,i} - \theta_{,i}^*)^2] + \frac{\alpha_t\sqrt{v_{t,i}}}{2(1-\beta_{1,t})} \left( \frac{m_{t,i}}{\sqrt{v_{t,i}}} \right)^2$$

$$+ \frac{\beta_{1,t}}{1-\beta_{1,t}} \cdot \frac{v_{t,i}^{\frac{1}{4}}}{\sqrt{\alpha_t}} \cdot (\theta_{,i}^* - \theta_{t,i}) \cdot \sqrt{\alpha_t} \cdot \frac{m_{t-1,i}}{v_{t,i}^{\frac{1}{4}}}$$

$$\le \frac{\sqrt{v_{t,i}}}{2\alpha_t(1-\beta_1)} [(\theta_{t,i} - \theta_{,i}^*)^2 - (\theta_{t+1,i} - \theta_{,i}^*)^2] + \frac{\alpha}{2(1-\beta_1)} \cdot \frac{m_{t,i}^2}{\sqrt{tv_{t,i}}} \tag{12}$$

$$+ \frac{\beta_{1,t}}{2\alpha_t(1-\beta_{1,t})}(\theta_{,i}^* - \theta_{t,i})^2\sqrt{v_{t,i}} + \frac{\beta_1\alpha}{2(1-\beta_1)} \cdot \frac{m_{t-1,i}^2}{\sqrt{tv_{t,i}}}, \tag{13}$$

where (13) bounds the last term of (12) by Cauchy-Schwarz Inequality and plugs in the value of $\alpha_t$. Plugging (13) into (11) and summing from $t=1$ to $T$, we obtain

$$R(T) = \sum_{t=1}^T \sum_{i=1}^d g_{t,i}(\theta_{t,i} - \theta_{,i}^*)$$

$$\le \sum_{t=1}^T \sum_{i=1}^d \frac{\sqrt{v_{t,i}}}{2\alpha_t(1-\beta_1)} [(\theta_{t,i} - \theta_{,i}^*)^2 - (\theta_{t+1,i} - \theta_{,i}^*)^2] + \sum_{t=1}^T \sum_{i=1}^d \frac{\alpha}{2(1-\beta_1)} \cdot \frac{m_{t,i}^2}{\sqrt{tv_{t,i}}}$$

$$+ \sum_{t=1}^T \sum_{i=1}^d \frac{\beta_{1,t}}{2\alpha_t(1-\beta_{1,t})}(\theta_{,i}^* - \theta_{t,i})^2\sqrt{v_{t,i}} + \sum_{t=1}^T \sum_{i=1}^d \frac{\beta_1\alpha}{2(1-\beta_1)} \cdot \frac{m_{t-1,i}^2}{\sqrt{tv_{t,i}}}$$

$$\le \sum_{i=1}^d \frac{\sqrt{v_{1,i}}}{2\alpha_1(1-\beta_1)}(\theta_{1,i} - \theta_{,i}^*)^2 + \frac{1}{2(1-\beta_1)} \sum_{t=2}^T \sum_{i=1}^d (\theta_{t,i} - \theta_{,i}^*)^2 \left( \frac{\sqrt{v_{t,i}}}{\alpha_t} - \frac{\sqrt{v_{t-1,i}}}{\alpha_{t-1}} \right) \tag{14}$$

$$+ \sum_{t=1}^T \sum_{i=1}^d \frac{\beta_{1,t}}{2\alpha_t(1-\beta_1)}(\theta_{,i}^* - \theta_{t,i})^2\sqrt{v_{t,i}} + \sum_{t=1}^T \sum_{i=1}^d \frac{\alpha}{1-\beta_1} \cdot \frac{m_{t,i}^2}{\sqrt{tv_{t,i}}}, \tag{15}$$

where (15) rearranges the first term of (14). Finally utilizing the assumptions in Theorem 1, we get

$$
\begin{aligned}
R(T) \leq & \sum_{i=1}^{d} \frac{\sqrt{v_{1,i}}}{2\alpha_1(1-\beta_1)} D_\infty^2 + \frac{1}{2(1-\beta_1)} \sum_{t=2}^{T} \sum_{i=1}^{d} D_\infty^2 \left( \frac{\sqrt{v_{t,i}}}{\alpha_t} - \frac{\sqrt{v_{t-1,i}}}{\alpha_{t-1}} \right) \\
& + \frac{D_\infty^2}{2(1-\beta_1)} \sum_{t=1}^{T} \sum_{i=1}^{d} \frac{\beta_{1,t} v_{t,i}^{\frac{1}{2}}}{\alpha_t} + \sum_{i=1}^{d} \frac{\alpha\sqrt{1+\log T}}{(1-\beta_1)^3(1-\gamma)\sqrt{1-\beta_2}} \|g_{1:T,i}\|_2 \\
= & \sum_{i=1}^{d} \frac{\sqrt{v_{T,i}}}{2\alpha_T(1-\beta_1)} D_\infty^2 + \frac{D_\infty^2}{2(1-\beta_1)} \sum_{t=1}^{T} \sum_{i=1}^{d} \frac{\beta_{1,t} v_{t,i}^{\frac{1}{2}}}{\alpha_t} \\
& + \sum_{i=1}^{d} \frac{\alpha\sqrt{1+\log T}}{(1-\beta_1)^3(1-\gamma)\sqrt{1-\beta_2}} \|g_{1:T,i}\|_2 ,
\end{aligned}
\tag{16}
$$

which is our desired result. $\qquad\square$

### C.1.2 Proof of Corollary 1

*Proof.* Plugging $\alpha_t = \frac{\alpha}{\sqrt{t}}$ and $\beta_{1,t} = \beta_1\lambda^t$ into (16), we get

$$
\begin{aligned}
R(T) \leq & \frac{D_\infty^2\sqrt{T}}{2\alpha(1-\beta_1)} \sum_{i=1}^{d} \sqrt{v_{T,i}} + \frac{D_\infty^2}{2\alpha(1-\beta_1)} \sum_{t=1}^{T} \sum_{i=1}^{d} \beta_1\lambda^t \sqrt{tv_{t,i}} \\
& + \sum_{i=1}^{d} \frac{\alpha\sqrt{1+\log T}}{(1-\beta_1)^3(1-\gamma)\sqrt{1-\beta_2}} \|g_{1:T,i}\|_2 .
\end{aligned}
\tag{17}
$$

Next, we employ Mathematical Induction to prove that $v_t, i \leq G_\infty$ for any $0 \leq t \leq T, 1 \leq i \leq d$. $\forall i$, we have $m_{0,i}^2 = 0 \leq G_\infty^2$. Suppose $m_{t-1,i} \leq G_\infty$, we have

$$
\begin{aligned}
m_{t,i}^2 &= (\beta_{1,t} m_{t-1,i} + (1-\beta_{1,t})g_{t,i})^2 \\
&\overset{(i)}{\leq} \beta_{1,t} m_{t-1,i}^2 + (1-\beta_{1,t})g_{t,i}^2 \\
&\leq \beta_{1,t} G_\infty^2 + (1-\beta_{1,t})G_\infty^2 = G_\infty^2,
\end{aligned}
$$

where (i) comes from the convexity of function $f = x^2$. Hence by induction, we have $m_{t,i}^2 \leq G_\infty^2$ for all $0 \leq t \leq T$. Furthermore, $\forall i$, we have $v_{0,i} = 0 \leq G_\infty^2$. Suppose $v_{t-1,i} \leq G_\infty^2$, we have

$$
\begin{aligned}
v_{t,i} &= \beta_2 v_{t-1,i} + (1-\beta_2)m_{t,i}^2 \\
&\leq \beta_2 G_\infty^2 + (1-\beta_2)G_\infty^2 = G_\infty^2.
\end{aligned}
$$

Therefore, by induction, we have $v_{t,i} \leq G_\infty^2, \forall i, t$. Combining this with the fact that $\sum_{i=1}^{d} \|g_{1:T,i}\|_2 \leq dG_\infty\sqrt{T}$ and (17), we obtain

$$
R(T) \leq \frac{dG_\infty D_\infty^2\sqrt{T}}{2\alpha(1-\beta_1)} + \frac{dG_\infty D_\infty^2\beta_1}{2\alpha(1-\beta_1)} \sum_{t=1}^{T} \lambda^t\sqrt{t} + \frac{dG_\infty\alpha\sqrt{1+\log T}}{(1-\beta_1)^3(1-\gamma)\sqrt{(1-\beta_2)T}}.
\tag{18}
$$

For $\sum_{t=1}^{T} \lambda^t\sqrt{t}$, we apply arithmetic geometric series upper bound:

$$
\sum_{t=1}^{T} \lambda^t\sqrt{t} \leq \sum_{t=1}^{T} t\lambda^t \leq \frac{1}{(1-\lambda)^2}.
\tag{19}
$$

Plugging (19) into (18) and dividing both sides by $T$, we obtain

$$
\frac{R(T)}{T} \leq \frac{dG_\infty\alpha\sqrt{1+\log T}}{(1-\beta_1)^3(1-\gamma)\sqrt{(1-\beta_2)T}} + \frac{dD_\infty^2 G_\infty}{2\alpha(1-\beta_1)\sqrt{T}} + \frac{dD_\infty^2 G_\infty\beta_1}{2\alpha(1-\beta_1)(1-\lambda)^2 T},
$$

which concludes the proof. $\qquad\square$

### C.2 Proof of the convergence results for the non-convex case

### C.2.1 USEFUL THEOREM

**Theorem 5.** (Chen et al. (2019)) Suppose Assumptions 2 and 4 are satisfied, $\beta_{1,t}$ is chosen such that $0 \leq \beta_{1,t+1} \leq \beta_{1,t} < 1, 0 < \beta_2 < 1, \forall t > 0$. There exists some constant G such that $\left\| \alpha_t \cdot \frac{m_t}{\sqrt{v_t}} \right\|_2 \leq G, \forall t$. Then Adam-type algorithms yield

$$
\mathbb{E}\left[ \sum_{t=1}^T \alpha_t \left\langle \nabla f(\theta_t), \nabla f(\theta_t)/\sqrt{v_t} \right\rangle \right]
$$

$$
\leq \mathbb{E}\left[ C_1 \sum_{t=1}^T \left\| \alpha_t g_t/\sqrt{v_t} \right\|_2^2 + C_2 \sum_{t=1}^T \left\| \frac{\alpha_t}{\sqrt{v_t}} - \frac{\alpha_{t-1}}{\sqrt{v_{t-1}}} \right\|_1 + C_3 \sum_{t=1}^T \left\| \frac{\alpha_t}{\sqrt{v_t}} - \frac{\alpha_{t-1}}{\sqrt{v_{t-1}}} \right\|_2^2 \right] + C_4, \quad (20)
$$

where $C_1, C_2$ and $C_3$ are constants independent of $d$ and $T$, $C_4$ is a constant independent of $T$, the expectation is taken $w.r.t$ all randomness corresponding to $\{g_t\}$.

Furthermore, let $\gamma_t := \min_{j \in [d]} \min_{\{g_i\}_{i=1}^t} \frac{\alpha_i}{\sqrt{v_{i,j}}}$ denotes the minimum possible value of effective stepsize at time $t$ over all possible coordinate and past gradients $\{g_i\}_{i=1}^t$. The convergence rate of Adam-type algorithm is given by

$$
\min_{t \in [T]} \mathbb{E}\left[ \| \nabla f(\theta_t) \|_2^2 \right] = O\left( \frac{s_1(T)}{s_2(T)} \right),
$$

where $s_1(T)$ is defined through the upper bound of RHS of (20), and $\sum_{t=1}^T \gamma_t = \Omega(s_2(T))$.

We present the proof of this Theorem in subsection C.2.4 and C.2.5 for completeness and reader's convenience.

### C.2.2 PROOF OF THEOREM 2

*Proof.* We will first bound each term on RHS of Equation (20). Given all conditions in Theorem 2 hold, we have that

$$
\mathbb{E}\left[ \sum_{t=1}^T \left\| \alpha_t \frac{g_t}{\sqrt{v_t}} \right\|_2^2 \right] \overset{(i)}{\leq} \frac{1}{c} \mathbb{E}\left[ \sum_{t=1}^T \| \alpha_t g_t \|_2^2 \right]
$$

$$
\leq \frac{1}{c} \mathbb{E}\left[ \sum_{t=1}^T \alpha_t^2 \| g_t \|_2^2 \right]
$$

$$
\overset{(ii)}{\leq} \frac{H^2}{c} \sum_{t=1}^T \alpha_t^2, \quad (21)
$$

where (i) arises from the fact that $0 < c \leq v_t, \forall t \in [T]$ and inequality (ii) is based on $\| g_t \|_2 \leq H, \forall t \in [T]$. Then,

$$
\mathbb{E}\left[ \sum_{t=1}^T \left\| \frac{\alpha_t}{\sqrt{v_t}} - \frac{\alpha_{t-1}}{\sqrt{v_{t-1}}} \right\|_1 \right] \overset{(i)}{=} \mathbb{E}\left[ \sum_{i=1}^d \sum_{t=1}^T \frac{\alpha_{t-1}}{\sqrt{v_{t-1,i}}} - \frac{\alpha_t}{\sqrt{v_{t,i}}} \right]
$$

$$
= \mathbb{E}\left[ \sum_{i=1}^d \frac{\alpha_1}{\sqrt{v_{1,i}}} - \frac{\alpha_T}{\sqrt{v_{T,i}}} \right]
$$

$$
\leq \mathbb{E}\left[ \sum_{i=1}^d \frac{\alpha_1}{\sqrt{v_{1,i}}} \right]
$$

$$
\overset{(ii)}{\leq} \frac{d\alpha}{\sqrt{c}}, \quad (22)
$$

and here (i) holds since we assume that $\frac{\alpha_t}{\sqrt{v_t}} \geq \frac{\alpha_{t+1}}{\sqrt{v_{t+1}}}, \forall t \in [T]$, and (ii) comes from the fact that $0 < c \leq v_t, 0 < \alpha_t \leq \alpha, \forall t \in [T]$. Next,

$$
\mathbb{E}\left[ \sum_{t=1}^T \left\| \frac{\alpha_t}{\sqrt{v_t}} - \frac{\alpha_{t-1}}{\sqrt{v_{t-1}}} \right\|_2^2 \right] = \mathbb{E}\left[ \sum_{t=1}^T \sum_{i=1}^d \left\| \frac{\alpha_t}{\sqrt{v_{t,i}}} - \frac{\alpha_{t-1}}{\sqrt{v_{t-1,i}}} \right\|_2^2 \right]
$$

$$
\overset{(i)}{\leq} \mathbb{E}\left[ \sum_{t=1}^T \sum_{i=1}^d \left\| \frac{\alpha_t}{\sqrt{v_{t,i}}} - \frac{\alpha_{t-1}}{\sqrt{v_{t-1,i}}} \right\|_2 \frac{\alpha}{\sqrt{c}} \right]
$$

$$
\leq \frac{d\alpha^2}{c}, \quad (23)
$$

where we have (i) because $\left\|\frac{\alpha_t}{\sqrt{v_{t,i}}} - \frac{\alpha_{t-1}}{\sqrt{v_{t-1,i}}}\right\|_2 = \frac{\alpha_{t-1}}{\sqrt{v_{t-1,i}}} - \frac{\alpha_t}{\sqrt{v_{t,i}}} \leq \frac{\alpha}{\sqrt{c}}$. On the other hand, we can obtain a lower bound of the LHS of Equation (20):

$$\mathbb{E}\left[\sum_{t=1}^{T} \alpha_t \left\langle \nabla f(\theta_t), \frac{\nabla f(\theta_t)}{\sqrt{v_t}} \right\rangle\right] \geq \frac{1}{H}\mathbb{E}\left[\sum_{t=1}^{T} \alpha_t \|\nabla f(\theta_t)\|_2^2\right] \geq \frac{T\alpha_T}{H}\min_{t\in[T]}\mathbb{E}\|\nabla f(\theta_t)\|_2^2, \quad (24)$$

where the last equality comes from the fact that $v_t$ is weighted average of $m_t^2$ and $\|m_t\|_2 \leq H$ since $m_t$ is also an exponential moving average of $g_t$.

By combining the results in (21), (22), (23) and (24) to (20), we obatain

$$\frac{T\alpha_T}{H}\min_{t\in[T]}\mathbb{E}\|\nabla f(\theta_t)\|_2^2$$

$$\leq \mathbb{E}\left[\sum_{t=1}^{T}\alpha_t\left\langle\nabla f(\theta_t), \frac{\nabla f(\theta_t)}{\sqrt{v_t}}\right\rangle\right]$$

$$\leq \mathbb{E}\left[C_1\sum_{t=1}^{T}\left\|\alpha_t\frac{g_t}{\sqrt{v_t}}\right\|_2^2 + C_2\sum_{t=1}^{T}\left\|\frac{\alpha_t}{\sqrt{v_t}} - \frac{\alpha_{t-1}}{\sqrt{v_{t-1}}}\right\|_1 + C_3\sum_{t=1}^{T}\left\|\frac{\alpha_t}{\sqrt{v_t}} - \frac{\alpha_{t-1}}{\sqrt{v_{t-1}}}\right\|_2^2\right] + C_4$$

$$\leq \frac{C_1 H^2}{c}\sum_{t=1}^{T}\alpha_t^2 + \frac{C_2 d\alpha}{\sqrt{c}} + \frac{C_3 d\alpha^2}{c} + C_4 = \frac{C_1 H^2 \eta(T)}{c} + C_2\frac{d\alpha}{\sqrt{c}} + C_3\frac{d\alpha^2}{c} + C_4.$$

After rearrangement, we can easily deduce that

$$\min_{t\in[T]}\mathbb{E}\|\nabla f(\theta_t)\|_2^2 \leq \frac{H}{T\alpha_T}\left[\frac{C_1 H^2 \eta(T)}{c} + C_2\frac{d\alpha}{\sqrt{c}} + C_3\frac{d\alpha^2}{c} + C_4\right]$$

$$= \frac{1}{T\alpha_T}(Q_1 + Q_2\eta(T)), \quad (25)$$

where

$$Q_1 = H\left(C_2\frac{d\alpha}{\sqrt{c}} + C_3\frac{d\alpha^2}{c} + C_4\right), \quad Q_2 = \frac{C_1 H^3}{c}.$$

$\square$

### C.2.3 PROOF OF COROLLARY 2

By choosing $\alpha_t = \alpha/\sqrt{t} \leq \alpha, \forall t \in [T]$, we have

$$T\alpha_T = \alpha\sqrt{T}, \quad \eta(T) = \sum_{t=1}^{T}\alpha_t^2 = \alpha^2\sum_{t=1}^{T}\frac{1}{t} \leq \alpha^2(1 + \log(T)).$$

Combining this with (25) and making some rearrangement, we have:

$$\min_{t\in[T]}\mathbb{E}\|\nabla f(\theta_t)\|_2^2 \leq \frac{H}{\alpha\sqrt{T}}\left[C_1\frac{\alpha^2 H^2(1 + \log(T))}{c} + C_2\frac{d\alpha}{\sqrt{c}} + C_3\frac{d\alpha^2}{c} + C_4\right]$$

$$= \frac{1}{\sqrt{T}}(Q_1^* + Q_2^*\log(T)).$$

where

$$Q_1^* = H\left(C_1\frac{H^2\alpha}{c} + C_2\frac{d\alpha}{\sqrt{c}} + C_3\frac{d\alpha^2}{c} + C_4\right), \quad Q_2^* = \frac{C_1 H^3 \alpha}{c}.$$

$\square$

### C.2.4 TECHNICAL LEMMAS FOR PROOF OF THEOREM 5

In this section, we introduce seven useful lemmas.

**Lemma 5** (Chen et al. (2019)). Let $\theta_0 \triangleq \theta_1$ in the Algorithm, consider the sequence

$$z_t = \theta_t + \frac{\beta_{1,t}}{1 - \beta_{1,t}}(\theta_t - \theta_{t-1}), \forall t \geq 2.$$

The following holds true:

$$z_{t+1} - z_t = -\left(\frac{\beta_{1,t+1}}{1 - \beta_{1,t+1}} - \frac{\beta_{1,t}}{1 - \beta_{1,t}}\right)\frac{\alpha_t m_t}{\sqrt{v_t}}$$
$$-\frac{\beta_{1,t}}{1 - \beta_{1,t}}\left(\frac{\alpha_t}{\sqrt{v_t}} - \frac{\alpha_{t-1}}{\sqrt{v_{t-1}}}\right)m_{t-1} - \frac{\alpha_t g_t}{\sqrt{v_t}}, \forall t > 1,$$

and

$$z_2 - z_1 = -\left(\frac{\beta_{1,2}}{1 - \beta_{1,2}} - \frac{\beta_{1,1}}{1 - \beta_{1,1}}\right)\frac{\alpha_1 m_1}{\sqrt{v_1}} - \frac{\alpha_1 g_1}{\sqrt{v_1}}.$$

**Lemma 6** (Chen et al. (2019)). Suppose that the conditions in Theorem 5 hold, then we have

$$\mathbb{E}\left[f(z_{t+1} - f(z_t))\right] \le \sum_{i=1}^6 T_i,$$

where

$$T_1 = -\mathbb{E}\left[\sum_{i=1}^t \left\langle \nabla f(z_i), \frac{\beta_{1,i}}{1 - \beta_{1,i}}\left(\frac{\alpha_i}{\sqrt{v_i}} - \frac{\alpha_{i-1}}{\sqrt{v_{i-1}}}\right)m_{i-1}\right\rangle\right], \tag{26}$$

$$T_2 = -\mathbb{E}\left[\sum_{i=1}^t \alpha_i \left\langle \nabla f(z_i), \frac{g_i}{\sqrt{v_i}}\right\rangle\right], \tag{27}$$

$$T_3 = -\mathbb{E}\left[\sum_{i=1}^t \left\langle \nabla f(z_i), \left(\frac{\beta_{1,i+1}}{1 - \beta_{1,i+1}} - \frac{\beta_i}{1 - \beta_i}\right)\frac{\alpha_i m_i}{\sqrt{v_i}}\right\rangle\right], \tag{28}$$

$$T_4 = \mathbb{E}\left[\sum_{i=1}^t \frac{3L}{2}\left\|\left(\frac{\beta_{1,i+1}}{1 - \beta_{1,i+1}} - \frac{\beta_{1,i}}{1 - \beta_{1,i}}\right)\frac{\alpha_i m_i}{\sqrt{v_i}}\right\|_2^2\right], \tag{29}$$

$$T_5 = \mathbb{E}\left[\sum_{i=1}^t \frac{3L}{2}\left\|\frac{\beta_{1,i}}{1 - \beta_{1,i}}\left(\frac{\alpha_i}{\sqrt{v_i}} - \frac{\alpha_{i-1}}{\sqrt{v_{i-1}}}\right)m_{i-1}\right\|_2^2\right], \tag{30}$$

$$T_6 = \mathbb{E}\left[\sum_{i=1}^t \frac{3L}{2}\left\|\frac{\alpha_i g_i}{\sqrt{v_i}}\right\|_2^2\right]. \tag{31}$$

**Lemma 7** (Chen et al. (2019)). Suppose that the condition in Theorem 5 hold, then for $T_1$ in (26) it holds that

$$T_1 = -\mathbb{E}\left[\sum_{i=1}^t \left\langle \nabla f(z_i), \frac{\beta_{1,i}}{1 - \beta_{1,i}}\left(\frac{\alpha_i}{\sqrt{v_i}} - \frac{\alpha_{i-1}}{\sqrt{v_{i-1}}}\right)m_{i-1}\right\rangle\right]$$
$$\le H^2 \frac{\beta_1}{1 - \beta_1}\mathbb{E}\left[\sum_{i=2}^t \sum_{j=1}^d \left|\left(\frac{\alpha_i}{\sqrt{v_i}} - \frac{\alpha_{i-1}}{\sqrt{v_{i-1}}}\right)_j\right|\right].$$

**Lemma 8** (Chen et al. (2019)). Suppose the conditions in Theorem 5 are satisfied, then $T_3$ in (28) can be bounded as:

$$T_3 = -\mathbb{E}\left[\sum_{i=1}^t \left\langle \nabla f(z_i), \left(\frac{\beta_{1,i+1}}{1 - \beta_{1,i+1}} - \frac{\beta_i}{1 - \beta_i}\right)\frac{\alpha_i m_i}{\sqrt{v_i}}\right\rangle\right]$$
$$\le \left(\frac{\beta_1}{1 - \beta_1} - \frac{\beta_{1,t+1}}{1 - \beta_{1,t+1}}\right)(H^2 + G^2).$$

**Lemma 9** (Chen et al. (2019)). Suppose assumptions in Theorem 5 are satisfied, then for $T_4$ in (29), it holds that

$$T_4 = \mathbb{E}\left[\sum_{i=1}^t \frac{3L}{2}\left\|\left(\frac{\beta_{1,i+1}}{1 - \beta_{1,i+1}} - \frac{\beta_{1,i}}{1 - \beta_{1,i}}\right)\frac{\alpha_i m_i}{\sqrt{v_i}}\right\|_2^2\right]$$
$$\le \frac{3L}{2}\left(\frac{\beta_1}{1 - \beta_1} - \frac{\beta_{1,t+1}}{1 - \beta_{1,t+1}}\right)^2 G^2.$$

**Lemma 10** (Chen et al. (2019)). Suppose the assumptions in Theorem 5 are satisfied, then for $T_5$ in (30), we have

$$T_5 = \mathbb{E}\left[\sum_{i=1}^t \frac{3L}{2}\left\|\frac{\beta_{1,i}}{1 - \beta_{1,i}}\left(\frac{\alpha_i}{\sqrt{v_i}} - \frac{\alpha_{i-1}}{\sqrt{v_{i-1}}}\right)m_{i-1}\right\|_2^2\right]$$
$$\le \frac{3L}{2}\left(\frac{\beta_1}{1 - \beta_1}\right)^2 H^2\mathbb{E}\left[\sum_{i=2}^t \sum_{j=1}^d \left(\frac{\alpha_i}{\sqrt{v_i}} - \frac{\alpha_{i-1}}{\sqrt{v_{i-1}}}\right)_j^2\right].$$

**Lemma 11** (Chen et al. (2019)). *Suppose the assumptions in Theorem 5 are satisfied, then $T_2$ in (27) can be bounded as:*

$$
\begin{aligned}
T_2 &= -\mathbb{E}\left[\sum_{i=1}^{t} \alpha_i \left\langle \nabla f(z_i), \frac{g_i}{\sqrt{v_i}}\right\rangle\right] \\
&\leq \mathbb{E}\sum_{i=2}^{t} \frac{1}{2}\left\|\frac{\alpha_i g_i}{\sqrt{v_i}}\right\|_2^2 + L^2\left(\frac{\beta_1}{1-\beta_1}\right)^2\left(\frac{1}{1-\beta_1}\right)^2 \mathbb{E}\left[\sum_{j=1}^{d}\sum_{i=2}^{t-1}\left(\frac{\alpha_i g_i}{\sqrt{v_i}}\right)_j^2\right] \\
&\quad + L^2 H^2\left(\frac{\beta_1}{1-\beta_1}\right)^4\left(\frac{1}{1-\beta_1}\right)^2 \mathbb{E}\left[\sum_{j=1}^{d}\sum_{i=2}^{t-1}\left(\frac{\alpha_i}{\sqrt{v_i}} - \frac{\alpha_{i-1}}{\sqrt{v_{i-1}}}\right)_j^2\right] \\
&\quad + 2H^2 \mathbb{E}\left[\sum_{j=1}^{d}\sum_{i=2}^{t}\left|\left(\frac{\alpha_i}{\sqrt{v_i}} - \frac{\alpha_{i-1}}{\sqrt{v_{i-1}}}\right)_j\right|\right] \\
&\quad + 2H^2 \mathbb{E}\left[\sum_{j=1}^{d}\left(\frac{\alpha_1}{\sqrt{v_1}}\right)_j\right] - \mathbb{E}\left[\sum_{i=1}^{t}\alpha_i \left\langle \nabla f(x_i), \nabla f(x_i)/\sqrt{v_i}\right\rangle\right].
\end{aligned}
$$

### C.2.5 PROOF OF THEOREM 5

*Proof.* We can prove Theorem 5 after combining Lemma 5, 6, 7, 8, 9, 10 and 11. Specifically, firstly based on Lemma 5 it holds that

$$
\begin{aligned}
&\mathbb{E}\left[f(z_{t+1}) - f(z_1)\right] \\
&\leq \sum_{i=1}^{6} T_i \\
&= \mathbb{E}\left[\sum_{i=1}^{t}\frac{3}{2}L\left\|\left(\frac{\beta_{1,t+1}}{1-\beta_{1,t+1}} - \frac{\beta_{1,t}}{1-\beta_{1,t}}\right)\alpha_t m_t/\sqrt{v_t}\right\|_2^2\right] + E\left[\sum_{i=1}^{t}\frac{3}{2}L\left\|\alpha_i g_i/\sqrt{v_i}\right\|_2^2\right] \\
&\quad + \mathbb{E}\left[\sum_{i=1}^{t}\frac{3}{2}L\left\|\frac{\beta_{1,i}}{1-\beta_{1,i}}\left(\frac{\alpha_t}{\sqrt{v_i}} - \frac{\alpha_{i-1}}{\sqrt{v_{i-1}}}\right)\odot m_{i-1}\right\|_2^2\right] \\
&\quad - \mathbb{E}\left[\sum_{i=1}^{t}\left\langle\nabla f(z_i), \frac{\beta_{1,i}}{1-\beta_{1,i}}\left(\frac{\alpha_i}{\sqrt{v_i}} - \frac{\alpha_{i-1}}{\sqrt{v_{i-1}}}\right)\odot m_{i-1}\right\rangle\right] - E\left[\sum_{i=1}^{t}\alpha_i\left\langle\nabla f(z_i), g_i/\sqrt{v_i}\right\rangle\right] \\
&\quad - \mathbb{E}\left[\sum_{i=1}^{t}\left\langle\nabla f(z_i), \left(\frac{\beta_{1,i+1}}{1-\beta_{1,i+1}} - \frac{\beta_{1,i}}{1-\beta_{1,i}}\right)\alpha_i m_i/\sqrt{v_i}\right\rangle\right].
\end{aligned}
$$

Then we can combine Lemma 6, 7, 8, 9, 10 and 11 and further merge similar terms. We have that

$$\mathbb{E}\left[f(z_{t+1}) - f(z_t)\right] \leq \sum_{i=1}^{6} T_i$$

$$\leq H^2 \frac{\beta_1}{1-\beta_1} \mathbb{E}\left[\sum_{i=2}^{t}\sum_{j=1}^{d}\left|\left(\frac{\alpha_i}{\sqrt{v_i}} - \frac{\alpha_{i-1}}{\sqrt{v_{i-1}}}\right)_j\right|\right]$$

$$+ \left(\frac{\beta_1}{1-\beta_1} - \frac{\beta_{1,t+1}}{1-\beta_{1,t+1}}\right)(H^2+G^2) + \frac{3L}{2}\left(\frac{\beta_1}{1-\beta_1} - \frac{\beta_{1,t}}{1-\beta_{1,t}}\right)^2 G^2$$

$$+ \frac{3L}{2}\left(\frac{\beta_1}{1-\beta_1}\right)^2 H^2 \mathbb{E}\left[\sum_{i=2}^{t}\sum_{j=1}^{d}\left(\frac{\alpha_i}{\sqrt{v_i}} - \frac{\alpha_{i-1}}{\sqrt{v_{i-1}}}\right)_j^2\right]$$

$$+ \mathbb{E}\sum_{i=2}^{t}\frac{1}{2}\left\|\frac{\alpha_i g_i}{\sqrt{v_i}}\right\|^2 + L^2\left(\frac{\beta_1}{1-\beta_1}\right)^2\left(\frac{1}{1-\beta_1}\right)^2 \mathbb{E}\left[\sum_{j=1}^{d}\sum_{i=2}^{t-1}\left(\frac{\alpha_i g_i}{\sqrt{v_i}}\right)_j^2\right]$$

$$+ L^2 H^2\left(\frac{\beta_1}{1-\beta_1}\right)^4\left(\frac{1}{1-\beta_1}\right)^2 \mathbb{E}\left[\sum_{j=1}^{d}\sum_{i=2}^{t-1}\left(\frac{\alpha_i}{\sqrt{v_i}} - \frac{\alpha_{i-1}}{\sqrt{v_{i-1}}}\right)_j^2\right]$$

$$+ 2H^2 \mathbb{E}\left[\sum_{j=1}^{d}\sum_{i=2}^{t}\left|\left(\frac{\alpha_i}{\sqrt{v_i}} - \frac{\alpha_{i-1}}{\sqrt{v_{i-1}}}\right)_j\right|\right] + 2H^2 \mathbb{E}\left[\sum_{j=1}^{d}\left(\frac{\alpha_1}{\sqrt{v_1}}\right)_j\right]$$

$$- \mathbb{E}\left[\sum_{i=1}^{t}\alpha_i\left\langle\nabla f(x_i), \nabla f(x_i)/\sqrt{v_i}\right\rangle\right].$$

Finally, after rearrangement and some calculation, it can be verified that

$$\mathbb{E}\left[\sum_{i=1}^{t}\alpha_i\left\langle\nabla f(x_i), \nabla f(x_i)/\sqrt{v_i}\right\rangle\right]$$

$$\leq \mathbb{E}\left[C_1\sum_{t=1}^{T}\|\alpha_t g_t/\sqrt{v_t}\|_2^2 + C_2\sum_{t=1}^{T}\left\|\frac{\alpha_t}{\sqrt{v_t}} - \frac{\alpha_{t-1}}{\sqrt{v_{t-1}}}\right\|_1\right.$$

$$\left. + C_3\sum_{t=1}^{T}\left\|\frac{\alpha_t}{\sqrt{v_t}} - \frac{\alpha_{t-1}}{\sqrt{v_{t-1}}}\right\|_2^2\right] + C_4,$$

where we have

$$C_1 \triangleq \frac{3}{2}L + \frac{1}{2} + L^2\frac{\beta_1}{1-\beta_1}\left(\frac{1}{1-\beta_1}\right)^2,$$

$$C_2 \triangleq H^2\frac{\beta_1}{1-\beta_1} + 2H^2,$$

$$C_3 \triangleq \left[1 + L^2\left(\frac{1}{1-\beta_1}\right)^2\left(\frac{\beta_1}{1-\beta_1}\right)\right]H^2\left(\frac{\beta_1}{1-\beta_1}\right)^2,$$

$$C_4 \triangleq \left(\frac{\beta_1}{1-\beta_1}\right)(H^2+G^2) + \left(\frac{\beta_1}{1-\beta_1}\right)^2 G^2 + 2H^2\mathbb{E}\left[\|\alpha_1/\sqrt{v_1}\|_1\right] + \mathbb{E}[f(z_1) - f(z^*)],$$

and $z^*$ is an optimal where $f(\cdot)$ takes its minimum. This result directly implies Theorem 5. □

## D  ADDITIONAL EXPERIMENTAL DETAILS

**Hyperparameter tuning rule**  For hyperparameter tuning, we perform extensive and careful grid search to choose the best hyperparameters for all the baseline algorithms.

For SGDM, we set the momentum parameter as 0.9 and search the optimal learning rate $\alpha$ among set $\{30.0, 1.0, 0.1, 0.01, 0.0015, 0.001, 0.0005, 0.0002\}$ for all the experiments.

For adaptive gradient method baselines (Adam, AdamW, Yogi, Adabound, RAdam and AdaBelief), we search for $\beta_1$ among $\{0.5, 0.6, 0.7, 0.8, 0.9\}$, $\beta_2$ among $\{0.9, 0.98, 0.99, 0.999, 0.9999\}$, $\alpha$ among $\{0.1, 0.01, 0.0015, 0.001, 0.0005\}$, weight decay parameter among $\{1.2 \times 10^{-6}, 10^{-4}, 5 \times 10^{-4}\}$ and $\epsilon$ among $\{10^{-8}, 10^{-12}, 10^{-16}\}$ for all the experiments except ImageNet. All the additional method-specific parameters

are carefully chosen according to the original paper setting. Due to extremely large computing workload on ImageNet dataset, we set $\beta_1 = 0.9, \beta_2 = 0.999, \alpha = 0.001$ and $\epsilon = 1 \times 10^{-8}$ for all the adaptive gradient methods, and finetune weight decay parameter from set $\{0, 1 \times 10^{-4}, 5 \times 10^{-4}, 10^{-2}, 5 \times 10^{-2}\}$ for the values we reported as we run.

For our AdaMomentum, we employ the default parameters as $\beta_1 = 0.9$ and choose learning rate and weight decay parameter using same parameter searching scheme as the adaptive gradient baseline methods. $\beta_2$ is set as 0.999 for all the tasks. We Choose $\epsilon = 10^{-8}$ for image classification tasks and $\epsilon = 1 \times 10^{-16}$ for other tasks. We find this setting is universally ample for satisfactory performance, which further demonstrates the superiority of AdaMomentum that little tuning effort is needed.

All the experiments reported are performed on NVIDIA GeForce RTX 2080Ti GPUs with Intel Core i7-8700K 3.70GHz CPUs. We provide some additional information concerning the empirical experiments for completeness.

## D.1 IMAGE CLASSIFICATION

Table 7: Well tuned hyperparameter configuration of the adaptive gradient methods for CNNs on CIFAR-10.

| Algorithm | Adam | AdamW | Yogi | AdaBound | RAdam | AdaBelief | AdaMomentum |
|---|---|---|---|---|---|---|---|
| Stepsize $\alpha$ | 0.001 | 0.001 | 0.001 | 0.001 | 0.001 | 0.001 | 0.001 |
| $\beta_1$ | 0.9 | 0.9 | 0.9 | 0.9 | 0.9 | 0.9 | 0.9 |
| $\beta_2$ | 0.999 | 0.999 | 0.999 | 0.999 | 0.999 | 0.999 | 0.999 |
| Weight decay | $5 \times 10^{-4}$ | $5 \times 10^{-4}$ | $5 \times 10^{-4}$ | $5 \times 10^{-4}$ | $5 \times 10^{-4}$ | $5 \times 10^{-4}$ | $5 \times 10^{-4}$ |
| $\epsilon$ | $10^{-8}$ | $10^{-8}$ | $10^{-8}$ | $10^{-8}$ | $10^{-8}$ | $10^{-8}$ | $10^{-8}$ |

**CIFAR datasets** The values of the hyperparameters after careful tuning of the reported results of the adaptive gradient methods on CIFAR-10 in the main paper is summarized in Table 7. For SGDM, the optimal hyperparameter setting is: the learning rate is 0.1, the momentum parameter is 0.9, the weight decay parameter is $5 \times 10^{-4}$. For Adabound, the final learning rate is set as 0.1 (matching SGDM) and the value of the hyperparameter gamma is $10^{-3}$.

Table 8: Well tuned hyperparameter configuration of the adaptive gradient methods for CNNs on ImageNet.

| Algorithm | Adam | AdamW | Yogi | AdaBound | RAdam | AdaBelief | AdaMomentum |
|---|---|---|---|---|---|---|---|
| Stepsize $\alpha$ | 0.001 | 0.001 | 0.001 | 0.001 | 0.001 | 0.001 | 0.001 |
| $\beta_1$ | 0.9 | 0.9 | 0.9 | 0.9 | 0.9 | 0.9 | 0.9 |
| $\beta_2$ | 0.999 | 0.999 | 0.999 | 0.999 | 0.999 | 0.999 | 0.999 |
| Weight decay | $1 \times 10^{-4}$ | $1 \times 10^{-4}$ | $1 \times 10^{-4}$ | $1 \times 10^{-4}$ | $1 \times 10^{-4}$ | $1 \times 10^{-2}$ | $5 \times 10^{-2}$ |
| $\epsilon$ | $10^{-8}$ | $10^{-8}$ | $10^{-8}$ | $10^{-8}$ | $10^{-8}$ | $10^{-8}$ | $10^{-8}$ |

**ImageNet** The values of the hyperparameters after careful tuning of the reported results of the adaptive gradient methods on CIFAR-10 in the main paper is summarized in Table 8. For SGDM, the stepsize is 0.1, the momentum parameter is 0.9 and the weight decay is $5 \times 10^{-4}$.

## D.2 LSTM ON LANGUAGE MODELING

Table 9: Well tuned hyperparameter configuration of adaptive gradient methods for 1-layer-LSTM on Penn Treebank dataset.

| Algorithm | Adam | AdamW | Yogi | AdaBound | RAdam | AdaBelief | AdaMomentum |
|---|---|---|---|---|---|---|---|
| Stepsize $\alpha$ | 0.001 | 0.001 | 0.01 | 0.01 | 0.001 | 0.001 | 0.001 |
| $\beta_1$ | 0.9 | 0.9 | 0.9 | 0.9 | 0.9 | 0.9 | 0.9 |
| $\beta_2$ | 0.999 | 0.999 | 0.999 | 0.999 | 0.999 | 0.999 | 0.999 |
| Weight decay | $1.2 \times 10^{-4}$ | $1.2 \times 10^{-4}$ | $1.2 \times 10^{-4}$ | $1.2 \times 10^{-4}$ | $1.2 \times 10^{-4}$ | $1.2 \times 10^{-4}$ | $1.2 \times 10^{-4}$ |
| $\epsilon$ | $10^{-12}$ | $10^{-12}$ | $10^{-8}$ | $10^{-8}$ | $10^{-12}$ | $10^{-16}$ | $10^{-16}$ |

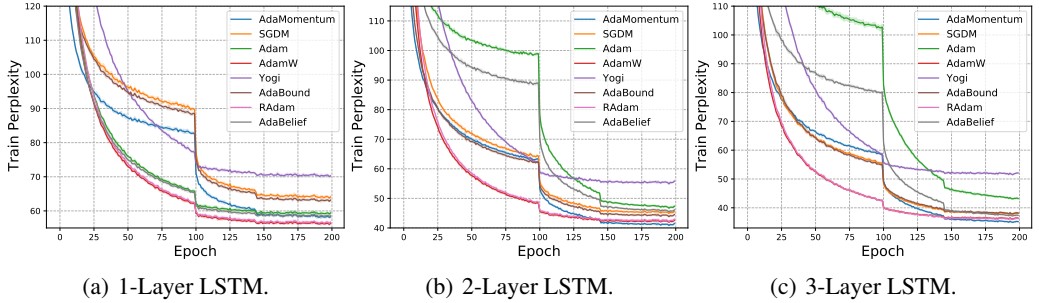

(a) 1-Layer LSTM.    (b) 2-Layer LSTM.    (c) 3-Layer LSTM.

Figure 5: Train perplexity curve on Penn Treebank dataset.

Table 10: Well tuned hyperparameter configuration of adaptive gradient methods for 2-layer-LSTM on Penn Treebank dataset.

| Algorithm | Adam | AdamW | Yogi | AdaBound | RAdam | AdaBelief | AdaMomentum |
|---|---|---|---|---|---|---|---|
| Stepsize $\alpha$ | 0.01 | 0.001 | 0.01 | 0.01 | 0.001 | 0.01 | 0.001 |
| $\beta_1$ | 0.9 | 0.9 | 0.9 | 0.9 | 0.9 | 0.9 | 0.9 |
| $\beta_2$ | 0.999 | 0.999 | 0.999 | 0.999 | 0.999 | 0.999 | 0.999 |
| Weight decay | $1.2 \times 10^{-4}$ | $1.2 \times 10^{-4}$ | $1.2 \times 10^{-4}$ | $1.2 \times 10^{-4}$ | $1.2 \times 10^{-4}$ | $1.2 \times 10^{-4}$ | $1.2 \times 10^{-4}$ |
| $\epsilon$ | $10^{-12}$ | $10^{-12}$ | $10^{-8}$ | $10^{-8}$ | $10^{-12}$ | $10^{-12}$ | $10^{-16}$ |

Table 11: Well tuned hyperparameter configuration of adaptive gradient methods for 3-layer-LSTM on Penn Treebank dataset.

| Algorithm | Adam | AdamW | Yogi | AdaBound | RAdam | AdaBelief | AdaMomentum |
|---|---|---|---|---|---|---|---|
| Stepsize $\alpha$ | 0.01 | 0.001 | 0.01 | 0.01 | 0.001 | 0.01 | 0.001 |
| $\beta_1$ | 0.9 | 0.9 | 0.9 | 0.9 | 0.9 | 0.9 | 0.9 |
| $\beta_2$ | 0.999 | 0.999 | 0.999 | 0.999 | 0.999 | 0.999 | 0.999 |
| Weight decay | $1.2 \times 10^{-4}$ | $1.2 \times 10^{-4}$ | $1.2 \times 10^{-4}$ | $1.2 \times 10^{-4}$ | $1.2 \times 10^{-4}$ | $1.2 \times 10^{-4}$ | $1.2 \times 10^{-4}$ |
| $\epsilon$ | $10^{-12}$ | $10^{-12}$ | $10^{-8}$ | $10^{-8}$ | $10^{-12}$ | $10^{-12}$ | $10^{-16}$ |

The training perplexity curve is illustrated in Figure 5. We can clearly see that AdaMomentum is able to make the perplexity descent faster than SGDM and most other adaptive gradient methods. In experimental settings, the size of the word embeddings is 400 and the number of hidden units per layer is 1150. We employ dropout in training and the dropout rate for RNN layers is 0.25 and the dropout rate for input embedding layers is 0.4.

The optimal hyperparameters of adaptive gradient methods for 1-layer, 2-layer and 3-layer LSTM are listed in Tables 9, 10 and 11 respectively. For SGDM, the Well tuned stepsize is 30.0 and the momentum parameter is 0.9. For Adabound, the final learning rate is set as 30.0 (matching SGDM) and the value of the hyperparameter gamma is $10^{-3}$.

## D.3 TRANSFORMER ON NEURAL MACHINE TRANSLATION

Table 12: Well tuned hyperparameter configuration of adaptive gradient methods for transformer on IWSTL'14 DE-EN dataset.

| Algorithm | Adam | AdamW | AdaBelief | AdaMomentum |
|---|---|---|---|---|
| Stepsize $\alpha$ | 0.0015 | 0.0015 | 0.0015 | 0.0005 |
| $\beta_1$ | 0.9 | 0.9 | 0.9 | 0.9 |
| $\beta_2$ | 0.98 | 0.98 | 0.999 | 0.999 |
| Weight decay | $10^{-4}$ | $10^{-4}$ | $10^{-4}$ | $10^{-4}$ |
| $\epsilon$ | $10^{-8}$ | $10^{-8}$ | $10^{-16}$ | $10^{-16}$ |

For transformer on NMT task, the well tuned hyperparameter values are summarized in Table 12. The stepsize of SGDM is 0.1 and the momentum parameter of SGDM is 0.9. Initial learning rate is $10^{-7}$ and the minimum learning rate threshold is set as $10^{-9}$ in the warm-up process for all the optimizers.

