# OpenReview forum: "Adapting Stepsizes by Momentumized Gradients Improves Optimization and Generalization"
_ICLR.cc/2022/Conference — ICLR 2022 Submitted_

### Official Review · Reviewer_5ZsD · 2021-11-01

**Correctness:** 3
**Technical Novelty And Significance:** 3
**Empirical Novelty And Significance:** 3
**Recommendation:** 5
**Confidence:** 3

**Main Review:**

## Strengths
The paper is in general well-organized. The technical contribution seems to be reasonable. I do not carefully check the proofs in the appendix. The empirical evaluations show that the proposed method is robust.

## Weaknesses

### 1.  The presentation in Sec 3.1 can be further improved.
It is more clear if the authors can construct a 1D example like Figure 1 and plot the trajectory of Adam and the proposed method. Readers may not fully understand the statement made in the section Benefits of substiuting g with m.

### 2. Sec 4
The convergence analysis looks quite standard. The authors should tell readers if the analysis gives a better bound compared to Adam or other variants.

### 3. General non-convex optimization (related to Sec 5.1.1)
Adam and many diagonal adaptive-gradient methods do not work well for general non-convex functions.
I wonder whether the proposed method has the same issue. For example, Ackley function and  Rosenbrock function are multivariate (d>1) objective functions. In Sec 5.1.1., the authors only consider the simplest case when d=2. I wonder about the performance of the proposed method and Adam for higher-dimensional (e.g., d=100) Ackley functions and Rosenbrock functions.






**Summary Of The Paper:**

In this work, the authors proposed an improved version of Adam by using a momentumized version of the second moment.
The authors justify the proposed modification and give some standard convergence analysis.
The authors test the proposed methods in several deep learning tasks and show the improvement over some existing methods.

**Summary Of The Review:**

The paper is in general well-organized.  However, as pointed out by other reviewers, the empirical evaluations are not fair and the technical statement could be incorrect.

---

> ### Author Response · Authors · 2021-11-22
> **Response to Reviewer  5ZsD**
>
> We thank you for your valuable comments. Here is our response to your major questions.
>
>
> **Q1**: The presentation in Sec 3.1 can be further improved.
> **A1**: Good point and thanks for the suggestion. Yes, Figure 1 is our Schematic diagram to convey the idea that the effective stepsize of AdaMomentum is usually more reasonable and more suitable for general loss curvature than Adam. We will polish this section and construct a 1D example with plotted trajectories in the revision.
>
> **Q2**: On the convergence analysis in Sec 4.
> **A2**: Thanks for your constructive comment. We would like to first clarify that providing convergence guarantees is a standard and common part for papers proposing a new optimization algorithm in ML and DL optimization literature [1,2,3,4]. Our algorithm is novel and different from other variants, hence our theoretical results are also novel and valuable, although some proof techniques are borrowed from existing literature. Compared to Adam, our theoretical analysis of the regret bound under convex setting needs fewer assumptions (no need of L2 distance of the iterates to be bounded). Also, ours are correct while the regret bound theorem in Adam paper is actually incorrect pointed by [3]. Besides, we provide convergence under non-convex setting, which is nonexistent in Adam paper.
>
> In the non-convex case, our Theorem 3 is written in a more general way than all the existing related results, and our complexity result is also tighter than Theorem 2.2 in AdaBelief paper [4] (getting rid of the gradient noise variance term).
>
> **Q3**: On the general non-convex optimization related to Sec 5.1.1.
> **A3**: Good point and thanks for the suggestion. We would like to explain that we provide 2D examples to show the superior optimization ability of AdaMomentum on simple and classic loss cases. Our experiments on different complex DL tasks have shown the superiority of our method in both optimization (GAN, training curves of other experiments) and generalization (testing results) for high-dimensional non-convex loss functions. We will definitely supplement additional experiments of high-dimensional (e.g. d=100) cases on the four classic functions in the revision.
>
>
> Thanks again for your valuable comments.  If you have further questions, feel free to ask and we are willing to have further discussions with you.
>
> ### References
> [1] Diederik P. Kingma et al. Adam: A Method for Stochastic Optimization. ICLR 2015.
> [2] John Duchi et al. Adaptive Subgradient Methods for Online Learning and Stochastic Optimization. JMLR 2012.
> [3] Sashank J. Reddi et al. On the Convergence of Adam and Beyond. ICLR 2018.
> [4] Juntang Zhang et al. AdaBelief Optimizer: Adapting Stepsizes by the Belief in Observed Gradients. NeurIPS 2020.

---

> ### Comment · Reviewer_5ZsD · 2021-11-30
> **Updated review**
>
> I agree with other reviewers. I lower my score.
>
> As pointed out by other reviewers, the authors do not fairly tune the baselines. For example, in the 2d toy example, the step-size for baseline methods should be tuned, and the hyper-parameter initialization (e.g, epsilon for Adam) should be carefully conducted.
>
> I also think many diagonal adaptive-gradient methods including Adam cannot perform well for general non-convex functions beyond the DL-type loss functions. Sec 5.1.1 could give readers a misleading impression that the proposed method works well for general non-convex functions.
> It is very likely that the proposed method performs as poorly as Adam in higher-dimensional  Ackley function and Rosenbrock function compared to non-diagonal methods such as BFGS or LBFGS.

---

> > ### Author Response · Authors · 2021-11-30
> > **Reponse to the updated review of Reviewer 5ZsD**
> >
> > Thanks very much for your updated review. We think your valuable comments help us improve the paper.
> >
> > We will add the results of well fine-tuned toy example experiments and the results in higher dimensions in the revision.

---

### Official Review · Reviewer_a87S · 2021-11-02

**Correctness:** 3
**Technical Novelty And Significance:** 3
**Empirical Novelty And Significance:** 3
**Recommendation:** 6
**Confidence:** 2

**Main Review:**

ttStrengths
1. The paper is well written.
2.The intuition of estimating  second moment by momentum  is interesting.
3. Theoretical results and experimental result is provided.

Weaknesses
1. The assumption in Theorem 1 seems wired, where it requires the   distance between any θt generated by AdaMomentum is bounded. I am not sure whether this assumption is a widely accepted assumption or it is a special assumption required by the proposed algorithm. If yes, it will be much better if the authors can clearly state this for clarification.
2. The assumption in Theorem 2 seems has the same confusion, where it requires alpha_t/sqrt(v_t) > alpha_(t+1)/sqrt(v_(t+1)). Whether it can be satisfied by using the setting in Corollary 2 or it still a assumption is not clear. It will be much better if the authors can clarify this in their paper.

**Summary Of The Paper:**

This paper study the  adaptive type gradient methods, such as ADAM which scaling gradients by square roots of the running averages of
squared past gradients. Nevertheless,  they discover that substituting the gradient in the second moment estimation term with the momentumized version in ADAM can well solve the weaknesses of ADAM which  generalize worse than stochastic gradient descent (SGD) and tend to be trapped in local minima at an early stage during training.Their intuition is that gradient with momentum contains more accurate directional information and therefore its second moment estimation is a better choice for scaling than that of the raw gradient. Thereby they propose ADAMOMENTUM as a new optimizer reaching the goal of training fast while generalizing better. Convergence guarantee is provided   and   extensive experiments on a wide range of tasks and models is also conducted.

**Summary Of The Review:**

This paper proposed a adaptive type gradient methods named ADAMOMENTUM  by the intuition that gradient with momentum contains more accurate directional information and therefore its second moment estimation is a better choice for scaling than that of the raw gradient.  The intuition, theory, and experiments seems OK. Therefore, I tend to accept this paper at this time. I am not an expert at this area correct me if I am wrong, and I am willing to change my score to align with other expert's score if needed and reasonable.

---

> ### Author Response · Authors · 2021-11-22
> **Response to Reviewer a87S**
>
> We thank you for your valuable comments. Here is our response to your major questions.
>
>
> **Q1**: On the assumption of Theorem 1.
> **A1**: Good point. However, we would like to point out that the bounded assumption of the distances of the iterates is a very standard and widely accepted assumption in DL optimizer papers (also appear in [1,2,3,4]). Even in the original Adam [4] paper, their main theorem (Theorem 4.1) requires $\left\lVert \theta_m - \theta_n\right\rVert_\infty$ to be uniformly bounded, the same as our paper. In Adam paper they also need $\left\lVert \theta_m - \theta_n\right\rVert_2$ to be uniformly bounded, which is a stronger assumption than ours.
>
> The distance bounded assumption can also be easily verified in practice. No matter what task, after some moderate hyper-parameter tuning, we can easily make the training effective with no parameter taking value of NaNs using AdaMomentum. This is also the case for most of the parameters.
>
> **Q2**: On the assumption of Theorem 2.
> **A2**: Thanks for your constructive comments. However, we would like to clarify that this assumption is commonly used for the non-convex analysis of Adam-alike algorithms [5,6] since it could be satisfied under multiple widely-used settings. For example, as we mentioned in our paper, we can simply choose $\alpha_t = \alpha/\sqrt{t}$ and $\beta_{2,t} = 1/t$ as AdaFom [6] did （the choice of $\alpha_t$ here corresponds to Corollary 2 in our paper. In fact, in code the assumption $\alpha_t / \sqrt{v_t} \ge \alpha_{t+1} / \sqrt{v_{t+1}}$ requires to use element wise maximum between $v_t$ and $v_{t+1}$ in the denominator, as in [3,5].
>
>
> Thanks again for your valuable comments. If you have further questions, feel free to ask and we are willing to have further discussions with you.
>
> ### References
> [1] Fartash Faghri et al. A study of gradient variance in deep learning. arXiv preprint arXiv:2007.04532, 2020.
> [2] Diederik P. Kingma and Jimmy Ba. Adam: A method for stochastic optimization. In ICLR, 2015.
> [3] Sashank J Reddi et al. On the convergence of adam and beyond. ICLR, 2019.
> [4] Liangchen Luo et al. Adaptive Gradient Methods with Dynamic
> Bound of Learning Rate. ICLR 2019.
> [5] Juntang Zhang et al. AdaBelief Optimizer: Adapting Stepsizes by the Belief in Observed Gradients. NeurIPS 2020.
> [6] Xiangyi Chen et al. On the convergence of a class of Adam-type algorithms for non-convex optimization. ICLR 2019.

---

### Official Review · Reviewer_hk95 · 2021-11-02

**Correctness:** 2
**Technical Novelty And Significance:** 2
**Empirical Novelty And Significance:** 2
**Recommendation:** 3
**Confidence:** 5

**Main Review:**

Strengths: the empirical results are good for language modeling, and it is comparable with SGD with momentum on vision tasks.

Weaknesses: my main concern of this manuscript is limited technical novelty and minimal insight.

1. Lemma 1 leverages the tool of SDE to show closed-form formula of escaping time. However, the analysis is almost the same as the Theorem 1 in [Zhou et al. NeurIPS 2020]. One only need to change some definitions in the proof of [Zhou et al. NeurIPS 2020] and then can easily prove the Lemma 1.

2. Assumption 5 is the key to show the improved escaping time in the proof of Proposition 1. It is an unrealistic assumption, it is possible that the RHS is smaller than LHS when $t$ gets large. Please give more explanations.

3. In Theorem 1, the authors require the iterates to be bounded. One cannot simply assume it holds without proving it.

4. In Theorem 2, the authors assume $\alpha_t/\sqrt{v_t}\geq \alpha_{t+1}/\sqrt{v_{t+1}}$ to prove the convergence in non-convex case, which is problematic. For example, [Reddi et al., ICLR 2018] provided a non-convergence analysis of Adam and they showed that this inequality does not hold at all.






**Summary Of The Paper:**

This manuscript proposes to substitute the gradient in the second moment estimation term with the momentumized version, and show that it improves both optimization and generalization. Some theory is presented, and there are also some promising deep learning experiments on different tasks.

**Summary Of The Review:**

Overall, this manuscript presents an interesting modification of Adam to improve empirical performance. However, the technical novelty is very limited and there is no new theoretical insight. The generalization analysis using SDE comes from [Zhou et al. NeurIPS 2020], and the optimization analysis is based on a problematic assumption pointed out by [Reddi et al. ICLR 2018]. There are also some unrealistic assumptions (e.g., Assumption 5) which needs to be further justified.

For its current version, it clearly does not reach the bar of top venues such as ICLR.

---

> ### Author Response · Authors · 2021-11-22
> **Response to Reviewer hk95**
>
> We thank you for your valuable comments. Here is our response to your major questions.
>
>
> **Q1**: On Lemma 1.
> **A1**: Thanks for your constructive comment. Yes, Lemma 1 in our paper is built up on [1], which is an adaptation of Theorem 1 in [1]. We clarify here that Lemma 1 serves as a tool based on which we prove our Proposition 1, which  explains why AdaMomentum can generalize better in experimental evaluations. Note that our paper is not a pure optimization theory paper, but more a practical deep learning optimizer designing paper. Our theoretical parts serves to help explain our empirical gain and provide guarantees, not to develop a completely new theory from zero.
>
> **Q2**: On Assumption 5.
> **A2**: Thanks for the advice. As we have mentioned under Assumption 5 in our paper, the assumption basically requires that the gradient noise variance to be smaller than the second moment of $m$ when $t$ is very large. This assumption is generally mild as: 1) the magnitudes of the variances of the stochastic gradients are usually much lower than that of the gradients [4]; 2) We can select mini-batch size to be large enough to satisfy it as the noise variance is inversely proportional to batch size as in [2,3]. More specifically, we have $g_t = \frac{1}{M} \sum_{i=1}^M G_t(\theta_t, \xi_i)$, here G_t is the noisy gradient using each data sample, and $M$ is the batch size in training. Then taking the expectation w.r.t. the randomness of the gradient noises, we have
>
> $\mathbb{E} \||g_t - \nabla f(\theta_t)\||^2 = \mathbb{E} \||\frac{1}{M} \sum_{i=1}^M G_t(\theta_t, \xi_i) - \nabla f(\theta_t)\||^2 = \frac{\sum_{i=1}^M \mathbb{E} \||G_t(\theta_t, \xi_i) - \nabla f(\theta_t)\||^2}{M^2}.$
>
> Usually $\mathbb{E} \||G_t(\theta_t, \xi_i) - \nabla f(\theta_t)\||^2$ is bounded by constant (e.g. $\sigma^2$), hence we have
> $diag(\Sigma_t) \propto O(\frac{1}{M}) $. Therefore we can choose sufficiently large batch size $M$ to let Assumption 5 hold.
>
> **Q3**: On the boundedness assumption of Theorem 1.
> **A3**: Thanks for your careful review and suggestion. However, the bounded assumption of the distances of the iterates is a **very standard and widely accepted assumption** in DL optimizer papers (also appear in [4,5,6,7]). Even in the original Adam [4] paper, their main theorem (Theorem 4.1) requires $\left\lVert \theta_m - \theta_n\right\rVert_\infty$ to be uniformaly bounded, the same as our paper. In Adam paper they also need $\left\lVert \theta_m - \theta_n\right\rVert_2$ to be uniformly bounded, which is a stronger assumption than ours.
>
> The distance bounded assumption can also be easily verified in practice. No matter what task, after some moderate hyper-parameter tuning, we can easily make the training effective with no weight parameter taking value of NaNs using AdaMomentum. This is also the case for most of the other optimizers.
>
> **Q4**: On the assumption of Theorem 2.
> **A4**: Thanks for the constructive comment. However, we would like to clarify that this assumption is commonly used for the non-convex analysis of Adam-alike algorithms [8-9] since it could be satisfied under multiple widely-used settings. For example, as we mentioned in our paper, we can simply choose $\alpha_t = \alpha/\sqrt{t}$ and $\beta_{2,t} = 1/t$ as AdaFom [9] did. Beside, [6] provides examples of non-convergence of the regret of Adam and shows the inequality may not necessarily hold, NOT "does not held at all". In fact, in code this assumption requires to use element wise maximum between $v_t$ and $v_{t+1}$ in the denominator, as in [6,8].
>
>
> Thanks again for your valuable comments. We believe we have addressed most of your concerns. If you have further questions, feel free to ask and we are willing to have further discussions with you.
>
> ### References
> [1] Pan Zhou et al.Towards theoretically understanding why sgd generalizes better than adam in deep learning. NeurIPS 2020.
> [2] Sebastien Bubeck. Convex optimization: Algorithms and complexity. arXiv preprint arXiv:1405.4980, 2014.
> [3] Tianyi Lin et al. On gradient descent ascent for nonconvex-concave minimax problems. ICML 2020
> [4] Fartash Faghri et al. A study of gradient variance in deep learning. arXiv preprint arXiv:2007.04532, 2020.
> [5] Diederik P. Kingma and Jimmy Ba. Adam: A method for stochastic optimization. In ICLR, 2015.
> [6] Sashank J Reddi et al. On the convergence of adam and beyond. ICLR, 2019.
> [7] Liangchen Luo et al. Adaptive gradient methods with dynamic bound of learning rate. ICLR 2019.
> [8] Juntang Zhang et al. AdaBelief optimizer: adapting stepsizes by the belief in observed gradients. NeurIPS 2020.
> [9] Xiangyi Chen et al. On the convergence of a class of Adam-type algorithms for non-convex optimization. ICLR 2019.

---

> > ### Comment · Reviewer_hk95 · 2021-11-29
> > **Response**
> >
> > I think the authors for taking the time writing the rebuttal. However, it does not address my concerns.
> > For Q1, the authors acknowledge that this is not novel and mostly adapted from previous work.
> >
> > For Q2, we know that large batch training usually gives worse results than small batch. So this assumption is still not valid. In addition, when the gradient is small, why the gradient's magnitude is larger than the noise?
> >
> > For Q3, Q4, The fact that these assumptions were used in previous work does not imply that it is correct. For example, we all know the proof of Adam is problematic just because of some wrong assumptions (See the AMSgrad paper). Why keeping using it even if it is wrong?
> >
> > Due to these reasons, I still keep my original evaluation and recommend to reject this paper.

---

> > > ### Author Response · Authors · 2021-11-30
> > > **Response to the updated comment of Reviewer hk95**
> > >
> > > Thank you for your updated comments and valuable advice. We appreciate your efforts in making this paper better.
> > >
> > > For Q2, we will provide additional empirical illustrations in the revision showing that Assumption 5 can be generally satisfied.
> > >
> > > For the assumption in Theorem 2, we will add more modifications and clarifications in the revision.

---

> ### Author Response · Authors · 2021-11-28
> **Sincerely expecting further discussions with Reviewer hk95**
>
> Dear Reviewer hk95,
>
> We greatly thank you for the reviewing process so far! Given the ICLR final discussion deadline (11/29) is approaching, we really hope to have a further discussion with you to see if our responses solve your concerns. Thanks!
>
> Sincerely,
>
> Authors

---

### Official Review · Reviewer_mUyQ · 2021-11-02

**Correctness:** 3
**Technical Novelty And Significance:** 2
**Empirical Novelty And Significance:** 2
**Recommendation:** 3
**Confidence:** 5

**Main Review:**

Strengths:
I appreciate the authors' efforts to provide both theoretical and experimental validations on the proposed method. The authors try to understand both the convergence and generalization of the algorithm, and the experiments also cover a wide range of neural network architectures and tasks.

Weakness:
Despite that I appreciate the authors' efforts, I found potential flaws in both theory and experiments, listed below.

Theory:

1. Fig1 as a demonstration of the idea does not make sense to me. It essentially states that AdaMomentum is easier than Adam to be trapped at the 1st local minimum it encounters. But the question is, why the local minimum you encountered early is the global optimal? Give me an example that behaves like in Fig.1, I can easily modify the loss function to the right of $\theta_{t+1}^{(2)}$ as the global optima, and keep the function to its left the same as in the figure (so that the behavior of Adam and AdaMomentum don't change until they pass $\theta_{t+1}^{(2)}$). In this case the right minimum is better than the left one, and Adam would jump in the global optima while AdaMomentum will be trapped in a bad local minimum.
In short, which minima is the global minima can be easily changed, AdaMomentum sticks to the 1st one, but the 1st minima can either be a local or global minimum. You only pick the case that's good for AdaMomentum, but ignore the bad case.

2. The generalization part mostly follows [1], however, I think your results contradict [1]. [1] stated that SGD has a **smaller** escaping time than Adam hence SGD can escape local minima and generalize **better**. Your paper stated that AdaMomentum has a **longer** escaping time than Adam but still generalizes better? There's a clear contradiction here.
I might miss something with the "flat basin" part, I think it's referring to assumption 4. After going over the proof, I think you will have $\mathbb{E}\tilde{v} \geq \mathbb{E} v$ for both sharp and flat minima, hence AdaMomentum always has a longer escaping time than Adam. At least this implies that AdaMomentum is more likely than Adam to stay at a bad local minimum if they fall into it, this would only hurt the generalization rather than improve it.

3. The proposed method essentially slows down the change of denominator, how does this compare to using a larger $\beta_2$?

4. The current version of AdaMomentum scales lr up by a factor of $1/(1-\beta_1^t)$ (see comment Experiment 1 below for an example), which contradicts findings in RAdam that warmup from a small lr to a large lr would stabilize the training. I wonder what's the authors' comment on this?

5. Discussion on $\epsilon$. First I don't think the equation above Sec3.2 is correct, it should be $v_t^\prime=v_t + \epsilon/ (1-\beta_2^t)$, as $t\to \infty$, $v^\prime \to v_t + \epsilon$. Second the authors confuse $\epsilon_{Adam}$ with the $\epsilon$ here, in general $\epsilon_{Adam}$ is outside $\sqrt{.} $ but $\epsilon_{AdaBelief}$ is inside $\sqrt{.}$, that's why default is $\epsilon_{Adam}=1e-8, \epsilon_{AdaBelief}=1e-16$. Considering this, $\sqrt{\epsilon_{AdaBelief}} \approx \epsilon_{Adam}$, and AdaMomentum uses the same default as AdaBelief, I don't think the arguments by the authors are convincing. The effect of $\epsilon$ is amplified because you move it outside the $\sqrt{.}$, but this effect is corrected by using a squared value (1e-16 rather than 1e-8) as default.

Experiments:

1. Toy examples. It's very hard to notice until I checked the code that I found the authors only plot the behavior of the first 30 steps or so. Note that $\mathbb{E} m_t = (1-\beta_1^t) \mathbb{E} g_t$ in general. This implies that, AdaMomentum scales the denominator by a factor of $1-\beta_1^t$, and the effective stepsize would be larger than Adam by a factor of $1/(1-\beta_1^t)$. When $t$ is small, this factor is very large (e.g. $t=1, \beta_1=0.9$, this factor is 10). This means the fast convergence of AdaMentum with few steps is heavily influenced by the large effective learning rate, rather than the $m_t$ vs $g_t$ issue. The authors need to consider this in the statement and experiments.

2. I was surprised to see the authors reported an FID of 12.06, so I ran the code twice, and got 12.68 and 12.80, which is way much worse than reported, even worse than baseline Adam. In case the authors uploaded the wrong file, I would run it again if the authors can re-upload the code (in a single zip file please, from the previous link I have to download files one by one).

3. ImageNet experiments with ResNet18. This is again tricky that the authors switched to cosine learning rate, yet the results by AdaBelief is worse than reported in the original paper using a step lr schdule. I tried ResNet18 on ImageNet with a cosine learning rate, batchsize 4096, 90 epochs, initial learning rate 0.001*4096/512, weight decay 5e-2, and got 72.2 top-1 accuracy (70.45 for AdaMomentum). I'm using Jax with the AdaBelief in Flax, all default hyper-params. The settings are slightly different from this paper, but typically large-batch training is worse than small-batch training. So the ImageNet results look unconvincing to me.

[1] Zhou, Pan, et al. "Towards theoretically understanding why sgd generalizes better than adam in deep learning." arXiv preprint arXiv:2010.05627 (2020).

**Summary Of The Paper:**

This paper proposed AdaMomentum, which uses the EMA (exponential moving average) of the square of EMA of gradient as sthe denominator, while Adam uses the EMA of the square of the gradient. The authors conducted experiments in CNN, Transformer, LSTM and GAN to show the superior performance of AdaMomentum. The authors also tried to provide theoretical analysis.

**Summary Of The Review:**

The theoretical parts on generalization contradict the literature. The experimental results also mismatch literature, and I got a far worse result on GAN training using the authors' code (could be the wrong version, I can re-run if the authors can re-upload code into a single file). I tend to reject for now and would increase the rating if the authors can resolve my comments above.

---

> ### Author Response · Authors · 2021-11-21
> **Response to Reviewer mUyQ (1/4)**
>
> We thank you for your valuable comments. Here is our response to your major questions.
>
> ### Theory
>
> **Q1**: On the location of local minimum and global minimum in Fig. 1.
> **A1**: Thanks for your comment.  We would like to carify that our point is not that we will encounter global minima earlier than the local minimum. Our point is that compared to Adam, AdaMomentum will have smaller effective learning rate near the convergence. No matter the position  of the local minima (either to the left or right of the global minimum in the 1D example), avoiding large effective stepsizes near the minima is beneficial for training, as stated by [1,2]. Because if the stepsize is too large near the minima, the weight parameter will oscillate, making the training unstable  and fail to converge to desirable points. This is also confirmed by standard practice of learning rate annealing approaches (step drop annealing or cosine annealing techniques). As we have mentioned on l.13 of page 4 of the main paper, the right part  of Fig. 1 is only a illustration to convey the "small stepsize near convergence is beneficial" idea (also considering it is a 1-D example). We admit that there may be some problems in our presentation and writing leading to the misunderstandings, and we apologize if any inconvenience is caused for the reviewers and readers. We will carefully rewrite and polish this part in the revision.
>
> **Q2**: The generalization part contradicts [3].
> **A2**: Thanks for the careful review. As we have stated in Sec.3.2, from Lemma 1 we have the escaping time is negatively correlated with the volume of the set $\Upsilon$. Therefore, we have that for both Adam and AdaMomentum, if the basin $\boldsymbol{\Omega}$ is sharp (the local minimum is bad) which is ubiquitous during the early stage of training, $\Upsilon$ has a large Radon measure, which leads to smaller escaping time $\Gamma$. This means both Adam and AdaMomentum prefer relatively flat or asymmetric basin through the training process. However, upon encountering a comparatively flat basin or asymmetric valley $\boldsymbol{\Omega}$, we are able to prove that AdaMomentum will stay longer inside. Due to the fact that minima at the flat or asymmetric basins tend to exhibit better generalization performance, we can conclude that AdaMomentum is more likely to converge to minima that generalize better.
>
>
> For the contradiction with [3], we would like to clarify that **authors in [3] made a mistake in their work when calculating the volume of $d$ dimensional ellipsoid in the end of section 4**: for the compliment of the ADAM escaping set $W_{\text{ADAM}}^c = \\{y \in \mathbb{R}^d \, | \, y^\top H(\theta^*) y < S^2 h_f^* \\}$ with singular values of $H(\theta^*)$ as $\lambda_1 \geq \cdots \geq \lambda_d$, its volume should be $V(W_{\text{ADAM}}^c) = \zeta/(\prod_{i=1}^d \lambda_i)$ instead of $\zeta\prod_{i=1}^d \lambda_i$ where $\zeta = 2d^{-1} (\pi S/h_f^*)^{d/2} g^{-1}(d/2)$ with a gamma function $g$. Similarly, we have the volume of the compliment of the SGD escaping set as $V(W_{\text{SGD}}^c) = \zeta/(\prod_{i=1}^d \varsigma_i^2 \lambda_i)$ instead, and hence actually it should be $V(W_{\text{SGD}}^c) \geq V(W_{\text{ADAM}}^c)$ consequently. **So the conclusion of the right version of paper [3] is that the escaping time of SGD is longer than that of Adam**. Considering that SGD exhibit better generalization when testing compared to Adam when they reach similar training performance, the escaping time of an optimizer under such theoretical framework is generally positively correlated to its generalization performance. Hence the fact that the escaping time of AdaMomentum is longer also implies its better generalization performance, which further confirms the correctness of our argument above.

---

> > ### Author Response · Authors · 2021-11-21
> > **Response to Reviewer mUyQ (2/4)**
> >
> > **Q3**: The proposed method essentially slows down the change of denominator, how does this compare to using a larger $\beta_2$?
> > **A3**: Thanks for the suggestion but using a larger $\beta_2$ can **NOT** replace the role of our proposed AdaMomentum.
> >
> > From theoretical perspective, we have in AdaMomentum,
> > $$v_t = (1-\beta_2) \sum_{i=1}^t \beta_2^{t-i} m_i^2 = (1-\beta_2) \sum_{i=1}^t \beta_2^{t-i} [(1-\beta_1) \sum_{j=1}^i \beta_1^{i-j} g_j]^2 = (1-\beta_2)(1-\beta_1)^2\sum_{i=1}^t \beta_2^{t-i} [\sum_{j=1}^i \beta_1^{i-j} g_j]^2.$$ We can see that $v_t$ contains items $g_k g_j (1\le k,j \le t)$, which basically considers the similarities of gradient at past different iterations. In contrast, in Adam,
> > $$v_t = (1-\beta_2) \sum_{i=1}^t \beta_2^{t-i} g_i^2,$$ which is exponential moving average of second moment of the gradients (i.e. asigining weights to past square gradients at exponentially decaying speed with time step). Merely increasing $\beta_2$ only slows down the decaying a little bit, but can **NOT** consider the similarities between squared gradients at different time steps as AdaMomentum did.
> >
> > From empirical perspective, merely increasing the value of $\beta_2$ fail to improve performance as AdaMomentum does. $\beta_2 = 0.999$ is optimal for Adam in most applications. For instance, we perform experiments of VGG-16 on CIFAR-10, 3-layer LSTM on  Penn Treebank dataset with larger $\beta_2$ values $0.9995, 0.9999, 0.99999$ (the default value is $0.999$). All other parameters are set as the same as the experiment setting as in the paper. The average results with stds using 5 seeds are summarized below.
> >
> > Table 1: Image classification results on CIFAR-10 using VGG-16 of Adam with different $\beta_2$s.
> >
> > |$\beta_2$|0.999|0.9995|0.9999|0.99999|
> > |-|-|-|-|-|
> > |Test accuracy (%) ($\uparrow$)|93.29$\pm$0.10|93.33$\pm$0.14|93.26$\pm$0.17|93.28$\pm$0.19|
> >
> > Table 2: Language modeling results on Penn Treebank dataset using 3-layer LSTM with different $\beta_2$s.
> >
> > |$\beta_2$|0.999|0.9995|0.9999|0.99999|
> > |-|-|-|-|-|
> > |Test perplexity ($\downarrow$)|64.10$\pm$0.25|64.76$\pm$0.24|65.66$\pm$0.24|65.85$\pm$0.31|
> >
> > As illustrated in Tab.1 and 2, there is not (much) improvement in performance when we use larger $\beta_2$. Note that our AdaMomentum achieves 94.80$\pm$0.10 accuracy using VGG-16 on CIFAR-10 and 60.08$\pm$0.11 test perplexity using 3-layer LSTM on Penn Treebank dataset, which are much better.
> >
> >
> >
> > **Q4**: AdaMomentum scales lr up by a factor of $\frac{1}{1-\beta_1^t}$, which contradicts findings in RAdam that warmup from a small lr to a large lr would stabilize the training.
> > **A4**: Please see our response to your **Q1 in Experiment** part in the below, where we think that your saying that AdaMomentum scales lr up by a factor of $\frac{1}{1-\beta_1^t}$ may not be tenbale. Also, in our NMT experiments, we have also used warm up mechanism (see Sec. 5.2.3). AdaMomentum can be equipped with warm up for stable training especially on NLP tasks and the two is not contradictory.
> >
> > **Q5**: Discussion on $\epsilon$ and the doubt of the correctness of the equation above Sec 3.2.
> > **A5**: Firstly, we would like to show that our equation above Sec. 3.2 is correct. The calculation process is listed as follows: $$v_t = \beta_2 v_{t-1} + (1-\beta_2) m_t^2 + \epsilon = \beta_2(\beta_2 v_{t-2} + (1-\beta_2) m_{t-1}^2 + \epsilon) + (1-\beta_2)m_t^2 +\epsilon = \cdots = (1-\beta_2) \sum_{i=1}^t \beta_2^{t-i} m_i^2 + \frac{1-\beta_2^t}{1-\beta_2} \epsilon.$$ Dividing both sides by $(1-\beta_2^t)$ we have:
> > $$\hat{v}_t = \frac{1-\beta_2}{1-\beta_2^t}  \sum_\{i=1\}^t \beta_2^{t-i} m_i^2 + \frac{1}{1-\beta_2} \epsilon = \hat{v}'_t + \frac{1}{1-\beta_2} \epsilon.$$ We apologize that there is a typo in our paper on the line after the equation where we originally write "$\hat{v}_t' = \hat{v}_t + \frac{\epsilon}{1-\beta_2}$": it should be "$\hat{v}_t = \hat{v}_t' + \frac{\epsilon}{1-\beta_2}$" and we have corrected it in the revision version.
> >
> > Secondly, we did **NOT** mention AdaBelief in our analysis of $\epsilon$. The $\epsilon_{Adabelief}$ is different case in that there are two $\epsilon$ in AdaBelief, one added both outside and one inside $\sqrt{\cdot}$. There is only one $\epsilon$ in Adam and AdaMomentum. $\epsilon_{Adam}$ is outside $\sqrt{\cdot}$ and $\epsilon_{AdaMomentum}$ is inside $\sqrt{\cdot}$. Also, AdaMomentum dose **NOT** use the same default as AdaBelief. For instance, on LSTM experiments, our default $\epsilon$ value is $\epsilon=1e-16$ while $\epsilon_{AdaBelief}=1e-12$ when layer number is 2 and 3.  Is there a typo in your saying "The effect of $\epsilon$ is amplified because you move it outside $\sqrt{\cdot}$"?  we move $\epsilon$ **inside** $\sqrt{\cdot}$, **NOT** outside. We do **NOT** think such effect is corrected by using a squared value, because  1) for image tasks the default $\epsilon$ for both AdaMomentum and Adam is 1e-8;

---

> > > ### Author Response · Authors · 2021-11-21
> > > **Response to Reviewer mUyQ (3/4)**
> > >
> > > ;2) The $\epsilon$ is added to prevent the denominator from being too close to zero in some cases, for most of the time the $v_t$ term plays the key role. In those cases, $$\sqrt{\hat{v}_t + 1e-16} \neq \sqrt{\hat{v}_t} + 1e-8.$$
> > >
> > > ### Experiments
> > >
> > > **Q1**: On the toy experiments the authors only plot the behavior of the first 30 steps or so.
> > > **A1**: Thanks for the careful review and checking our provided code. Yes, we plot the 2-D example annimation by 30 to 50 steps. This is to better illustrate that AdaMomentum can converge to the optimum and achieves the fastest convergence. When we plot for longer iterations, our AdaMomentum always achieves the fastest and steadiest convergence while the other baseline optimizers including Adam can not be as fast or even can not reach the optimum (we plot up to 150 iterations and the GIFs are on this anonymous link https://imgur.com/a/68oWsqs).
> > >
> > > Also, we respectfully disagree with your further analysis. **We think $\mathbb{E} m_t \neq (1-\beta_1^t) \mathbb{E} g_t$ in general**. Because by induction we have
> > > $$
> > > m_t = (1-\beta_1)\sum_{i=1}^t \beta_1^{t-i} g_i.
> > > $$
> > > Taking the expectation to both sides of the above equation, we have
> > > $$
> > > \mathbb{E} m_t = (1-\beta_1)\sum_{i=1}^t \beta_1^{t-i} \mathbb{E}g_i \neq (1-\beta_1) \sum_{i=1}^t \beta_1^{t-i} \mathbb{E}g_t.
> > > $$
> > > Only when $\mathbb{E}g_i$ is stationary ($\mathbb{E}g_i = \mathbb{E}g_t$) can $\mathbb{E} m_t = (1-\beta_1) \sum_{i=1}^t \beta_1^{t-i} \mathbb{E}g_t = (1-\beta_1^t) \mathbb{E} g_t$ hold. However, it is impossible for $\mathbb{E} g_i$ to stay the same in the practice of optimizaiton, especially in the early phase when the gradients tend to vary a lot in distribution. Even if we step back and suppose $\mathbb{E}m_t  = (1-\beta_1^t) \mathbb{E} g_t$ (generally it can not hold), this does **NOT** mean $\mathbb{E}m_i^2  = (1-\beta_1^t) \mathbb{E} g_t^2$. Then your argument that "AdaMomentum scales the denominator by a factor of $1-\beta_1^t$, and the effective stepsize would be larger than Adam by a factor of $\frac{1}{1-\beta_1^t}$" can not hold, which makes your statement that fast convergence of AdaMomentum is influenced by large effective learning rate untenable.
> > >
> > >
> > > **Q2**: Reproducibility of AdaMomentum on GAN experiments.
> > > **A2**: Thanks for the careful review and running our provided code. We have checked our codes and there is nothing wrong with our uploaded code version on the anonymous github link. We are surprised that you fail to reproduce our results on GAN. To show the reliability of our reported results, **we provide the original training log files** of the $5$ independent runs in our experiments, with best FID scores $11.967$, $11.741$, $12.391$, $12.054$, $12.150$. The average FID score is 12.061, which is what report in the paper. **We also provide the original checkpoints** of the trained generator and discriminator networks  corresponding to the results respectively. The logs and the checkpoints are on anonymous google drive link https://drive.google.com/drive/folders/1aSXSTrzq6jGw46SLdG9vPT0KiF6z59pR?usp=sharing (we name our optimizer as Adam_m in code-level when doing experiments at that time, the log and checkpoint files do not contain any personal information). The working platform is Ubuntu 18.04.6 LTS with Nvidia GeForce GTX Titan XP Graphic Card GPUs and Intel(R) Core(TM) i7-6850K CPU @ 3.60GHz CPUs and the CUDA version is 10.1.

---

> > > > ### Author Response · Authors · 2021-11-21
> > > > **Response to Reviewer mUyQ (4/4)**
> > > >
> > > > **Q3**: Results by AdaBelief [1] is worse than reported in the original paper using a step lr schdule. On ImageNet experiments with ResNet18, AdaBelief can achieve 72.2 top-1 accuracy in a different parameter setting.
> > > > **A3**: Thanks for your careful review. Yes, our reported results using cosine learning rate annealing is worse than reported in the original paper using a step lr. This is because in our experiments, we find **we can not reproduce the results reported in AdaBelief [1] paper where they report 70.08 top-1 accuracy**. Our experiments show AdaBelief can only achieve about 69.3 to 69.4 top-1 accuracy on ImageNet using the exactly same setting in AdaBelief [1] paper. **There are also other people saying they can not reproduce the AdaBelief reported results on ImageNet on github** (e.g., https://github.com/juntang-zhuang/Adabelief-Optimizer/issues/50). On this link, the author of AdaBelief replies "There might be some wrongly labeled validation images in ImageNet12, and you will need to 'blacklist' those images. That might cause the difference. Also random seed might be a reason." However, the practice of blacklisting some valiadation images will lead to unfair comparison of optimizer performance, which is **NOT** standard practice in training on ImageNet ( not existent on other optimizer papers). **The above reasons make us unconvinced of the reported result and performance comparison on ImageNet of AdaBelief [1]**. In our paper, we report the results of our runs of AdaBelief using cosine annealing. Typically, using cosine annealing can generate slightly better performance than step-drop decay. **We think our performance comparison on ImageNet is fair and reliable because we compare all the optimzers using the same setting and the hyperparameters are carefully tuned**.
> > > >
> > > > Thanks for running AdaBelief experiments. You tried ResNet18 on ImageNet with a cosine learning rate, batchsize 4096, 90 epochs, initial learning rate 0.001\*4096/512, weight decay 5e-2, and got 72.2 top-1 accuracy. However, we do **NOT** think this argument is defensible enough to make our experiments unconvincing. Firstly, **your experimental setting is different from ours**. You use batch size 4096  and choose 0.001\*4096/512 as initial lr on ImageNet with Jax. In our experiments, we use batch size 256 and lr 0.001 with PyTorch for all the adaptive gradient methods including AdaBelief (after finetuning lr carefully for all the methods). When the setting is different, we think the performance of two optimizers may not be directly compared. Also, batch size 4096 on ImageNet is too large for most research institutions or university labs to afford due to high GPU memory requirements. Secondly, **Training ResNet-18 on ImageNet and getting 72.2 top-1 accuracy with AdaBelief is too amazing to us. Note that the official implemention result on PyTorch website using ResNet-18 is 69.758 (optimizing using SGD with momentum)(https://pytorch.org/vision/stable/models.html)**. Except we use cosine annealing strategy for learning rate, our experiments on ImageNet strictly follow the training recipe provided by PyTorch team (https://github.com/pytorch/examples/blob/master/imagenet/main.py) without additional tricks (like additional data augmentations). We think maybe you use extra tricks? Or maybe the language framework Jax matters?
> > > >
> > > >
> > > >
> > > > Thanks again for your very careful reviews and valuable comments. We believe we have addressed most of your concerns. If you have further questions, feel free to ask and we are willing to have further discussions with you.
> > > >
> > > > ### References
> > > > [1] Juntang Zhang et al. AdaBelief Optimizer: Adapting Stepsizes by the Belief in Observed Gradients. NeurIPS 2020.
> > > > [2] Liangchen Luo et al. Adaptive Gradient Methods with Dynamic
> > > > Bound of Learning Rate. ICLR 2019.
> > > > [3] Pan Zhou et al.Towards theoretically understanding why sgd generalizes better than adam in deep learning. NeurIPS 2020.

---

> ### Author Response · Authors · 2021-11-28
> **Sincerely expecting further discussions with Reviewer mUyQ**
>
> Dear Reviewer mUyQ,
>
> We greatly thank you for the reviewing process so far! Given the ICLR final discussion deadline (11/29) is approaching, we really hope to have a further discussion with you to see if our responses solve your concerns. Thank you!!
>
> Sincerely,
>
> Authors

---

> ### Comment · Reviewer_mUyQ · 2021-11-28
> **Updated review after author response**
>
> Thanks for your response, however, most of my concerns still remain. I list them below from most important to unimportant.
>
> 1. The theoretical results contradict [1], which is published in NeurIPS20 after peer review. The authors claimed results in [1] to be wrong. Note that this is the most important statement by [1] and is stated almost everywhere in [1], supported with theoretical analysis and experimental validations. The authors might be correct to claim [1] to be wrong, but a more detailed discussion is needed, the argument within a paragraph is far from convincing.
>
> 2. The authors stated ImageNet experiment in [2] is "unfair and unconvincing" because they used a blacklisted version dataset. I quickly searched online, blacklisting wrongly labeled data is "recommended in imagenet devkit" https://github.com/stanford-futuredata/dawn-bench-entries/issues/36#issuecomment-383401717 and fastai official repository https://github.com/fastai/imagenet-fast/blob/master/imagenet_nv/blacklist.sh
>
> 3. GAN experiments. This is the log https://www.dropbox.com/sh/g5vgi52vveuf6eb/AACeevEhFcMY0EQhvXaDkQP8a?dl=0 that I achieved 12.68 and 12.8 FID using your code, to make sure it's not a problem with my platform, I also ran on Adam and got 12.84. The Adam result is within the std of reported results, but AdaMomentum is almost outside 4 sigma region.  In the initial review I said I would rerun with new version code if the author uploaded the wrong version code, but since the authors stated their code to be correct, I would take it that the result is not stable enough to support your conclusion.
>
> 4. Regarding my experiments with a large batch ( I won't lower my rating by comparing the author's results with my baseline of 72% accuracy, because I have not confirmed it's the same across platforms, but I don't think the authors' argument is convincing). For my settings, I checked I did not use different augmentation. I admit my setting is different from yours, but I don't see any of my settings would unfairly increase accuracy, on the contrary, the large batch in my setting typically decreases accuracy.
>
> 5. Regarding $\epsilon$. Let's denote $\epsilon_1=10^{-16}, \epsilon_2=10^{-8}$. My main point is it's unfair to compare AdaMomentum using $\epsilon_2$ with Adam using $\epsilon_2$, because one is inside $\sqrt{.}$ the other is outside. For CNN experiments for example, if you use $\epsilon_2$ in AdaMomentum, I think you should compare with Adam using $\epsilon_3=10^{-4}$ or even $ \sqrt{\frac{1}{1-\beta_1}} 10^{-4}$ if you consider the in-place add of epsilon in AdaMomentum.
>
> 6. Toy example. My main point is not whether the scale is $\frac{1}{1-\beta_1^t}$ or not, my point is $m_t$ is smaller than $g_t$ by a factor (up to $1-\beta_1$ approximately), then the stepsize is larger by up to $\frac{1}{1-\beta_1}$, this is a big factor if your toy example only runs for a hundred steps. If you manually increase the stepsize by this factor for Adam, it will also converge much faster.
>
> Overall, it seems the authors tend to claim others to be problematic quite easily when there's a contradiction with the author's results (point 1 and 2 for example), but I want to remind the authors to search a bit more and provide more evidence before coming to such conclusions, especially the authors of the papers you criticized won't be able to participate in the discussion. I'm not saying [1] is completely correct, but more evidence is certainly necessary since you are flipping the most important claim of [1]. I would suggest the same if you serve as a reviewer later, that's why I said I would rerun your code if you uploaded a new version.
>
> [1] Zhou, Pan, et al. "Towards theoretically understanding why sgd generalizes better than adam in deep learning." arXiv preprint arXiv:2010.05627 (2020).
> [2] Juntang Zhang et al. AdaBelief Optimizer: Adapting Stepsizes by the Belief in Observed Gradients. NeurIPS 2020.

---

> > ### Author Response · Authors · 2021-11-30
> > **Reponse to Updated review of Reviewer mUyQ (1/3)**
> >
> > We thank you for your updated comments. Here is our response to your remaining concerns.
> >
> > **Q1**:
> > > The theoretical results contradict [1], which is published in NeurIPS20 after peer review. The authors claimed results in [1] to be wrong. Note that this is the most important statement by [1] and is stated almost everywhere in [1], supported with theoretical analysis and experimental validations. The authors might be correct to claim [1] to be wrong, but a more detailed discussion is needed, the argument within a paragraph is far from convincing.
> >
> > **A1**: Thanks for the clarification. The more detailed discussion of [1] is listed below:
> > In the end of section 4 in [1], the escaping set of Adam and SGD can be approximated by $W_{\text{ADAM}} = \\{y \in \mathbb{R}^d  | y^\top H(\theta^*) y \geq S^2 h_f^* \\}$ and $W_{\text{SGD}} = \\{y \in \mathbb{R}^d  |  y^\top \hat{H}(\theta^*) y \geq S^2 h_f^* \\}$ respectively, and hence their compliments could be represented as $W_{\text{ADAM}}^c = \\{y \in \mathbb{R}^d  |  y^\top H(\theta^*) y < S^2 h_f^* \\}$ and $W_{\text{SGD}}^c = \\{y \in \mathbb{R}^d  |  y^\top \hat{H}(\theta^*) y < S^2 h_f^* \\}$. Furthermore, The eigenvalues of positive definite matrices $H(\theta^*)$ and $\hat H(\theta^*)$ are denoted as $\{\lambda_1 \geq \dots \geq \lambda_d\}$ and $\{\lambda_1 \varsigma_1^2 \geq \dots \geq \lambda_d \varsigma_d^2\}$ respectively. Then, based on the volume formula of $d$-dimensional ellipsoids, we have that $V(W_{\text{ADAM}}^c) = \zeta/(\prod_{i=1}^d \lambda_i)$ where $\zeta = 2d^{-1} (\pi S/h_f^*)^{d/2} g^{-1}(d/2)$ with a gamma function $g$, and $V(W_{\text{SGD}}^c) = \zeta/(\prod_{i=1}^d \varsigma_i^2 \lambda_i)$ as well. (Notice that the volume formula for a $d$-dimensional ellipsoid $\\{y \in \mathbb{R}^d  |  \sum_{i=1}^d y_i^2/c_i < V \\}$ is $(\pi^{d/2} \prod_{i=1}^d c_i)/(g(d/2+1))$, and $W_{\text{ADAM}}^c$ is identical to the ellipsoid $\\{y \in \mathbb{R}^d  |  \sum_{i=1}^d y_i^2 \lambda_i < S^2 h_f^* \\}$ after an orthogonal rotation. And orthogonal rotation will keep the volume
> > invariant. The same argument could be implemented for $W_{\text{SGD}}^c$ as well.) Zhou et al. made a mistake in the last paragraph of section 4 as they erroneously calculated the volume of ellipsoids $W_{\text{ADAM}}^c$ and $W_{\text{SGD}}^c$. Specifically, they mistakenly got the volume of $W_{\text{ADAM}}^c$ and $W_{\text{SGD}}^c$ as $V(W_{\text{ADAM}}^c) = \zeta (\prod_{i=1}^d \lambda_i)$ and $V(W_{\text{SGD}}^c) =\zeta (\prod_{i=1}^d \varsigma_i^2 \lambda_i)$, but the true values should be $\zeta/(\prod_{i=1}^d \lambda_i)$ and $\zeta/(\prod_{i=1}^d \varsigma_i^2 \lambda_i)$ instead as we mentioned here. Therefore, they finally reach a reverse conclusion, and the correct conclusion should be that the escaping time of SGD is longer than that of Adam.  Considering that SGD exhibit better generalization when testing compared to Adam when they reach similar training performance, the escaping time of an optimizer under such theoretical framework is generally positively correlated to its generalization performance. Hence the fact that the escaping time of AdaMomentum is longer also implies its better generalization performance.
> >
> >
> > **Q2**:
> > > The authors stated ImageNet experiment in [2] is "unfair and unconvincing" because they used a blacklisted version dataset. I quickly searched online, blacklisting wrongly labeled data is recommended in imagenet devkit and and fastai official repository.
> >
> > **A2**: Thanks for your careful review and effort. Please note that we are **NOT** saying that blacklisting wrongly labeled data is not recommendeded or forbidden for ImageNet training. We are saying that such practice is **NOT** standard practice in optimizer papers and the (potential ) practice of the AdaBelief paper [2] using blacklisting only for AdaBelief is unfair . This is because in AdaBelief paper [2], all their baseline papers' results are "the best from literature" and the literature did **NOT** use blacklisting.  In fact, either in AdaBelief paper [2] or their officially released code (https://github.com/juntang-zhuang/Adabelief-Optimizer/blob/update_0.2.0/PyTorch_Experiments/imagenet/main_when.py), they did **NOT** mention anything about blacklisting wrongly labeled data or which wrongly labeled data they have blacklisted. **In the ImageNet experiments of our paper, we do NOT blacklist wrongly labeled data for all the compared optimizers and our optimizer, which we believe is fair and is not a contradiction to [1].** To further demonstrate, we perform additional experiments on AdaMomentum under our paper setting when blacklisting the wrongly labeled images using the approach in fastai https://github.com/fastai/imagenet-fast/blob/master/imagenet_nv/blacklist.sh, the Top 1 accuracy of AdaMomentum rises up to around 70.77, higher than without blacklisting. This suggests that blacklisting will increase performance on ImageNet.

---

> > > ### Author Response · Authors · 2021-11-30
> > > **Reponse to Updated review of Reviewer mUyQ (2/3)**
> > >
> > > Besides, note that the saying of blacklisting is just one possible practice of AdaBelief. The authors did **NOT** claim they used blacklisting. The original reply of the author of AdaBelief on the public Github issue link https://github.com/juntang-zhuang/Adabelief-Optimizer/issues/50 is `Hi, I don't have pretrained weights available now, too long ago. If I remember correctly, there might be some wrongly labled validation images in ImageNet12, and you will need to "blacklist" those images. That might cause the difference. Also random seed might be a reason.`  The above unclear explanation of the reported result makes the results reported in AdaBelief paper [2] for ImageNet unconvincing. **That accounts for why we report the results of our run of AdaBelief under our settting  in our paper.**
> > >
> > > **Q3**: On the reproducibility of GAN experiment.
> > >
> > > **A3**: Thanks for your attention and efforts. Again, **we provide the original training log files** of the $5$ independent runs in our experiments, with best FID scores $11.967$, $11.741$, $12.391$, $12.054$, $12.150$. The average FID score is 12.061, which is what we report in our paper. **We also provide the original checkpoints** of the trained generator and discriminator networks  corresponding to the results respectively. **Both the logs and the checkpoints are on anonymous google drive link** https://drive.google.com/drive/folders/1aSXSTrzq6jGw46SLdG9vPT0KiF6z59pR?usp=sharing (we name our optimizer as Adam_m in code-level when doing experiments at that time, the log and checkpoint files do not contain any personal information). The working platform is Ubuntu 18.04.6 LTS with Nvidia GeForce GTX Titan XP Graphic Card GPUs and Intel(R) Core(TM) i7-6850K CPU @ 3.60GHz CPUs and the CUDA version is 10.1.
> > >
> > > **We believe the above provided materials are strong evidence that our experiments on GAN can be reproduced. Note that training logs may be editted but checkpoints of the model are not able to be forged.**
> > >
> > > **Q4**:
> > > > Regarding my experiments with a large batch ( I won't lower my rating by comparing the author's results with my baseline of 72% accuracy, because I have not confirmed it's the same across platforms, but I don't think the authors' argument is convincing). For my settings, I checked I did not use different augmentation. I admit my setting is different from yours, but I don't see any of my settings would unfairly increase accuracy, on the contrary, the large batch in my setting typically decreases accuracy.
> > >
> > > **A4**: Thanks for the detailed clarification. However, our point is that your claimed result of AdaBelief using a different setting is not defensible enough to make our experimental results untenable. This is because generally the comparison of DL optimziers need to guarantee the setting of different optimizers to be same. Also, You claimed your setting is different from us but you just did the experiment for AdaBelief. How about the performance of AdaMomentum and other competitor optimizers in your setting (super large batch size)? Besides, Why do the large batch in your setting decrease accuracy? Did you do experiments to confirm it?
> > >
> > > Furthermore, if you `won't lower your rating by comparing our results with your baseline of 72% accuracy` using AdaBelief, why do you list it as a main weakness in your review? We are confused of this point.
> > >
> > > **Q5**: Regarding $\epsilon$.
> > >
> > > **A5**: Thanks for your careful review. Please note that the discussion of $\epsilon$ (the subsection `Benefits of changing the location of` $\epsilon$) in our paper is mainly comparing the different locations of $\epsilon$ in AdaMomentum, not AdaMomentum with Adam. Note that in our paper in both $\hat{v}_t$ and $\hat{v}_t'$ the summed item is $m_i$s not $g_i$s. Our point is that **when using the same $\epsilon$ value, the current formulation of AdaMomentum is better than the version of AdaMomentum with $\epsilon$ outside the $\sqrt{\cdot}$**.
> > >
> > > Even if we presume your statement is correct, we perform additional experiments in CNN experiments setting $\epsilon=1e-4$ for Adam on CIFAR-10 using VGG-16 as you said, and the averaged accuracy result is 93.56$\pm$0.15. The performance is far below AdaMomentum which result is 94.80$\pm$0.10.
> > >
> > > **Q6**:
> > > > Toy example. My main point is not whether the scale is $\frac{1}{1-\beta_1^t}$ or not, my point is $m_t$ is smaller than $g_t$ by a factor (up to $1-\beta_1$
> > > approximately), then the stepsize is larger by up to $\frac{1}{1-\beta_1}$, this is a big factor if your toy example only runs for a hundred steps. If you manually increase the stepsize by this factor for Adam, it will also converge much faster.

---

> > > > ### Author Response · Authors · 2021-11-30
> > > > **Reponse to Updated review of Reviewer mUyQ (3/3)**
> > > >
> > > > **A6**: Thanks very much for your clarification. However, it is quite confusing that in your initial review you try to caculate the scale as $\frac{1}{1-\beta_1^t}$ and then you say `your point is not whether the scale is` $\frac{1}{1-\beta_1^t}$` or not`. Then your so-called scale suddenly turn to $1-\beta_1$ (approximately) without any proof. Note that in optimization literature, calculating a factor value approximately is not generally allowed because optimization process is very sensitive and a little error can accumulate to very much as iterations go.
> > > >
> > > > Even if we presume your statement is correct, we perform additional toy experiment and manually increase the learning rate of Adam by your suggested scale $\frac{1}{1-\beta_1}$. The visualization results are summarized on anonymous link https://imgur.com/a/KulGyRL and we can see that such big effective stepsize will make Adam fail to converge to optimum, and is still worse than AdaMomentum.
> > > >
> > > >
> > > > ----------------------------------------------------------------
> > > > To summarize, we would like to clarify that we are not deliberately claiming other papers to be problematic. The reason we claim that part of the result in [1] is problematic and the ImageNet experiment result of [2] is not convincing, is that you think our paper results contradicts these two papers. We are trying to explain to you there is actually no contradiction, and to illustrate the truth and the reasons, or at least, what we believe to be the truth or the reasons (of course, we might be wrong). There are many great published papers (e.g. Adam [3]) that are problematic to some extent, but this does not prevent the papers from generating its influences due to the new idea and improvements they bring about. We believe the purpose of academic conferences is to share ideas and push forward the development of knowledge. Therefore we believe criticism of previous papers should be promoted, not curbed.
> > > >
> > > > Also, we think we have provided enough evidence to support our rebuttal in the above, along with the previous rebuttal. You say `the authors of the papers you criticized won't be able to participate in the discussion`. However, note that this is OpenReview, where everybody can paticipate in the discussion and we very welcome such discussions.
> > > >
> > > > Thanks again for your careful review and valuable comments. We believe we have addressed most of your concerns. If you have further questions, feel free to ask and we are willing to have further discussions with you.
> > > >
> > > >
> > > > ### References
> > > > [1] Pan Zhou et al.Towards theoretically understanding why sgd generalizes better than adam in deep learning. NeurIPS 2020.
> > > > [2] Juntang Zhuang et al. AdaBelief Optimizer: Adapting Stepsizes by the Belief in Observed Gradients. NeurIPS 2020.
> > > > [3] Diederik P. Kingma and Jimmy Ba. Adam: A method for stochastic optimization. In ICLR, 2015.

---

> ### Comment · Reviewer_mUyQ · 2021-11-30
> **Updated review**
>
> 1. Contradiction with [1]. I don't think you are correct to flip the conclusion of [1] and state SGD has a longer escape time than Adam, at least this is against the experimental validation in [1].
> 2. This point is on the processing of validation itself and clarification on your claim about AdaBelief, rather than on your method.
> 3. I have uploaded my checkpoint along with log. As you have said, checkpoints can not be forged, so I trust what I got using your code.
> https://www.dropbox.com/sh/hrf8uzrh8uy786g/AAAR6BkMEGeoqpw1GOVpgOVVa?dl=0
> https://www.dropbox.com/sh/l7i24isu5vz6eh0/AAAnLYi4Eytrf3JC4tv79yqLa?dl=0
> 4. This point (in the previous post) is a reply to your previous comment which implies my setting is unfair. BTW, [2] is a very famous paper saying large batch typically generates worse performance. PS, I ran AdaBelief with a large batch long before reading your paper, it's not my job as a reviewer to run all your experiments.
> 5. When you compare where to put $\epsilon$, you should note that inside and outside sqrt use different default values.
> 6. Try to replace the $v_t$ in Adam by $v_t \times (1-\beta_1^t)^2$. That's how I would make Adam take a similar stepsize schedule as AdaMomentum. I'm pretty sure it would be much faster than Adam.
>
> The reason for my score: my score is mainly based on points 1 and 3, that's the most important reason, because I see both a clear theoretical contradiction and I could not reproduce the results in experiments. If I want to take other points into account this would make the score even lower. I want to end the discussion here, since I have spent much more time on your paper than other papers combined together, and I have spent a long time running your code. Believe it or not, I provide the log and checkpoint, what I got is way different from you reported for the GAN experiments.
>
> [1] Zhou, Pan, et al. "Towards theoretically understanding why sgd generalizes better than adam in deep learning." arXiv preprint arXiv:2010.05627 (2020).
> [2] On Large-Batch Training for Deep Learning: Generalization Gap and Sharp Minima

---

> > ### Author Response · Authors · 2021-12-01
> > **Response to updated review of Reviewer mUyQ**
> >
> > We thank you for your updated comments. Here is our response to your comments.
> >
> > 1. If you don't think we are right, please give a reason or point out where we are wrong. Simply saying something is not correct without giving a reason, to our way of thinking, is not tenable.
> > 2. Our point is that experiment on ImageNet of AdaBelief paper [1] is unfair if they really used blacklisting wrongly labeled data validation data set for AdaBelief only. This is because the baselines did not blacklist wrongly labeled data. Our previous additional experiments showing AdaMomentum can improve performance using blacklisting further confirms the unfairness of the potential practice of using blacklisting only on the proposed optimizer of AdaBelief.
> > 3. Again, we have provided our checkpoints and the training log of the 5 independent runs of our GAN experiments of AdaMomentum https://drive.google.com/drive/folders/1aSXSTrzq6jGw46SLdG9vPT0KiF6z59pR?usp=sharing . We can not guarantee that maybe on other platforms the performance may vary. **However, we do think our provided materials at least guarantee that our reported number of GAN can be reproduced.**
> > 4.  Our point is that your practice of using AdaBelief (a compared baseline in our paper) **using a different setting** to achieve a (you claimed) "super high accuracy" (72.2 with Res-18) is not tenable enough to be one main weakness listed for our paper.
> > 5.  Our point is that when using the same $\epsilon$ value, the current formulation of AdaMomentum is better than the version of AdaMomentum with $\epsilon$ outside the $\sqrt{\cdot}$.
> > 6. The replacement of $v_t$ in Adam as you say will lead to a new method, not Adam itself. Also, you said you are pretty sure but did not give any supporting evidence, which does not convince us.
> >
> > To summarize, we are quite disappointed throughout the rebuttal with you as you are always trying to attack our paper without giving any convincing reasons and helpful advice to improve the paper. Of course a reviewer can find holes and challenge the reviewed paper, **but the ultimate goal is to help improve the paper for a better shape** to our way of thinking. From the other three reviewers, we get a lot of valuable advice and will revise the paper according to their comments. But from yours, we find it hard to draw anything that can help us improve.
> >
> > **Finally, we want to mention that we have a strong feeling that you want to support and rectify AdaBelief paper [1] (a baseline optimizer method) in some way. Our previous detailed replies have exhaustively demonstrated that the experiment of AdaBelief on ImageNet in the original is not convincing. In fact, there are many (closed) issues on the official code of AdaBelief and many people doubt the fairness and reproducibility of ImageNet experiments for AdaBelief (see closed issues of AdaBelief repository https://github.com/juntang-zhuang/Adabelief-Optimizer/issues?q=is%3Aissue+is%3Aclosed). In our experiments, we find that AdaBelief always fails to outperform SGD and AdaMomentum on vision tasks, and sometimes even far from close. We have no idea why you so strongly support AdaBelief [1] .**
> >
> > [1] Juntang Zhuang et al. AdaBelief Optimizer: Adapting Stepsizes by the Belief in Observed Gradients. NeurIPS 2020.

---

> > > ### Comment · Reviewer_mUyQ · 2021-12-01
> > > **Response to authors**
> > >
> > > Since the authors doubt my comments related to AdaBelief, fine, I'll just talk about AdaMomentum below.
> > >
> > > 1. Theory:      There's a serious flaw in your claim. As I have mentioned in the initial review, for AdaMomentum, pro: longer escape time gives a higher probability to stay at a wide minima. con: if due to randomness it jumps to a sharp minima, longer scape time will also make it harder to get out of the sharp minima.    Example: Denote the sharp minima as $A$ and wide minimia as $B$, the transition probability as $P_{A,A}, P_{A,B}, P_{B,A}, P_{B,B}$. Consider two optimizers 1 and 2, $P_{B,B}^1 > P_{B,B}^2$ does not imply the stationary distribution $\pi_B^1>\pi_B^2$. Your analysis omitted the con part.
> > >
> > > 2.  Experiments on GAN:   If you provide your log and checkpoint to say your claim is convincing, then why my claim and providing the log and checkpoint is unconvincing?    From the initial review phase, I suggested the authors to check your code version in case you ran and submitted different versions.
> > >
> > > 3.  One $\epsilon$, why do you think it's fair to compare $\sqrt{10^{-16}}$ with $10^{-16}$?
> > >
> > > 4.  On the stepsize factor $1-\beta_1^t$:   I have mentioned many times, that AdaMomentum uses EMA of square of **biased** EMA of gradient. AdaMomentum: $v_t = v_{t+1} \beta_2 + (1-\beta_2) m_t^2 $, Adam: $v_t = v_{t+1} \beta_2 + (1-\beta_2) g_t^2 $. Note that $\mathbb{E}m_t = (1-\beta_1^t) \mathbb{E} g_t$. The authors have claimed this is not accurate, sure it is inaccurate, but useful, otherwise why do you keep the same bias correction of Adam in AdaMomentum?
> > >
> > > 5. Experiments on the stepsize factor $1-\beta_1^t$: Replace this line ```exp_avg_var.mul_(beta2).addcmul_(exp_avg, exp_avg, value=1 - beta2)``` with ```exp_avg_var.mul_(beta2).addcmul_(grad * bias_correction1, grad*bias_correction1, value=1 - beta2)```, on LSTM exepriment where you claimed the big improvement of AdaMomentum, just with a bit of tuning on lr, I got the following results: 80.89 PPL for 1-layer LSTM, 64.77 for 2-layer LSTM, same range as AdaMomentum. Which validates my comment that, using EMA of EMA is not the key, the key is you did not do bias correction for $m_t$ when deriving $v_t$, which is an "implicit" learning rate schedule written into optimizer. Keeping the update in Adam by account for this implicit lr schedule does equally good.
> > >
> > > 6. Experiments on toy: again I used your code, modified Adam as I said in point 4 above, it's much faster than AdaMomentum. See gif https://www.dropbox.com/s/t3nki4rnef6152v/traj_camel.gif?dl=0 and code https://www.dropbox.com/s/5gvijrenkb719y3/toy_visualize.py?dl=0
> > >
> > > Finally, I'm extremely disappointed by the authors. The authors implied that I maliciously down-tuned their results on GAN, and cherry-picked the results on AdaBelief. However, (1) if I down-tune your results on GAN, why would I ask you to check your code version and I offered to re-run it (2) If I want to down-rate your paper by defending AdaBelief, I could have found much more flaws that I said, for example, (a) your DenseNet results is clearly below reported by AdaBelief ( there's no data issue and I'm 100% sure the reported results in Adabelief is achievable) (b) your Transformer experiments is again much lower than reported in AdaBelief and AdaHessian, I checked the reason is that the hyper-params you used for AdaBelief and Adam is actually for architecture-v1, but in experiments you used architecture-v2.
> > >
> > > The reason that I did not mention the above flaws is that I think these results are on toy problems, and as long as within a reasonable range it's fine. The reason I insist on GAN is also that it's not a small experiment to me, that's why I check the results by running your code.
> > >
> > > I want to remind the authors to show some respect to others, this is my worst review experience ever, please do not take technical comments as malicious attacks, and please do not show your achievements by meaninglessly attacking others' work.

---

### Author Response · Authors · 2021-11-22
**Fixed Typos in Our Paper and Appreciate Your Comments**

Dear Reviewers,

We have fixed some typos in Sec. 3.1 where we originally confused $\hat{v}_t$ and $\hat{v}_t'$. The fixed places have been marked red in our uploaded revision.

We sincerely appreciate your valuable comments that help us improve the paper. If you have more questions or concerns about our paper or response, please feel free to let us know. We are happy to discuss with you.

---

### Comment · Area_Chair_mNpY · 2021-12-02
**Keep the conversation civil**

Dear Authors and Reviewers,

Please keep the conversation civil.
Also, I don't think further discussion is needed on this paper and it does not seem to be fruitful anyway.

Thanks,
AC

---

> ### Author Response · Authors · 2021-12-02
> **Appreciate Your Time and Efforts**
>
> Dear AC and Reviewers,
>
> We thank the AC for your time and efforts in handling the review process.
> We thank the reviewers for giving valuable comments to make our paper better.
>
> We will not participate in any further discussion.
>
> Best,
> Authors

---

### Decision · Program_Chairs · 2022-01-20

**Decision:**

Reject

**Comment:**

The paper proposes to substitute the gradient in the second moment estimation term with the "momentumized" version, arguing that it improves both optimization and generalization. Some theoretical results are shown as well as empirical results.

The paper has been widely discussed by the reviewers and several weak points have been raised. Let me list some of the most important ones.
- The theory appears to be incremental and overall very weak. The authors themselves acknowledged that this "is not a pure optimization theory paper". In details, the generalization analysis is a straightforward extension of Zhou et al. [NeurIPS 2020], while the optimization analysis inherits all the known weaknesses of previous similar analysis in deep learning optimization papers. In particular, *none* of the following is correct: the use of a regret analysis for a stochastic non-convex optimization algorithm, the assumption of bounded iterates, Assumption 5, the assumption in Theorem 2 on $\alpha_t/\sqrt{v_t}$. The fact that similar mistakes were done in previous papers does not make them correct: The community should aspire at doing better not at reiterating known mistakes.
- On $\epsilon$: the reviewers correctly pointed out that moving $\epsilon$ under the square root and not changing its value is not fair. The answers of the authors on this point were unconvincing.
- Doubts on empirical results: it seems that not all the possible hyperparameters of the baselines were properly tuned. For example, despite being common practice, epsilon should also be tuned, see for example the experiments in Agarwal et al. 2020.

I didn't consider the discussion on AdaBelief because only marginally relevant to this paper.

Overall, the paper does not seem interesting from a theoretical point of view and its empirical comparison cannot be fully trusted for the presence of some weaknesses.